# Climatological impact of the Brewer-Dobson Circulation on the $N_2O$ budget in WACCM, a chemical reanalysis and a CTM driven by four dynamical reanalyses

Daniele Minganti[1], Simon Chabrillat[1], Yves Christophe[1], Quentin Errera[1], Marta Abalos[2], Maxime Prignon[3], Douglas E. Kinnison[4], and Emmanuel Mahieu[3]

[1]Royal Belgian Institute for Space Aeronomy, BIRA-IASB, Brussels, 1180 Belgium
[2]Universidad Complutense de Madrid, Madrid, Spain
[3]Institute of Astrophysics and Geophysics, University of Liège, 4000 Liège, Belgium
[4]National Center for Atmospheric Research, Boulder, CO, USA

**Correspondence:** Daniele Minganti (daniele.minganti@aeronomie.be)

**Abstract.**

The Brewer-Dobson Circulation (BDC) is a stratospheric circulation characterized by upwelling of tropospheric air in the Tropics, poleward flow in the stratosphere, and downwelling at mid and high latitudes, with important implications for chemical tracers distribution, stratospheric heat and momentum budgets and mass exchange with the troposphere. Since the photochem-

ical losses of nitrous oxide ($N_2O$) are well-known, model differences in its rate of change are due to transport processes that can be separated in the mean residual advection and the isentropic mixing terms in the Transformed Eulerian Mean (TEM) framework. Here the climatological impact of the stratospheric BDC on the long-lived tracer $N_2O$ is evaluated through a comparison of its TEM budget in the Whole Atmosphere Community Climate Model (WACCM), in a chemical reanalysis of the Aura Microwave Limb Sounder version 2 (BRAM2) and in a Chemistry-Transport Model (CTM) driven by four modern

reanalyses (ERA-Interim, JRA-55, MERRA and MERRA2). The effects of stratospheric transport on the $N_2O$ rate of change, as depicted in this study, have not been compared before across this variety of datasets and never investigated in a modern chemical reanalysis. We focus on the seasonal means and climatological annual cycles of the two main contributions to the $N_2O$ TEM budget: the vertical residual advection and the horizontal mixing terms.

The $N_2O$ mixing ratio in the CTM experiments has a spread of approximately $\sim 20\%$ in the middle stratosphere, reflecting

the large diversity in the mean Age of Air obtained with the same CTM experiments in a previous study. In all datasets the TEM budget is well-closed and the agreement between the vertical advection terms is qualitatively very good in the Northern Hemisphere, and good in the Southern Hemisphere except above the Antarctic region. The datasets do not agree as well with respect to the horizontal mixing term, especially in the Northern Hemisphere where horizontal mixing has a smaller contribution in WACCM than in the reanalyses. WACCM is investigated through three model realizations and a sensitivity test

using the previous version of gravity waves parameterization. The internal variability of the horizontal mixing in WACCM is large in the polar regions, and comparable to the differences between the dynamical reanalyses. The sensitivity test has a relatively small impact on the horizontal mixing term, but significantly changes the vertical advection term and produces a less

realistic $N_2O$ annual cycle above the Antarctic. In this region, all reanalyses show a large wintertime $N_2O$ decrease, which is mainly due to horizontal mixing. This is not seen with WACCM, where the horizontal mixing term barely contributes to the TEM budget. While we must use caution in the interpretation of the differences in this region, where the reanalyses show large residuals of the TEM budget, they could be due to the fact that the polar jet is stronger and not tilted equatorward in WACCM compared with the reanalyses.

We also compare the inter-annual variability in the horizontal mixing and the vertical advection terms between the different datasets. As expected, the horizontal mixing term presents a large variability during austral fall and boreal winter in the polar regions. In the tropics, the inter-annual variability of the vertical advection term is much smaller in WACCM and JRA-55 than in the other experiments. The large residual in the reanalyses and the disagreement between WACCM and the reanalyses in the Antarctic region highlight the need for further investigations on the modeling of transport in this region of the stratosphere.

## 1 Introduction

The Brewer-Dobson Circulation (BDC, Dobson et al., 1929; Brewer, 1949; Dobson, 1956) in the stratosphere is characterized by upwelling of tropospheric air to the stratosphere in the Tropics, followed by poleward transport in the stratosphere and extratropical downwelling. For tracer-transport purposes the BDC is often divided into an advective component, the residual mean meridional circulation (hereafter residual circulation), and a quasi-horizontal two-way mixing which causes net transport of tracers, not of mass (Butchart, 2014).

The BDC is driven by tropospheric waves breaking into the stratosphere (Charney and Drazin, 1961), which transfer angular momentum and force the stratosphere away from its radiative equilibrium. This is balanced by a poleward displacement of air masses, which implies tropical upwelling and extra-tropical downwelling (Holton, 2004). The residual circulation can be further separated in three branches: the transition, the shallow and the deep branch (Lin and Fu, 2013). The transition branch encompasses the upper part of the transition layer between the troposphere and the stratosphere (the tropical tropopause layer, Fueglistaler et al., 2009). The shallow branch is an all year-round lower stratospheric two-cell system driven by breaking of synoptic-scale waves, and the deep branch is driven by Rossby and gravity waves breaking in the middle and high parts of the stratosphere during winter (Plumb, 2002; Birner and Bönisch, 2011). The contributions of different wave types to the driving of the BDC branches has been quantified using the downward control principle, which states that the poleward mass flux across an isentropic surface is controlled by the Rossby or gravity waves breaking above that level (Haynes et al., 1991; Rosenlof and Holton, 1993), and using eddy heat flux calculations as an estimate of the wave activity from the troposphere (e.g., Newman and Nash, 2000).

The quasi-horizontal two-way mixing is generated by two-way transport due to the adiabatic motion of Rossby waves. In the stratosphere this motion is ultimately combined with the molecular diffusion which makes the total process irreversible (Shepherd, 2007). The two-way mixing is stronger in a specific latitudinal region of the winter stratosphere, the 'surf zone' (McIntyre and Palmer, 1983), and in the subtropical lower stratosphere all year round (e.g. Fig.1 of Bönisch et al., 2011). The mixing process homogenizes the tracer concentration in the surf zone and creates sharp tracer and Potential Vorticity (PV)

gradients on its edges (in the subtropics and at the polar vortex edge), indicating an inhibition of mixing. For this reason the subtropics and the polar vortex edge are often called transport barriers (Shepherd, 2007).

The BDC plays major roles in controlling the spatial and temporal distributions of chemical tracers such as ozone, water vapor, aerosols, and greenhouse gases, as well in coupling stratospheric processes with the climate system (Riese et al., 2012; Butchart, 2014; Tweedy et al., 2017). The natural variability of the atmosphere largely influences the BDC (Hardiman et al., 2017). All three branches of the BDC are affected by changes in sea surface temperatures and El Niño Southern Oscillation (Yang et al., 2014; Diallo et al., 2019), as well as the phase of the Quasi Biennal Oscillation (QBO, Diallo et al., 2018), and the Arctic oscillation (Salby and Callaghan, 2005).

Modeling studies predict an acceleration of the BDC over the last decades and the twenty-first century due to the increase in well-mixed greenhouse gases (Butchart et al., 2010; Hardiman et al., 2014; Palmeiro et al., 2014) and ozone-depleting substances (Polvani et al., 2018), but these results cannot be evaluated easily because the BDC cannot be observed directly (Butchart, 2014). Observational studies over short periods (typically 2003-2012) show significant evidence of a changing BDC in the boreal lower stratosphere (Schoeberl et al., 2008; Stiller et al., 2012; Hegglin et al., 2014; Mahieu et al., 2014; Haenel et al., 2015), but balloon-borne observations of $SF_6$ and $CO_2$ in the northern mid-latitudes show a non-significant trend of the deep branch of the BDC in the past decades (Engel et al., 2009, 2017). The difficulty to derive observational trends in the BDC can be partly attributed to the spatial and temporal sparseness of the observations, together with its large dynamical variability and the uncertainty of trends derived from non-linearly increasing tracers (Garcia et al., 2011; Hardiman et al., 2017; Fritsch et al., 2020). Before investigating multi-decadal changes of the BDC, it is important to perform an accurate evaluation of its climatological state and inter-annual variability, which is the aim of this paper.

In this study we use $N_2O$ as a tracer to study the BDC. $N_2O$ is continously emitted in the troposphere (with larger abundances in the Northern Hemisphere, NH), and transported into the stratosphere where it is destroyed by photodissociation and, to a lesser extent, by reaction with $O(^1D)$. The estimated lifetime of $N_2O$ is approximately 120 years, which makes it an excellent long-lived tracer for transport studies in the middle atmosphere (Brasseur and Solomon, 2006; Seinfeld and Pandis, 2016).

We use the Transformed Eulerian Mean (TEM, Andrews et al., 1987) analysis to separate the local rates of change of $N_2O$ due to transport and chemistry (Randel et al., 1994). The transport term can be further separated into the contribution of isentropic mixing and residual advection as done previously for $O_3$ and CO (Abalos et al., 2013). The isentropic mixing and the residual advection can be additionally separated in their horizontal and vertical contributions. In the tropical lower stratosphere, the distinction between vertical and horizontal transport is important, as they impact differently the seasonality of $N_2O$ in the northern and southern tropics (Tweedy et al., 2017). We choose to focus our study on the horizontal mixing and vertical advection, because their magnitudes are larger than the vertical mixing and the meridional residual advection in most of the stratosphere.

Chemistry Climate Models (CCMs) include the full representation of dynamical, radiative, and chemical processes in the atmosphere and their interactions. In particular, they combine the feedbacks of the chemical tracers on the heat budget and dynamics, that ultimately affects tracer transport. We use the Whole Atmosphere Community Climate Model version 4 (WACCM, Garcia et al., 2017) to simulate the $N_2O$ TEM budget in the stratosphere for the 2005-2014 period. WACCM has been widely

used for studies of tracers transport in the stratosphere and upper troposphere based on the TEM analysis (e.g. Abalos et al., 2017). WACCM simulations of the climatological $N_2O$ over the 2005-2014 period have also been evaluated favourably with satellite observations in the stratosphere (Froidevaux et al., 2019).

In order to assess their representation of the atmopsheric processes, CCMs are often compared to reanalyses (e.g. Gerber et al., 2010). Reanalysis products merge dynamical atmospheric observations (e.g. surface pressure, wind, temperature) with a global forecast model using an assimilation scheme to offer the best reproduction of the past climate. They provide a multivariate, consistent record of the global atmospheric state. Reanalyses are made using different assimilation methods and forecast models (Cameron et al., 2019), and they are often compared among each other and with CCMs (Rao et al., 2015). The SPARC (Stratosphere-troposphere Processes And their Role in Climate) Reanalysis Intercomparison Project (S-RIP) coordinates the intercomparison of all major global atmospheric reanalyses and provides reports to document these results (Fujiwara et al., 2017; Long et al., 2017).

Meteorological fields from reanalyses are often used to drive Chemistry-Transport Model (CTM) in order to study the BDC through a common diagnostic, namely the Age of Air (AoA, Waugh and Hall, 2002), and simulate realistic distributions of chemical tracers (Monge-Sanz et al., 2012; Ménard et al., 2020). Thanks to their simplicity, CTMs are useful to compare different reanalyses within the same transport framework, thereby contributing to the study of the BDC in S-RIP (chapter 5; see Fig. 1 in Fujiwara et al., 2017). CTMs may use either sigma-pressure levels with a kinematic transport scheme and vertical velocities simply derived from mass conservation, or isentropic levels with a diabatic transport scheme (Chipperfield, 2006). Recent intercomparisons showed that the AoA depends to a large extent on the input reanalysis, both using the kinematic approach (Chabrillat et al., 2018) and the diabatic approach (Ploeger et al., 2019).

Here we use the same CTM as for the kinematic AoA study, i.e. the Belgian Assimilation System of Chemical ObsErvation (BASCOE) CTM. Observations of another long-lived stratospheric tracer, HCFC-22, were recently interpreted with WACCM and BASCOE CTM simulations, showing the interest of this model intercomparison (Prignon et al., 2019). In order to contribute further to the S-RIP BDC activity, four different dynamical reanalyses are used here to drive the BASCOE CTM simulations, compute the $N_2O$ TEM budget and compare its components with the results derived from WACCM. Namely we consider: the European Centre for Medium-Range Weather Forecasts Interim Reanalysis (ERA-Interim, Dee et al., 2011), the Japanese 55-year Reanalysis (JRA55, Kobayashi et al., 2015), the Modern-Era Retrospective analysis for Research and Applications version 1 (MERRA Rienecker et al., 2011), and version 2 (MERRA2 Gelaro et al., 2017).

While dynamical reanalyses do not assimilate observations of chemical compounds, chemical reanalyses achieve this step, and can be used to evaluate CCMs or study differences between instruments using the reanalysis as a transfer tool (Errera et al., 2008; Lahoz and Errera, 2010; Davis et al., 2016). Chemical reanalyses driven by meteorological fields from modern dynamical reanalyses have not been used to study tracer transport in the stratosphere using the TEM framework to our knowledge. WACCM and the CTM experiments are compared with a chemical reanalysis of Aura Microwave Limb Sounder (MLS) using the BASCOE Data Assimilation System (DAS) driven by the ERA-Interim reanalysis (BRAM2, Errera et al., 2019).

To summarize, in this study we analyze the representation of the BDC in WACCM through an analysis of the TEM budget of $N_2O$, and we evaluate the simulation of this budget through comparisons with the BASCOE CTM (driven by four dynamical

reanalyses) and the BRAM2 chemical renalysis. In Section 2 we describe the datasets used in the study and the TEM analysis of $N_2O$. In Section 3 we analyse the seasonal mean patterns of the TEM $N_2O$ budget in each dataset and their differences. Sections 4 and 5 investigate respectively the mean annual cycle and the variability of the $N_2O$ TEM budget terms, with a focus on the differences between the datasets. Section 6 concludes the study with a summary of our findings and possible future research.

## 2 Data and method

This work uses seven datasets that were generated by WACCM, the BASCOE CTM and the BASCOE DAS. Table 1 provides an overview of these datasets and their main differences, and the next three subsections provide details about the models and systems that generated them.

### 2.1 WACCM

WACCM (Garcia et al., 2017) is the atmospheric component of the Community Earth System Model version 1.2.2 (Hurrell et al., 2013), which has been developed by the U.S. National Center of Atmospheric Research. It is the extended (whole atmosphere) version of the Community Atmosphere Model version 4 (CAM4, Neale et al., 2013).

We ran one realization of the public version of WACCM (hereafter WACCM4, Marsh et al., 2013), with a similar setup (e.g. lower boundary conditions) as the CTM experiments; the source code of WACCM4 is available for download at https://svn-ccsm-models.cgd.ucar.edu/cesm1/release_tags/cesm1_2_2cesm1_2_2. In this study we also use 3 realizations of the REF-C1 simulation used in the SPARC Chemistry-Climate Model Initiative (CCMI, Morgenstern et al., 2017). The CCMI experiments, hereafter WACCM-CCMI, differ from WACCM4 for the modified gravity waves parameterization and the updated heterogenous chemistry (Garcia et al., 2017). The inclusion of WACCM4 allows us to make a sensitivity test for the impact of the modified gravity waves parameterization on the simulation of the $N_2O$ trasport (see Sect. 4 for detailed analysis). We use three-dimensional daily-mean output over the 2005-2014 period to allow a fair comparison with the BRAM2 dataset (see Sec. 2.3 for detailed analysis). WACCM has a longitude-latitude grid of 2.5°x1.9°and 66 vertical levels ranging from the surface to about 140 km altitude. The vertical coordinate is hybrid-pressure, i.e. terrain-following below 100 hPa and purely isobaric above. The vertical resolution depends on the height: it is approximately 3.5 km above 65 km, 1.75 km around the stratopause (50 km), 1.1-1.4 km in the lower stratosphere (below 30 km), and 1.1 km in the troposphere. The time step for the physics in the model is 30 minutes.

The physics of WACCM is the same as CAM4 and the dynamical core is a finite volume with a horizontal discretization based on a conservative flux-form semi Lagrangian (FFSL) scheme (Lin, 2004). The gravity wave parameterization accounts for momentum and heat deposition separating orographic and non-orographic sources. The orographic waves are modified according to Garcia et al. (2017), while non-orographic waves are parameterized depending on the convection and the fronto-genesis occurrence in the model (Richter et al., 2010).

In this study, the considered WACCM versions are not able to internally generate the Quasi Biennial Oscillation (QBO, see e.g. Baldwin et al., 2001). Thus, the QBO is nudged by a relaxation of stratospheric winds to observations in the Tropics (Matthes et al., 2010). The solar forcing uses the Lean et al. (2005) approach.

WACCM includes a detailed coupled chemistry module for the middle atmosphere based on the Model for Ozone and Related Chemical Tracers, version 3 (MOZART-3) (Kinnison et al., 2007; Marsh et al., 2013). The species included within this mechanism are contained within the $O_x$, $NO_x$, $HO_x$, $ClO_x$ and $BrO_x$ chemical families, along with $CH_4$ and its degradation products. In addition, 20 primary non-methane hydrocarbons and related oxygenated organic compounds are represented along with their surface emission. There is a total of 183 species and 472 chemical reactions; this includes 17 heterogeneous reactions

on multiple aerosol types, i.e. sulfate, nitric acid trihydrate, and water-ice. In WACCM-CCMI the heterogeneous chemistry is updated by Solomon et al. (2015).

## 2.2   BASCOE CTM

The BASCOE data assimilation system (Errera et al., 2019) is built on a Chemistry-Transport Model, which consists in a kinematic transport module with the FFSL advection scheme (Lin and Rood, 1996) and an explicit solver for stratospheric

chemistry, comprising 65 species and 243 reactions (Prignon et al., 2019). Chabrillat et al. (2018) explain in detail the pre-processing procedure that allows the BASCOE CTM to be driven by arbitrary reanalysis datasets, and the set-up of model transport. As usual for kinematic transport modules, the FFSL scheme only needs the surface pressure and horizontal wind fields from reanalyses as input, because it is set on a coarser grid than the input reanalyses, and relies on mass continuity to derive vertical mass fluxes corresponding to its own grid. Similar to Chabrillat et al. (2018), the model is driven by four different

reanalysis datasets on a common, low-resolution latitude–longitude grid ($2.5°$x$2°$), but keeping their native vertical grids. In this way, we avoid any vertical regridding and the intercomparison explicitly accounts for the different vertical resolutions.

    The four input reanalyses are part of the SPARC Reanalysis Intercomparison Project (S-RIP) which is a coordinated inter-comparison of all major global atmospheric reanalyses. They are described in Fujiwara et al. (2017): the European Centre for Medium-Range Weather Forecasts Interim Reanalysis (ERA-Interim, hereafter ERAI; Dee et al., 2011), the Japanese 55-year

Reanalysis (JRA55; Kobayashi et al., 2015), the Modern-Era Retrospective analysis for Research and Applications (MERRA; Rienecker et al., 2011) and its version 2 (MERRA2; Gelaro et al., 2017). ERAI and JRA55 have 60 levels up to 0.1 hPa while MERRA and MERRA2 have 72 levels up to 0.01 hPa. The CTM time step is set to 30 minutes. As for the WACCM experiment, we used the daily mean outputs from the BASCOE CTM over the 2005-2014 period.

## 2.3   BASCOE Reanalysis

BRAM2 is the BASCOE Reanalysis of Aura MLS, version 2, which covers the period August 2004-August 2019 (Errera et al., 2019). For BRAM2, BASCOE is driven by dynamical fields from ERA-Interim, with a horizontal resolution of $3.75°$x$2.5°$longitude-latitude. The vertical grid is represented by 37 hybrid-pressure levels which are a subset of the ERA-Interim 60 levels.

In BRAM2, N$_2$O profiles from the MLS version 4 standard product have been assimilated within the 0.46-68 hPa pressure
ranges (Livesey et al., 2015). This dataset is retrieved from the MLS 190 GHz radiometer instead of the 640 GHz radiometer
in earlier MLS version. The 640 GHz radiometer, which provided a slightly better quality retrieval down to 100 hPa, ceased to
be delivered after August 2013 because of instrumental degradation in the band used for that retrieval. To avoid any artificial
discontinuity due to switching from one product to the other in August 2013, BRAM2 has assimilated the 190 GHz N$_2$O
during the whole reanalysis period.

BRAM2 N$_2$O has been validated between 3 and 68 hPa against several instruments with a general agreement between 15 %
depending on the instrument and the atmospheric region (the middle stratosphere or the polar vortex, see Errera et al., 2019).
It is not recommended to use BRAM2 N$_2$O reanalysis outside this pressure range. BRAM2 N$_2$O is also affected by a small
drift of around -4 % between 2005 and 2015 (see also Froidevaux et al., 2019).

## 2.4 TEM diagnostics

For atmospheric tracers the TEM analysis (Andrews et al., 1987) allows to separate the local change of a tracer with volume
mixing ratio $\chi$ in terms due to transport and chemistry (Eq. (1)).

$$\bar{\chi}_t = -\bar{v}^* \bar{\chi}_y - \bar{w}^* \bar{\chi}_z + e^{z/H} \boldsymbol{\nabla} \cdot \boldsymbol{M} + \bar{S}, \tag{1}$$

where $\chi$ is the volume mixig ratio of N$_2$O, and $\boldsymbol{M}$ is the eddy flux vector, defined as:

$$M^{(y)} \equiv -e^{-z/H}(\overline{v'\chi'} - \overline{v'\theta'}\bar{\chi}_z/\bar{\theta}_z), \tag{2a}$$

$$M^{(z)} \equiv -e^{-z/H}(\overline{w'\chi'} + \overline{v'\theta'}\bar{\chi}_y/\bar{\theta}_z). \tag{2b}$$

$\bar{v}^*$ and $\bar{w}^*$ are the meridional and vertical components of the residual mean meridional circulation, and defined respectively
as:

$$\bar{v}^* \equiv \bar{v} - e^{z/H}(e^{-z/H}\overline{v'\theta'}/\bar{\theta}_z)_z, \tag{3a}$$

$$\bar{w}^* \equiv \bar{w} + (acos\phi)^{-1}(cos\phi\overline{v'\theta'}/\bar{\theta}_z)_\phi. \tag{3b}$$

Where $\bar{v}$, $\bar{w}$ and $\bar{\theta}$ are respectively the Eulerian zonal-mean meridional and vertical velocities and the potential temperature,
$\phi$ is the latitude, and $S$ is the net rate of change due to chemistry i.e. $\bar{S} = \bar{P} - \bar{L}$, where $\bar{P}$ and $\bar{L}$ are respectively the zonal-
mean chemical production and loss rates. Overbar quantities represent zonal mean fields, primed quantities the departures from
the zonal mean, and subscripts denote derivatives. Meridional derivatives are evaluated in spherical coordinates and vertical
derivatives with respect to log-pressure altitude $z \equiv -Hlog_e(p/p_s)$, with $p_s = 10^5 Pa$ and $H = 7km$.

Hence, transport is separated into advection due to the residual circulation (first 2 terms on the right-hand side (RHS) of Eq.
(1)) and irrevesible quasi-horizontal isentropic eddy mixing, $e^{z/H}\boldsymbol{\nabla} \cdot \boldsymbol{M}$.

In order to better understand the role of each term in the tracer balance, it is useful to separate the components of the vector $\mathbf{M}$ and rearrange the terms of Eq. (1):

$$\bar{\chi}_t = A_y + M_y + A_z + M_z + (\bar{P} - \bar{L}) + \bar{\epsilon}, \tag{4}$$

where:

$$A_y = -\bar{v}^* \bar{\chi}_y, \tag{5a}$$

$$M_y = e^{z/H} \cos\phi^{-1} (M^{(y)} \cos\phi)_y, \tag{5b}$$

$$A_z = -\bar{w}^* \bar{\chi}_z, \tag{5c}$$

$$M_z = e^{z/H} (M^{(z)})_z, \tag{5d}$$

with $A_y$ representing the meridional residual advection, $M_y$ the horizontal transport due to eddy mixing, $A_z$ the vertical residual advection and $M_z$ the vertical eddy mixing (all expressed in $\mathrm{ppbv\,day}^{-1}$). It is important to note that the total mixing term ($M_y + M_z$) includes not only the effects of irreversible mixing, but also some effects of the advective transport which are not resolved by the residual advection (Andrews et al., 1987; Holton, 2004).

Before any TEM calculation all the input fields are interpolated to constant pressure levels from the hybrid-sigma coefficients, that retain the same vertical resolution as the original vertical grid of each dataset (Table 1). Each derivative is computed using a centered differences method.

In addition to the physical TEM terms (Eq. (1)), it is necessary to include an additional term on the RHS of Eq. (4): the residual term $\epsilon$. It is the difference between the actual rate of change of $\chi$ (LHS of Eq. (4)) and the sum of all the transport and chemical terms of the TEM budget. This non-zero residual has several causes (Abalos et al., 2017). The TEM calculations for WACCM rely on the diagnostic variable $w$, which is not used to advect the tracers, because the model is based on a Finite Volume dynamical core (Lin, 2004). Furthermore in WACCM, an implicit numerical diffusion is added to the transport scheme in order to balance the small-scale noise without altering the large-scale. This numerical diffusion is not included in the TEM budget and is larger in regions with large small-scale features, i.e. regions where gradients are stronger (Conley et al., 2012). All TEM calculations are done using daily mean data, even though WACCM and BASCOE both run with a much smaller time step of 30 minutes. The daily mean fields are interpolated from their native hybrid-sigma levels to constant pressure levels prior to the TEM analysis, leading to numerical errors in the lower stratosphere. The BASCOE datasets have a coarser horizontal resolution than their input reanalyses (especially BRAM2; see Table 1). This affects the accuracy of the vertical and horizontal derivatives, with possible implications for the residual. The possible causes of the residual in the five reanalysis datasets are discussed in more detail in Sect. 3. For WACCM-CCMI, the TEM budget is computed for each realization, allowing the examination of both the ensemble mean (e.g. for seasonal means) or the model envelope (e.g. for line plots). In order to validate our $N_2O$ TEM budget, we reproduced the findings reported in Tweedy et al. (2017, Fig. 7) with WACCM-CCMI in the tropical lower stratosphere, and we noticed similar results (not shown).

In order to interpret the TEM analysis of the N$_2$O budget, we also compute the Eliassen-Palm Flux Divergence (EPFD). The Eliassen-Palm flux is a 2-D vector defined as $\boldsymbol{F} \equiv (F^{(\phi)}, F^{(z)})$ (Andrews et al., 1987), with its meridional and vertical components given respectively by:

$$F^{(\phi)} \equiv e^{-z/H} a\cos\phi(\overline{u}_z \overline{v'\theta'}/\overline{\theta}_z - \overline{v'u'}), \tag{6a}$$

$$F^{(z)} \equiv e^{-z/H} a\cos\phi\{[f - (a\cos\phi)^{-1}(\overline{u}\cos\phi)_\phi]\overline{v'\theta'}/\overline{\theta}_z - \overline{w'u'}\}. \tag{6b}$$

The EPFD reflects the magnitude of the eddy processes, and provides a direct measure of the dynamical forcing of the zonal-mean state by the resolved eddies (Edmon et al., 1980).

The four dynamical reanalyses used in this study provide overall consistent temperature and winds in the stratosphere, but can lead to a different representation of large-scale transport (e.g. Chabrillat et al., 2018) due to the biases in the temperature and wind fields (Kawatani et al., 2016; Tao et al., 2019). Note that the TEM quantities are not directly constrained by observations, especially the upwelling velocity $\overline{w}^*$, that can vary considerably in the dynamical reanalyses, as it is a small residual quantity (Abalos et al., 2015).

In the rest of the paper, we will assume that the BRAM2 product provides the best available approximation of the TEM budget for N$_2$O, at least where the residual is smaller than the vertical advection and horizontal mixing terms. This assumption relies on the combination in BRAM2 of dynamical constraints from ERA-Interim with chemical constraints from MLS (Errera et al., 2019).

Figs. 1 and 2 show the N$_2$O TEM budget terms at 15 hPa for all the datasets for the boreal winter (December-January-February, DJF mean) and summer (June-July-August, JJA mean) respectively. The 15 hPa level (around 30 km altitude) was chosen because large differences can be found between WACCM-CCMI, BRAM2, and the CTM runs at this level, and because the dynamical reanalyses are not constrained as well by meteorological observations at higher levels (Manney et al., 2003). Figs. 1 and 2 aim to show how the dynamical and chemical terms of the budget balance each other to recover the tendency $\overline{\chi}_t$ at different latitudes. The discussion about the differences between the datasets, and their possible physical causes, are addressed in the next Sections.

The vertical advection term $A_z$ shows how the upwelling contributes to increasing the N$_2$O abundances in the tropics and summertime mid-latitudes, and how polar downwelling contributes to decreasing the N$_2$O abundances in the winter hemisphere. The horizontal transport out of the tropics due to eddies, as represented by $M_y$, reduces the N$_2$O abundance in the tropical latitudes of the wintertime hemisphere, and increases the N$_2$O mixing ratio at high latitudes in the winter hemisphere. The other terms of the TEM budget are weaker than $A_z$ and $M_y$: the meridional advection term $A_y$ tends to increase the N$_2$O abundance in the winter subtropics and extratropics, while the vertical transport term due to eddy mixing, $M_z$, decreases the N$_2$O mixing ratio over the northern polar latitudes, and the chemistry term $P - L$ shows that N$_2$O destruction by photodissociation and O($^1$D) oxidation contributes to the budget in the tropics and also in the summertime hemisphere. All budget terms are weaker in the summer hemisphere than the winter hemisphere. Over the southern polar winter latitudes, the reanalyses deliver negative $M_y$ that are balanced by large positive residuals, which implies a less robust TEM balance (Fig. 2). This is

not the case with WACCM, where $M_y$ tends to increase the $N_2O$ abundance in the polar vortex. Such differences between the datasets are highlighted and discussed in the next sections.

## 3  Latitude pressure cross sections

Figures 3 and 4 show respectively the DJF and JJA means of three contributions to the $N_2O$ TEM budget, namely horizontal mixing $M_y$, vertical advection $A_z$ and residual terms $\epsilon$, for WACCM-CCMI, JRA55, MERRA2 and BRAM2. For those datasets, the remaining terms of the TEM budget ($A_y$, $M_z$ and $P - L$) for DJF and JJA are shown respectively in supplemental figures S1 and S2. The full $N_2O$ TEM budgets obtained with MERRA and ERAI for DJF and JJA are shown respectively in Figs. S3 and S4. In the case of WACCM-CCMI, the seasonal means were computed separately for each realization and we verified that the ensemble means show the same features as the individual realizations. Large differences arise in the dynamical terms of the TEM budget between summer and winter for both hemispheres in the extratropics. The strong seasonality of the deep branch of the BDC and of the transport barriers are the causes of these differences, as for the seasonal variations of the Age of Air spectum (Li et al., 2012).

We also reproduced the results of Randel et al. (1994, Fig. 8) for the WACCM-CCMI multi-model mean and the reanalysis mean in DJF (Figs S5 and S6 respectively). The WACCM-CCMI and the reanalysis means agree with the Community Climate Model version 2 of the early 1990's with regard to the general pattern of the TEM terms, but both deliver stronger contributions, especially the reanalyses mean.

We first compare the contribution of $A_z$ across the datasets in Figs. 3 and 4. The tropical upwelling increases the abundance of $N_2O$ mostly in the mid-high stratosphere (between 1 and 15 hPa) with the maximum contribution in the summer tropics, while the downwelling decreases it mostly in the wintertime extratropics in the middle and low stratosphere (between 5 and 100 hPa). This reflects the path followed by the deep branch of the BDC (Birner and Bönisch, 2011). During boreal winter, these features are very similar across all datasets (Fig. 3), but noticeable differences appear during the austral winter (Fig. 4): the tropical upwelling has a larger secondary maximum in the southern tropics with JRA55 and MERRA2 than with the other datasets, and the extra-tropical downwelling extends to the South Pole in WACCM-CCMI and JRA55 while it is mostly confined to the mid-latitudinal surf zone in the other reanalyses. In the lower stratosphere, $A_z$ shows the contribution of the residual advection by the shallow branch of the BDC to the $N_2O$ abundances in the winter and summer hemispheres. The two-cell structure, consisting in upwelling of $N_2O$ in the subtropics and downwelling in the extratropics, consistently agrees across all datasets. The meridional residual advection term $A_y$ contributes to the poleward transport of air masses in the middle stratosphere, mostly during the winter, and its contribution to the $N_2O$ TEM budget is weaker than $A_z$. $A_y$ agrees well among the datasets in boreal winter (Figs. S1 and S3), while during austral winter WACCM-CCMI overestimates it around $30°$ S compared to the reanalyses (Figs. S2 and S4).

We move now to the mixing contributions to the $N_2O$ budget. The horizontal mixing is the predominant contribution to the poleward tracer transport in the middle and lower stratosphere (Abalos et al., 2013), as it flattens the tracer gradients generated by the the residual advection. In the $N_2O$ TEM budget during boreal winter, $M_y$ mostly balances the extratropical downwelling

and part of the the tropical upwelling (Figs 3 and S4). The surf zone is characterized by strong horizontal mixing, depicted here as large positive $M_y$ contributions, and delimited by transport barriers which appear as intense gradients of $M_y$ in the winter hemispheres (middle colummns of Figs. 3 and 4). In the wintertime NH, the patterns of $M_y$ are similar in all datasets (Fig. 3), but the effect of horizontal eddy mixing on $N_2O$ is stronger in the reanalyses than in WACCM-CCMI. In Sect. 4 we analyze quantitatively the differences of the mid-stratospheric $M_y$ between datasets. The residual terms in the reanalyses (right column of Fig. 3) are largest in the middle stratosphere at the latitudes of the transport barriers, and their signs are opposite to $M_y$.

In the austral winter, over the Antarctic Polar cap and below 30 hPa, $M_y$ agrees remarkably well in all datasets (Fig. 4). Closer to the vortex edge and above 30 hPa, the wintertime decrease of $N_2O$ is mainly due to downwelling in WACCM-CCMI, while the reanalyses, especially BRAM2, show that the horizontal mixing plays a major role (Fig. 4). The impact of horizontal mixing on $N_2O$ inside the wintertime polar vortex is not negligible (e.g. de la Cámara et al., 2013; Abalos et al., 2016a), as Rossby waves breaking occurs there as well as in the surf zone. In constrast with the reanalyses, in WACCM-CCMI the $M_y$ contribution is close to zero in the Antarctic vortex and maximum along the vortex edge (Fig. 4). This disagreement can be related to differences in the zonal wind: it is overestimated in WACCM above 30 km in subpolar latitudes compared to MERRA (Garcia et al., 2017) and the polar jet is not tilted equatorward as in the reanalyses (see black thin lines in Fig. 4, and Fig. 3 of Roscoe et al., 2012). Yet, the differences in $M_y$ and $A_z$ above the Antarctic in winter should be put into perspective with the relatively large residual terms that points to incomplete TEM budgets in the reanalyses (Fig. 4 and S4 right columns). Near the Antarctic polar vortex, the assumptions of the TEM analysis (such as small amplitude waves) are less valid leading to larger errors in the evaluation of the mean transport and eddy fluxes (Miyazaki and Iwasaki, 2005).

Since the relative importance of the residual is considerable above the Antarctic in the reanalyses (Fig. 4), it is necessary to better understand its physical meaning. Dietmüller et al. (2017) applied the TEM continuity equation to the Age of Air in CCM simulations. Computing the "resolved aging by mixing" (i.e. the AoA counterpart of $M_y + M_z$) as the time integral of the local mixing tendency along the residual circulation trajectories, and the "total aging by mixing" as the difference between the mean AoA (mAoA) and the residual circulation transit time, they defined the "aging by mixing on unresolved scales" (i.e. by diffusion) as the difference between the latter and the former. This "aging by diffusion", which can be related by construction to our residual term, arises around $60°$ S from the gradients due to the polar vortex edge. Even though we use a real tracer ($N_2O$), we find a qualitative agreement with this analysis based on AoA: our residual term is larger in regions characterized by strong gradients such as the antarctic vortex edge, and larger with dynamics constrained to a reanalysis than with a free-running CCM (see EMAC results in Fig. 1d by Dietmüller et al., 2017). We thus interpret the residual as the sum of mixing at unresolved scales and numerical errors (Abalos et al., 2017).

In the summertime lower stratosphere, we note a stronger contribution of $M_y$ to the $N_2O$ abundances above the subtropical jets in both hemispheres and for all datasets compared to higher levels in summer (Figs 3 and 4 middle columns). This behavior is consistent with calculations of the effective diffusivity and age spectra (Haynes and Shuckburgh, 2000; Ploeger and Birner, 2016). It is due to transient Rossby waves that cannot travel further up into the stratosphere due to the presence of critical lines, i.e. where the phase velocity of the wave matches the background wind velocity, generally leading to wave breaking (Abalos et al., 2016b). In particular, above the northern tropics during the boreal summer (Figs. 4, S2 and S4), the horizontal mixing

is primarily associated with the Asian monsoon anticyclone, and causes a decrease in $N_2O$ (Konopka et al., 2010; Tweedy et al., 2017). In the lower stratosphere, the contributions from $M_y$ combine with that from $A_z$ in the total impact of the shallow branch of the BDC on $N_2O$ all year round (Diallo et al., 2012).

The vertical mixing contribution $M_z$ is very small during boreal winter, except in the middle and lower stratosphere poleward of $60°$ N, where it tends to balance the $M_y$ contribution (Figs. S1 and S3). In austral winter, there is a strong disagreement between WACCM-CCMI and the reanalyses around $60°$ S between 5 and 15 hPa (Figs. S2). WACCM-CCMI simulates a strong $M_z$ contribution at the polar jet core, that decreases the $N_2O$ abundances and tends to balance $M_y$, while in the reanalyses $M_z$ is weaker and increases $N_2O$ in the higher stratosphere.

In the next section, we focus on a single level in the middle stratosphere to study quantitatively the disagreement between WACCM-CCMI and the reanalyses.

## 4   Climatological seasonal cycles

After investigating the seasonal means of $A_z$ and $M_y$, it is interesting to examine their climatological mean annual cycles in order to study the month-to-month variations over the year and their dependance on the latitude in the middle stratosphere. The cycles are shown for three latitude bands in each hemisphere corresponding to the tropics ($0°$-$20°$), the surf zones ($40°$-$60°$) and the polar regions ($60°$-$80°$). For WACCM-CCMI, we examine the envelope of the three model realizations in order to evaluate the role of the internal variability and its relative importance for each month and latitude band. In the following, we will consider BRAM2 as the reference when comparing $N_2O$ mixing ratios between datasets, because its dynamics and chemistry are both constrained to observational datasets.

### 4.1   Polar regions

The EPFD is often used to quantify the forcing of the wave drag due to resolved (planetary) waves (e.g. Gerber, 2012; Konopka et al., 2015). We first show the monthly mean climatological annual cycles of EPFD averaged between 3 and 50 hPa, and the residual vertical velocity $\bar{w}^*$ at 15 hPa for the polar regions ($60°$-$80°$ S and N, Fig. 5). We arbitrarily average the EPFD between 3 and 50 hPa in order to identify the wave forcing for the deep branch of the BDC (Plumb, 2002; Konopka et al., 2015). However, the qualitative results do not depend on the choice of the lower boundary level. We also show one realization of the earlier version WACCM4 which suffered from a larger cold bias above the Antarctic (see Sect. 2.1). In WACCM-CCMI, the parameterization of gravity waves was adjusted in order to reduce this issue while not significantly changing the dynamics in the NH, that results in an enhanced polar downwelling above the southern polar region (Garcia et al., 2017). Above the Antarctic, the forcing from resolved waves peaks in October in the reanalyses, as a result of the vortex breakup that allows an enhanced wave activity compared to austral winter (Randel and Newman, 1998). The WACCM simulations miss this strong springtime peak, and they are in good agreement with the reanalyses in the rest of the year (Fig. 5(a)). The residual vertical velocity $w^*$ above the Antarctic is shown in Fig. 5(c). This comparison between the WACCM versions was already shown in Garcia et al. (2017, Fig. 10), we repeat it here adding the dynamical reanalyses. In November-December the weaker

downwelling in WACCM-CCMI agrees well with the reanalyses. Throughout the rest of the year WACCM-CCMI simulates a stronger downwelling than all reanalyses (also at lower levels, not shown). This difference raises the question whether the residual vertical velocity is correctly represented in WACCM-CCMI or in the dynamical reanalyses. Above the Arctic, the WACCM simulations underestimate the EPFD contribution during boreal winter compared to the reanalyses (Fig. 5(b)), and the downwelling velocities simulated by WACCM are weaker than the reanalyses in that period, with no significant differences between the WACCM versions (Fig. 5(d)). The differences between WACCM and the reanalyses in EPFD and $w^*$ in the polar regions will help the interpretation of the differences in $A_z$ and $M_y$.

Figure 6 shows the monthly mean climatological annual cycle of the $N_2O$ mixing ratio, $A_z$, $M_y$ for the polar regions (60°-80° S and N) at 15 hPa for all the datasets. First, we investigate the $N_2O$ mixing ratio in the Antarctic region (Fig. 6(a)). During winter, the $N_2O$ abundances are smaller than the rest of the year, because of the suppressed transport from the lower latitudes caused by the onset of the polar barrier. After the vortex breakup, the $N_2O$ increase during spring and early summer is smaller in all the simulations than in BRAM2. In WACCM-CCMI, the modification of the parameterization of gravity waves results also in a shift towards earlier vortex breakup dates in the austral spring compared to WACCM4 (Garcia et al., 2017). The earlier vortex breakup in WACCM-CCMI allows the transport of $N_2O$-rich air from lower latitudes for a longer period compared to WACCM4, resulting in larger and more realistic simulations of the $N_2O$ mixing ratios during austral spring and early summer (Fig. 6(a)).

In the antarctic region, the downwelling decreases $N_2O$ during most of the year ($A_z$ term in Fig. 6(c)). Here, JRA55 and WACCM-CCMI are outliers: both present stronger $A_z$ contributions in fall and winter, especially WACCM-CCMI reaching values three times stronger than BRAM2 in early winter, as a result of the stronger downwelling velocity simulated by WACCM-CCMI in that region. While this strong disagreement is questioned by the large residuals, we note that all the reanalyses confirm it except JRA55. During fall and summer, $A_z$ is stronger in WACCM-CCMI than in WACCM4, as a consequence of the stronger downwelling in WACCM-CCMI resulting from the modification of the gravity waves parameterization.

We now turn to the contribution from $M_y$. In the antarctic region, $M_y$ is very different among the datasets during winter: in BRAM2 it contributes to the $N_2O$ decrease during fall and winter, with the strongest contribution in July, but with the CTM simulations this contribution is two times weaker, while in WACCM-CCMI the horizontal mixing has almost no effect on $N_2O$ (Fig. 6(e)). As already mentioned, the TEM analysis suffers from large residuals in the wintertime antarctic region. Yet, we note that the disagreement between WACCM-CCMI and BRAM2 is significant, because in fall and winter the envelope of WACCM-CCMI realizations falls completely outside of the possible BRAM2 values when accounting for the residual. During the austral spring, the vortex breakup leads to an increased wave activity reaching the Antarctic, and $M_y$ is in better agreement among all datasets compared to austral winter. Note that WACCM-CCMI exhibits large internal variability in this season (Fig. 6(e)).

It is interesting to highlight the differences between the wintertime Arctic and Antarctic regions, because the hemispheric differences in wave activity (Scaife and James, 2000; Kidston et al., 2015) play a crucial role in the $N_2O$ abundances and TEM budget. Above the Arctic, the $N_2O$ abundances simulated by WACCM agree with the BRAM2 reanalysis, except in December and January, and the CTM experiments driven by MERRA and MERRA2 deliver smaller $N_2O$ mixing ratios compared to

BRAM2 (Figs. 6(b)). $A_z$ is also in good agreement between the datasets above the Arctic, with the exception of ERAI and JRA55 that provide stronger contributions (Fig. 6(d)). The Arctic is characterized by a very variable polar vortex with a shorter life span than the antarctic vortex (Randel and Newman, 1998; Waugh and Randel, 1999), resulting in an enhanced contribution of the horizontal mixing to the $N_2O$ budget during winter compared to the Antarctic (Fig. 6(f)). Compared to the dynamical

reanalyses and BRAM2, WACCM shows in the Arctic a 2-fold underestimation of the $N_2O$ changes due to horizontal mixing during winter. Note that the Arctic extended winter presents the largest internal variability compared to the other regions, as shown by the spread in the WACCM realizations. The weaker contribution from $M_y$ in WACCM is meaningful because the relative importance of the residual term is small in the NH. The horizontal mixing is predominately influenced by the forcing from breaking of resolved (planetary) waves (Plumb, 2002; Dietmüller et al., 2018). In the Arctic region, WACCM

underestimates the forcing from resolved waves compared to the dynamical reanalyses in the middle stratosphere (see Fig. 5(d)). This discrepancy in the resolved wave driving could contribute to the large differences in the wintertime $M_y$ between the CCM simulations and the CTM experiments above the Arctic. On the other hand, the role of different waves driving on mixing processes is an ongoing research topic, and additional data and sensitivity tests are needed in order to establish a clear separation of the waves contribution (e.g. gravity waves parameterization, spatial resolution, etc., Dietmüller et al., 2018). It

should also be emphasized that WACCM is among the CCMI models with the lowest contribution of aging by mixing to Age of Air (Fig. 2 in Dietmüller et al., 2018).

## 4.2 Middle latitudes

Figure 7 shows the monthly mean climatological annual cycle of $w^*$ at 15 hPa and EPFD averaged between 3 and 50 hPa over the surf zones (40°-60° S and N), and Fig. 8 shows the monthly mean climatological annual cycle of the $N_2O$ mixing ratio, $A_z$

and $M_y$ at 15 hPa averaged over the same latitudes. The subtropical barriers are not shown because $M_y$ and $A_z$ change sign in these regions, and averaging across them would hinder the interpretation of their means.

In the southern mid-latitudes, the EPFD peaks in austral spring in the reanalyses, because of the enhanced wave activity in the Southern Hemisphere (SH) during austral spring compared to winter (Konopka et al., 2015), while the WACCM simulations deliver an earlier and weaker peak during autral winter (Fig. 7(a)). The downwelling velocity $w^*$ shows a similar pattern

as the EPFD (Fig. 7(c)), as it is also driven by the breaking of resolved waves (Abalos et al., 2015). In the northern mid-latitudes, the EPFD peaks in winter in all the datasets, reflecting the stronger wave forcing in the surf zone in this season, and WACCM simulates lower EPFD values compared to the reanalyses (Fig. 7(b)) that leads to a weaker downwelling velocity in the WACCM simulations (Fig. 7(d)). As for the polar regions, the differences in EPFD and $w^*$ between the WACCM simulations and the reanalyses will help intepreting of the differences in $A_z$ and $M_y$.

With regard to the $N_2O$ mixing ratio in both hemispheres, the CTM driven by JRA55 and ERAI are in good agreement with BRAM2, while MERRA2 and MERRA underestimate it (Fig. 8(a) and 8(b)). The WACCM-CCMI simulations agree well with the chemical reanalysis BRAM2, confirming the results obtained through the direct comparison with MLS observations (Froidevaux et al., 2019).

We now investigate the contribution from $A_z$ and $M_y$. In the southern mid-latitudes, $A_z$ is negative in all seasons except during summer and there is again a good agreement among the datasets except for WACCM-CCMI and JRA55 (Fig. 8(c)). These two datasets appear to have a purely annual cycle in this region, while the other four show a semi-annual component. The peak in the $A_z$ contribution in the reanalyses in September results from the increased forcing from the resolved waves (see Fig. 7(a)) and from the stronger contribution from gravity waves to the mass flux during spring (Sato and Hirano, 2019, Fig. 11). In the same region, $M_y$ increases throughout the winter, reflecting the mixing associated to the surf zone, and also peaks in early spring in the reanalyses (Fig. 8(e)). During summer and early fall, $M_y$ does not contribute significantly to the TEM budget, and in November $M_y$ reaches negative values which are comparable to the residual term. Both $A_z$ and $M_y$ peak in mid-winter in the WACCM-CCMI simulations, while in the reanalyses these maxima are reached three months later. This difference is related to the earlier minimum in the downwelling velocity $\bar{w}^*$ simulated by WACCM-CCMI (see Fig. 7(c)), that directly affects $A_z$ (Fig. 8(c)) and, by compensation, $M_y$ (Fig. 8(e)). Among the reanalyses, the compensating contributions of $A_z$ and $M_y$ are stronger for JRA55 than for the other reanalyses (up to twice larger in September, see Fig. 8(c) and 8(e)). This reflects the strong BDC in JRA55 that resulted in the youngest mean AoA in the whole stratosphere (Chabrillat et al., 2018).

In the northern middle latitudes, $A_z$ shows the effect of the wintertime downwelling to lower levels on $N_2O$, with the WACCM experiments simulating a sligthly weaker contribution than the reanalyses (Fig. 8(d)). Such disagreement mostly originates from the weaker downwelling velocity in the CCM compared to the reanalyses showed in Fig. 7(b). In the northern mid-latitudes, the strong $M_y$ contribution tends to increase the $N_2O$ abundances in the surf zone during winter (Fig. 8(f)). The reanalyses show a large spread, with values reaching $\sim 1.5 \, \mathrm{ppbv \, day^{-1}}$ in BRAM2 and $\sim 0.9 \, \mathrm{ppbv \, day^{-1}}$ in the MERRA runs, and WACCM-CCMI presents a large underestimation with respect to the reanalyses. While the spread across the reanalyses cannot be explained by the forcing from the resolved waves, the weaker $M_y$ contribution simulated by WACCM could be partly attributed to the weaker EPFD in the CCM compared to the reanalyses (see Fig. 7(d)).

## 4.3 Tropics

Figure 9 shows the climatological annual cycle for the $N_2O$ mixing ratio, $A_z$ and $M_y$ for the southern and northern tropics ($0°$-$20°$ S and N) at 15 hPa across all the datasets. The same latitude bands for the cycles of $w^*$ and EPFD are shown in the Supplement (Fig. S7). In the tropical regions, the $N_2O$ mixing ratio in WACCM-CCMI agrees well with the reanalysis of Aura MLS, while the CTM results show large differences in the $N_2O$ abundances depending on the input reanalysis (Figs. 9(a) and 9(b)). In regions where the AoA is less than 4.5 years and $N_2O$ is greater than 150 ppb, i.e. in the tropical regions and lower stratospheric middle latitudes (Strahan et al., 2011), the $N_2O$ mixing ratio is inversely proportional to the mAoA, because faster upwelling (younger air) implies more $N_2O$ transported from lower levels, decreasing its residence time and resulting in a limited chemical destruction (Hall et al., 1999; Galytska et al., 2019). The dynamical reanalyses also produce large differences in mAoA at 15 hPa: MERRA delivers the oldest mAoA and MERRA2, ERAI and JRA55 progressively show younger mAoA (Fig. 4(b) in Chabrillat et al., 2018). Hence, the large discrepancies in $N_2O$ mixing ratio can be explained by the large differences in mAoA, while $M_y$ and $A_z$ contribute to rates of change of $N_2O$.

We continue by investigating the contribution from $A_z$. In both the tropical regions, the upwelling term $A_z$ is positive all year round showing the effect of tropical upwelling, and agrees very well in the reanalyses (Figs. 9(c) and 9(d)), as a result of the good agreement in the tropical upwelling velocity at 15 hPa (Fig. S7 bottom row), and also as depicted by mAoA diagnostics (Fig. 4(d) in Chabrillat et al., 2018). Large inter-hemispheric differences arise in the seasonality of $A_z$ between the tropical regions. The largest values of $A_z$ in the southern tropics are in the boreal late-fall and winter (Fig. 9(c)), while no large seasonal variations can be detected in the annual cycle of the $A_z$ in the northern tropics (Fig 9(d)). This is the result of the more pronounced seasonality of the upwelling velocity in the southern tropics compared to the northern tropics (Fig. S7 bottom row).

We now turn to the contribution from $M_y$. In the southern tropics, $M_y$ causes a decrease of the $N_2O$ abundances from May to October (when $N_2O$ is transported to the middle latitudes), and has a near-zero contribution in the rest of the year, generally common in all the considered datasets (Fig. 9(e)). The BRAM2 uncertainty is smaller than for the polar region and middle latitudes indicating a better performances of the TEM analysis outside the high latitudes. In the northern tropics, $M_y$ is negative from November to April and presents a marked seasonality in the reanalyses that is much weaker in WACCM-CCMI (Fig. 9(f)). With respect to inter-hemispheric differences, WACCM disagrees with the reanalyses: according to WACCM, $M_y$ has a larger impact in the southern tropics than in the northern tropics, but according to the reanalyses $M_y$ has a much larger impact in the northern tropics (Figs. 9(e) and 9(f)). These inter-hemispheric differences in the $M_y$ contributions can be partly attributed to different forcings from the resolved waves between northern and southern tropics. The EPFD presents a stronger seasonality in the northern tropics than in the southern tropics in all the datasets (Fig. S7 top row), that could partly explain the differences in the seasonality of $M_y$ in the reanalyses, but it does not impact the $M_y$ simulated by WACCM.

## 5    Interannual variability of the seasonal cycles

To analyse the inter-annual variability of the annual cycle, we compute for each month the 1-sigma standard deviations of the $N_2O$ mixing ratio, $M_y$ and $A_z$ across the ten simulated years. Figure 10 shows the annual cycles of these standard deviations for each dataset in the polar regions (60°-80° S and N) at 15 hPa for both hemispheres. First, we consider the variabilities of the $N_2O$ mixing ratio. In the Antarctic, during austral winter and early spring the year-to-year change of the $N_2O$ abundances is very small (Fig. 10(a)), because duration, extension and strength of the polar vortex are very stable in a climatological sense, isolating air masses in the vortex from the highly variable mid-latitudes (Waugh and Randel, 1999). The variability of the $N_2O$ mixing ratio increases in October i.e. during the breaking vortex period that is highly variable in time (Strahan et al., 2015). Furthermore, the mid-latitude air masses, which have more variable composition, become free to reach the higher latitudes during this period. In the arctic region, the inter-annual variability of the $N_2O$ mixing ratio is also largest during springtime but this is spread over a longer period, i.e. from February to June (Fig. 10(b)), reflecting the large interannual variability in the duration, extension and zonal asymmetry of the Arctic polar vortex (Waugh and Randel, 1999). In both polar regions, WACCM-CCMI agrees well with BRAM2 while the CTM experiments simulate a smaller variability.

We now look at the interannual variability of $A_z$ and $M_y$ in the polar regions. Above the Antarctic, the inter-annual vari-
abilities of $A_z$ and $M_y$ are maximum during spring (Figs. 10(c) and 10(e)), due to the large inter-annual variability in vortex
breakup dates (Strahan et al., 2015). While the maximum variability of $M_y$ is consistently reached in October in all the re-
analyses, WACCM-CCMI simulates an earlier maximum (September) that does not correspond with the maximum in its mean
values. The lower wintertime variability of both $A_z$ and $M_y$ would increase if a longer period was considered to include the
exceptional Antarctic vortices of 2002 (Newman and Nash, 2005) and 2019 (Yamazaki et al., 2019). Above the Arctic, $M_y$
and $A_z$ are most variable during winter, reflecting the frequent disruptions of the northern polar vortex by sudden stratospheric
warmings (SSWs, Butler et al., 2017). A case study of the effect of a SSW on the $N_2O$ TEM budget showed that $A_z$ and $M_y$
contribute more to this budget during the SSW event than in the seasonal mean (Randel et al., 1994). Thus, the large wintertime
variabilities of $A_z$ and $M_y$ are explained by the occurrence of seven major SSWs detected in the reanalyses for the 2005-2014
period (Butler et al., 2017).

In Fig. 11 we show the inter-annual variabilities of the $N_2O$ mixing ratio, $M_y$ and $A_z$ for each dataset in the surf zones
(40°-60° S and N) at 15 hPa for both hemispheres. Regarding the $N_2O$ mixing ratio, the inter-annual variability in the southern
middle latitudes reaches the lowest values during austral winter. The datasets deliver very diverse values, with WACCM show-
ing the largest variability and JRA55 the lowest across the climatological year (Fig. 11(a)). In the northern mid-latitudes, the
inter-annual variability of the $N_2O$ mixing ratio increases in late winter across all the datasets, as a response to the increased
wintertime variability of the surf zone (Fig. 11(b)). The variability of WACCM-CCMI largely depends on the considered
realization, except in October and November. Strong differences between ensemble members with respect to inter-annual vari-
ability indicate that the considered period is not long enough to explore the inter-annual variability in the northern mid-latitudes,
and that the mean variability from this ensemble (with only three members) would not be representative of the internal variabil-
ity of WACCM. The inter-annual variabilities of $A_z$ and $M_y$ in the southern mid-latitudes are shown in Figs. 11(c) and 11(e)
respectively. As their mean value, $A_z$ and $M_y$ are most variable during austral spring and late summer in the reanalyses, while
WACCM simulates an earlier peak during winter in the inter-annual variabilities of $A_z$ and $M_y$ compared to the reanalyses.
In the northern mid-latitudes, the inter-annual variabilities of $A_z$ and $M_y$ peak in winter, as their mean values, and WACCM
simulates smaller variabilities compared to the reanalyses (Fig. 11(d) and 11(f)).

Figure 12 shows the annual cycles of the standard deviations of the $N_2O$ mixing ratio, $M_y$ and $A_z$ for each dataset in the
tropical regions (0°-20° S and N) at 15 hPa for both hemispheres. The inter-annual variability of the $N_2O$ mixing ratio in both
southern and northern tropics depends considerably on the dataset (Figs. 12(a) and 12(b)). WACCM-CCMI and the BASCOE
reanalysis of Aura MLS show very similar variabilities, especially in the southern tropics. Since the QBO is the major source of
variability in the tropical stratosphere (Baldwin et al., 2001), this confirms an earlier comparison that showed a good agreement
between the WACCM model and MLS observations in the middle stratosphere in terms of the inter-annual variability of $N_2O$
due to the QBO (Park et al., 2017). Among the CTM simulations, ERAI succeeds to deliver $\sigma(\bar{X})$ as large as BRAM2 and
WACCM-CCMI in the southern tropics, but not in the northern tropics.

The inter-annual variability of $A_z$ in both hemispheres can be related to the impact of the QBO on the tropical upwelling
(Flury et al., 2013). Among MERRA, ERAI and JRA55 the fraction of variance in deseasonalized tropical upwelling $\bar{w}^*$ that is

associated with the QBO is the largest with ERAI (Abalos et al., 2015). Our findings support this conclusion since the largest

$\sigma(\bar{A}_z)$ among the reanalyses is again found with ERAI (Figs. 12(c) and 12(d)). However, a detailed analysis of the impact of the QBO on the BDC as illustated here goes beyond the scope of this study. The variability of $M_y$ in the tropical regions is small compared to the extratropical regions (Figs. 12(e) and 12(f)), in agreement with calculations of effective diffusivity that show small variabilities within the tropical pipe (Abalos et al., 2016a). The reanalyses deliver a larger inter-annual variability in the northern tropics during boreal winter, while in the southern tropics the variability of $M_y$ presents a much weaker annual cycle.

WACCM-CCMI does not reproduce this hemispheric asymmetry, with a rather flat profile in both hemispheres and a clear underestimation in the northern tropics, as shown for its mean values. In the tropical regions, both the variabilities of $M_y$ and $A_z$ fail to explain the good agreement in the variability of N$_2$O between WACCM and BRAM2, as well as their disagreement with the dynamical reanalyses, because $M_y$ and $A_z$ directly contribute to the N$_2$O tendency rather than its mixing ratio.

## 6    Summary and Conclusions

We have evaluated the climatological (2005-2014) N$_2$O transport processes in the stratosphere using the tracer continuity equation in the TEM formalism. In particular we emphasized the horizontal mixing and the vertical advection terms ($M_y$ and $A_z$ respectively). The upwelling term $A_z$ reduces the N$_2$O concentrations in the tropics and increases them in the extratropics, while $M_y$ tends to reduce the meridional gradients of N$_2$O and presents large hemispheric differences. Since $M_y$ or $A_z$ contribute to the local rates of change of N$_2$O, this analysis is complementary to time-integrated diagnostics such as mAoA.

The comparison investigates a variety of datasets, from a free-running chemistry-climate model to a reanalysis where dynamics and chemistry are both constrained. The former comprises three realizations of the CCMI REF-C1 experiment by WACCM, and the latter is the chemical reanalysis of Aura MLS driven by ERA-Interim: BRAM2. The intercomparison also includes the BASCOE CTM driven by four dynamical reanalyses: ERAI, JRA55, MERRA and MERRA2 in order to contribute to the S-RIP.

Considering the N$_2$O mixing ratio in the middle stratosphere, all datasets agree in the annual cycle with the large spread in the N$_2$O abundances of the CTM experiments ($\sim 20\%$) reflecting the diversity of mAoA obtained with the same model (Chabrillat et al., 2018). The upwelling term $A_z$ also agrees among the datasets, especially in the NH where WACCM follows closely the reanalyses. The horizontal mixing term $M_y$ in the NH is weaker in WACCM compared to the reanalyses. In the extratropics, this could be attributed to the weaker forcing from the planetary waves in WACCM compared to the reanalyses.

The differences in $M_y$ become striking in the wintertime Antarctic, where the polar vortex has a major role. According to the reanalyses, the horizontal mixing plays an important role in that region, but that is not found by WACCM. However, this large wintertime $M_y$ in the reanalyses is challenged by a nearly as large residual term. It should be noted that the residual term also includes effects from mixing by diffusion. An additional WACCM run with different gravity waves in the SH is used as a sensitivity test. Over the Antarctic, this test has small impact on $M_y$, but significantly modifies $A_z$ in the austral fall and winter

due to the enhanced downwelling, and the N$_2$O mixing ratio during spring as a consequence of a more realistic timing of the vortex breakup.

The inter-annual variability of the mid-stratospheric horizontal mixing term $M_y$ is largest in the polar regions. In the Antarctic it is related to the variability in the vortex breakup dates during spring, while in the Arctic it is related to the highly variable polar vortex in winter. The inter-annual variability of $A_z$ is characterized by a large spread in the mid-stratospheric tropical regions where WACCM-CCMI and JRA55 deliver a smaller contribution than the other reanalyses. This variability reflects the impact of the QBO on the tropical upwelling (Abalos et al., 2015).

The application of the TEM framework to tracer transport with reanalyses suffers from a poor closure of the budget in the polar regions. We chose to analyse these regions nonetheless because the differences in $M_y$ between WACCM and the reanalyses are larger than the residual term, but it remains important to better understand the causes of these large uncertainties. To this end, detailed studies of transport in the polar stratosphere are needed, e.g. comparing the residual circulations with indirect estimates derived from momentum and thermodynamic balances, and evaluating the effective diffusivity in each dataset (Abalos et al., 2015, 2016a).

The next step of this reasearch consists in the analysis of the inter-annual variations of the BDC, including the impact of the QBO and the El-Nino Southern Oscillation. Further extensions of this work would include the addition of new reanalysis products such as ERA5 and an intercomparison of several CCMs as already done for the residual circulation itself (Chrysanthou et al., 2019).

*Data availability.* The 9 monthly climatologies of the $N_2O$ mixing ratios and TEM budget terms are freely available at the BIRA-IASB repository (http://repository.aeronomie.be) under https://doi.org/10.18758/71021057.

*Author contributions.* DM, SC and EM designed the study. YC provided support in installing and running the models. QE provided the chemical reanalysis BRAM2 and helped in its interpretation. MP ran the CTM experiments. DK provided the WACCM-CCMI realizations and helped in the interpretation of the WACCM datasets. DM wrote and ran the software tools to compute the TEM budgets and realized all the figures. DM, MA and SC analyzed the TEM budgets. DM and SC wrote the text. All co-authors contributed to the interpretation of the results and the reviews of the draft manuscripts.

*Competing interests.* The authors declare that thay have no conflict of interest.

*Acknowledgements.* DM and MP are financially supported by the F.R.S. – FNRS (Brussels) through the ACCROSS research project (grant no. PDR.T.0040.16). EM is a research associate with the F.R.S. – FNRS. MA acknowledges funding from the Atracción de Talento Comunidad de Madrid grant 2016-T2/AMB-1405, and the Spanish National project STEADY (CGL2017-83198-R). We thank the reanalysis centers (ECMWF, NASA GSFC and JMA) for providing their support and data products. WACCM is a component of NCAR's CESM, which

is supported by the NSF and the Office of Science of the US Department of Energy. The authors wish to acknowledge the contribution of

Rolando Garcia in the discussion of the paper specifically for the model results of WACCM.

| Dataset name | Reference | Dynamical Reanalysis | Chemical reanalysis of | Model grid | Top level |
|---|---|---|---|---|---|
| WACCM4 | Marsh et al. (2013) | none | none | 2.5°x1.9°, L66 | $5.1 \times 10^{-6}$ hPa |
| WACCM-CCMI | Garcia et al. (2017) | none | none | 2.5°x1.9°, L66 | $5.1 \times 10^{-6}$ hPa |
| ERAI | Chabrillat et al. (2018) | ERA-Interim (Dee et al., 2011) | none | 2.5°x2°, L60 | 0.1 hPa |
| JRA55 | Chabrillat et al. (2018) | JRA-55 (Kobayashi et al., 2015) | none | 2.5°x2°, L60 | 0.1 hPa |
| MERRA | Chabrillat et al. (2018) | MERRA (Rienecker et al., 2011) | none | 2.5°x2°, L72 | 0.01 hPa |
| MERRA2 | Chabrillat et al. (2018) | MERRA2 (Gelaro et al., 2017) | none | 2.5°x2°, L72 | 0.01 hPa |
| BRAM2 | Errera et al. (2019) | ERA-Interim (Dee et al., 2011) | MLS (Livesey et al., 2015) | 3.75°x2.5°, L37 | 0.1 hPa |

**Table 1.** Overview of the datasets used in this study.

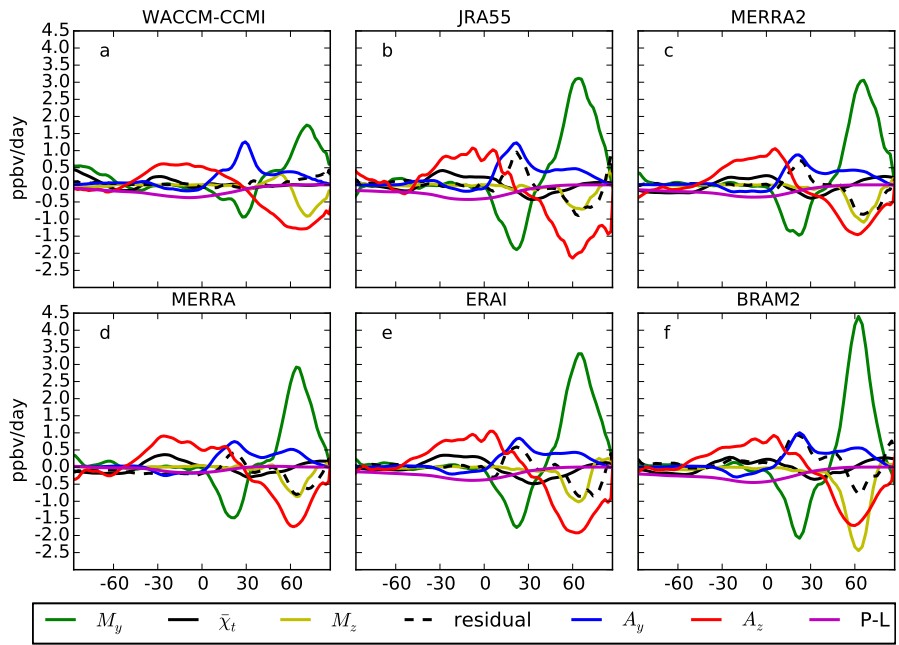

**Figure 1.** Latitudinal profiles of the $N_2O$ TEM budget terms at 15 hPa averaged in DJF (2005-2014). Top row (left to right): WACCM-CCMI (a), JRA55 (b) and MERRA2 (c); bottom row (left to right): MERRA (d), ERAI (e) and BRAM2 (f). The color code is shown in the legend. Units are $ppbv\,day^{-1}$

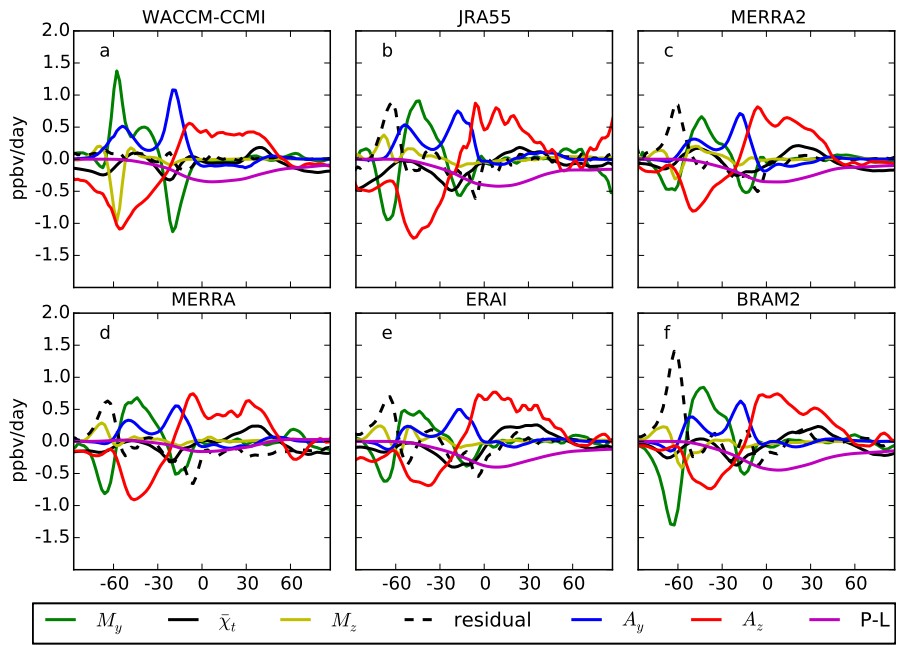

**Figure 2.** Same as previous figure but for JJA.

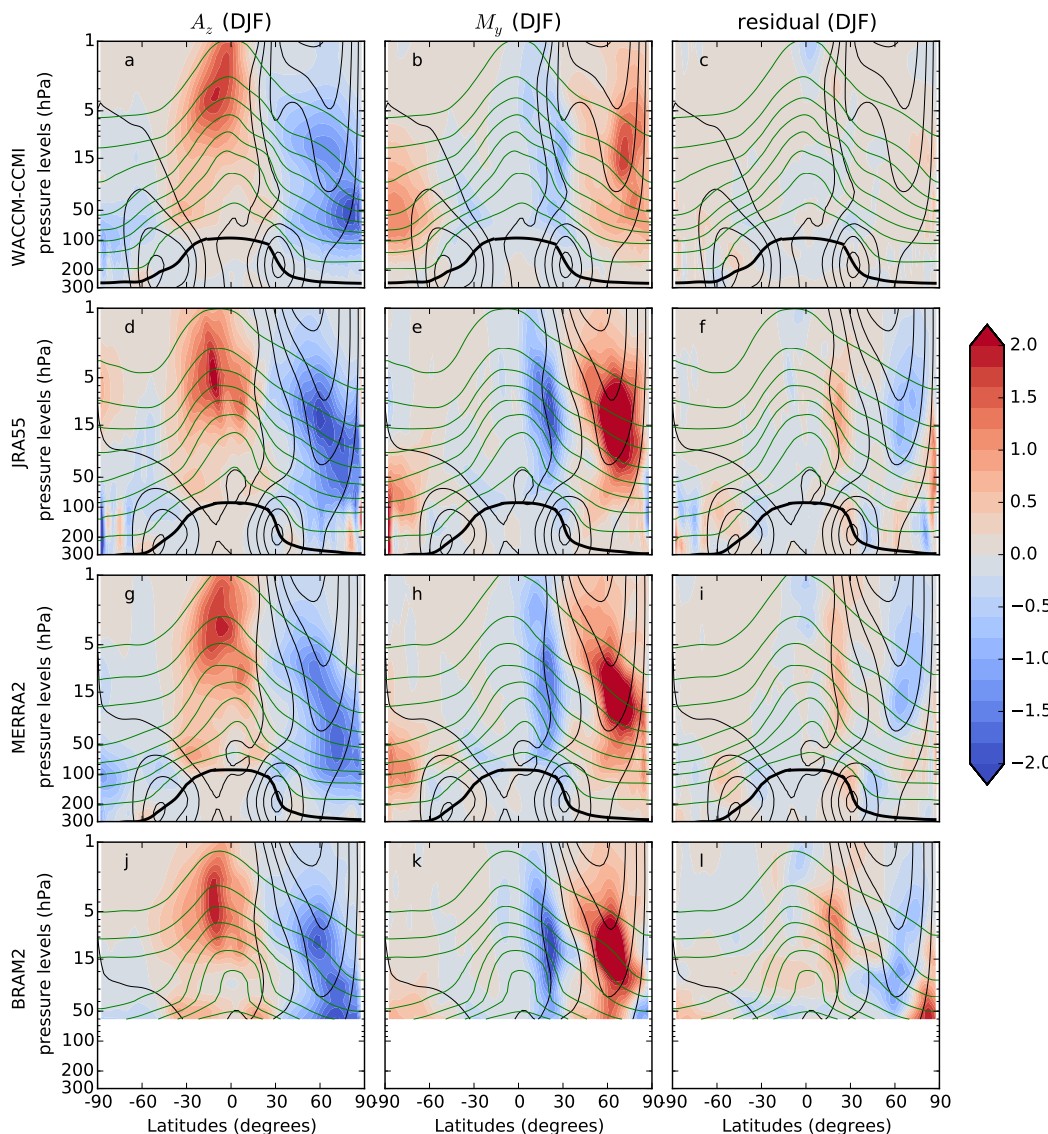

**Figure 3.** Climatological (2005-2014) latitude-pressure cross sections of three $N_2O$ TEM budget terms averaged in DJF (ppbv day$^{-1}$): horizontal mixing term (left column), vertical residual advection term (central column) and residual term (right column). The datasets are, from top to bottom: WACCM-CCMI, JRA55, MERRA2, and BRAM2. The residual term for WACCM-CCMI is from a single realization of the model. The thin black lines show the zonal mean zonal wind (from 0 to 40 m/s every 10 m/s), the black thick line represents the dynamical tropopause for the considered season and the green thin lines show the climatological mixing ratio of $N_2O$ (from 20 to 300 ppbv with 40 ppbv spacing).

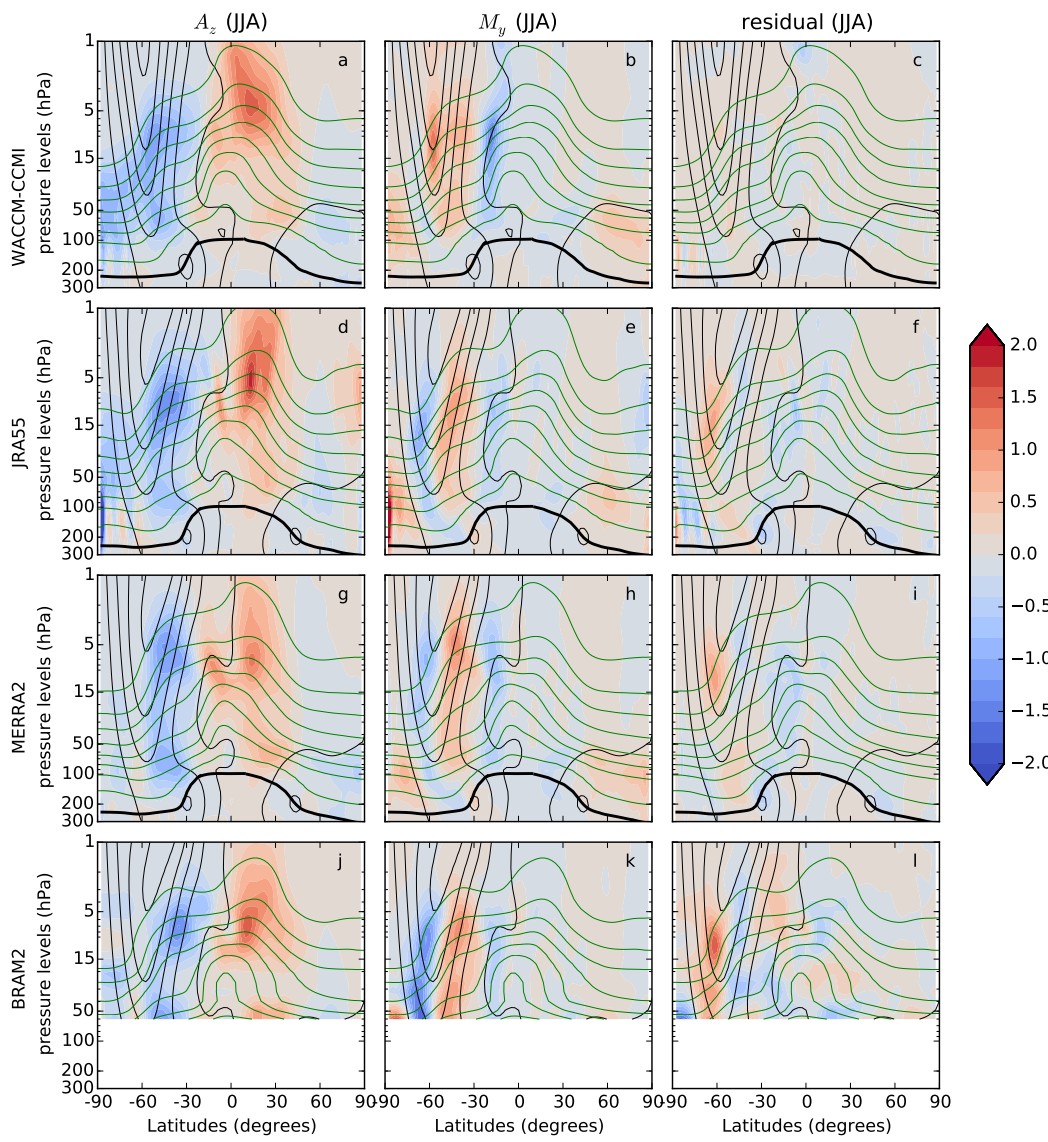

**Figure 4.** Same as previous figure but for JJA and with a different color scale. The thin black contours show the zonal mean zonal wind (from 0 to 100 m/s every 20 m/s).

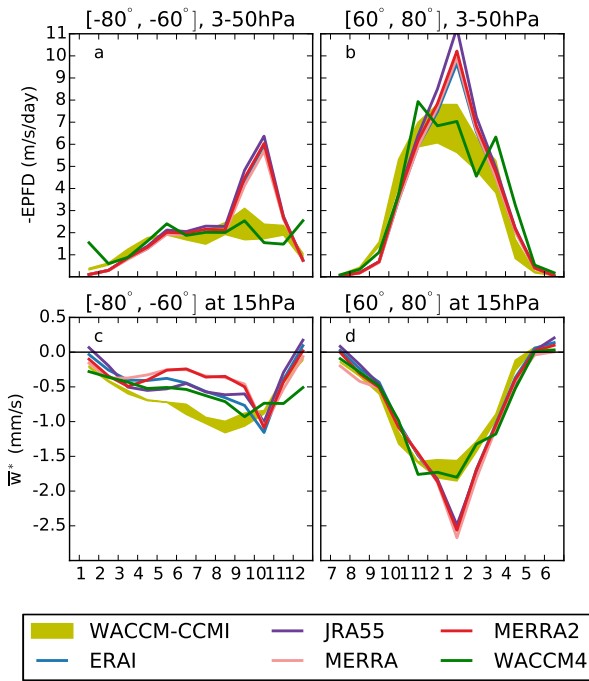

**Figure 5.** Monthly mean annual cycle of $-EPFD$ [$\mathrm{m\,s^{-1}\,day^{-1}}$] averaged between 3 and 50 hPa (upper row), and $\bar{w}^*$ [$\mathrm{mm\,s^{-1}}$] at 15 hPa (bottom row). Left column: Antarctic region (60°-80° S). Right column: Arctic region (60°-80° N). The color code is shown in the legend. The yellow envelope shows the 3 realizations of the WACCM-CCMI simulation.

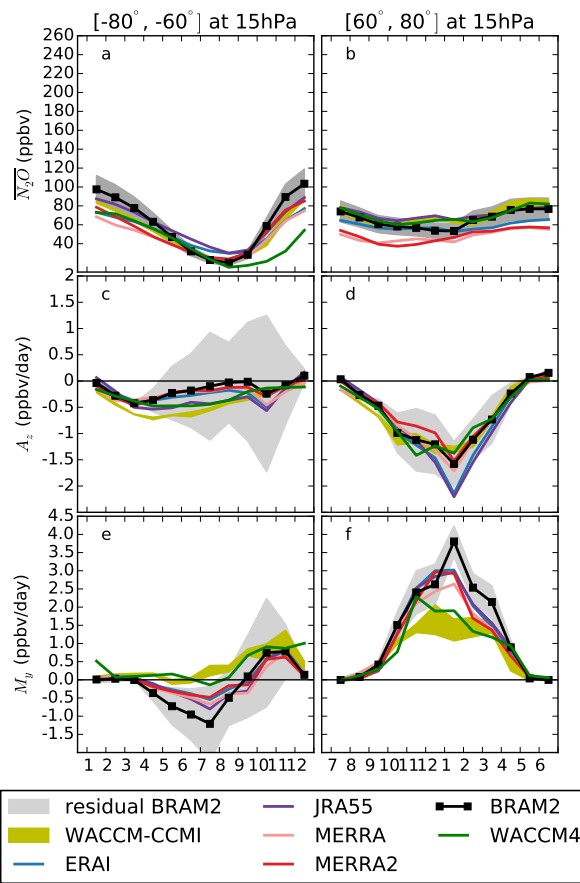

**Figure 6.** Annual cycles over 2005-2014 at 15 hPa. First row: $N_2O$ volume mixing ratio [ppbv], second row: $M_y$ [ppbv day$^{-1}$]; third row: $A_z$ [ppbv day$^{-1}$]. Left column: Antarctic region (60°-80° S). Right column: Arctic region (60°-80° N). The vertical scale differs for $M_y$ and $A_z$. The olive envelope shows the 3 realizations of the WACCM-CCMI simulation. The color codes for the four CTM simulations follow the conventions of S-SRIP (Fujiwara et al., 2017). BRAM2 is depicted with a black line and symbols, as usually done for observations, because it is constrained by both dynamical and chemical observations. Since the $N_2O$ mixing ratio in BRAM2 has been evaluated with a 15% uncertainty (1-sigma standard deviation) at 15 hPa (Errera et al., 2019), this is highlighted by a dark grey region in top rows. The light grey shading around the BRAM2 cycles represents the uncertainty arising from the residual term in the TEM budget, i.e. it is entirely interpreted first as an uncertainty on $A_z$ and then as an uncertainty on $M_y$ in order to remain cautious.

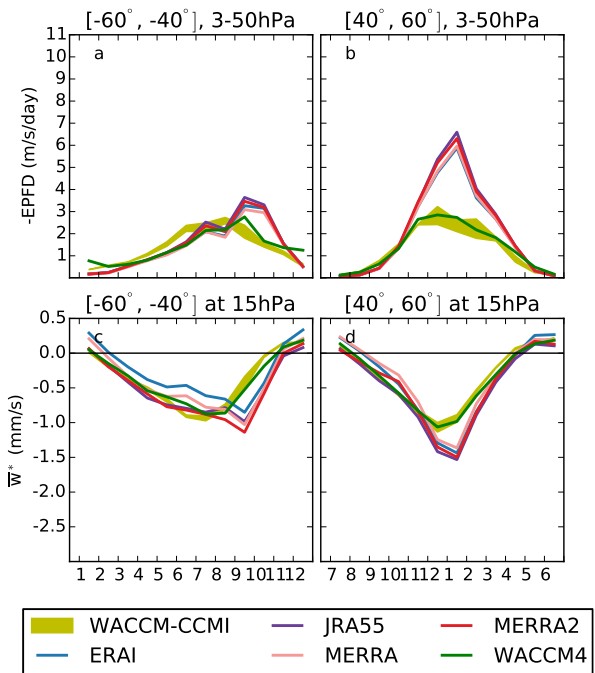

**Figure 7.** As for Fig. 5 but for the middle latitudes. Left column: southern mid-latitudes (40°-60° S). Right column: northern mid-latitudes (40°-60° N).

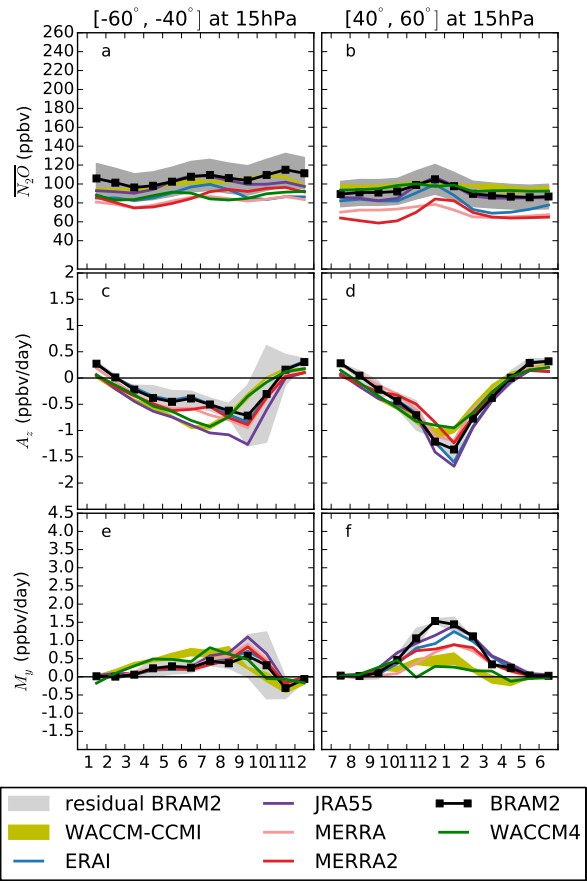

**Figure 8.** As for Fig. 6 but for the middle latitudes. Left column: southern mid-latitudes (40°-60° S). Right column: northern mid-latitudes (40°-60° N).

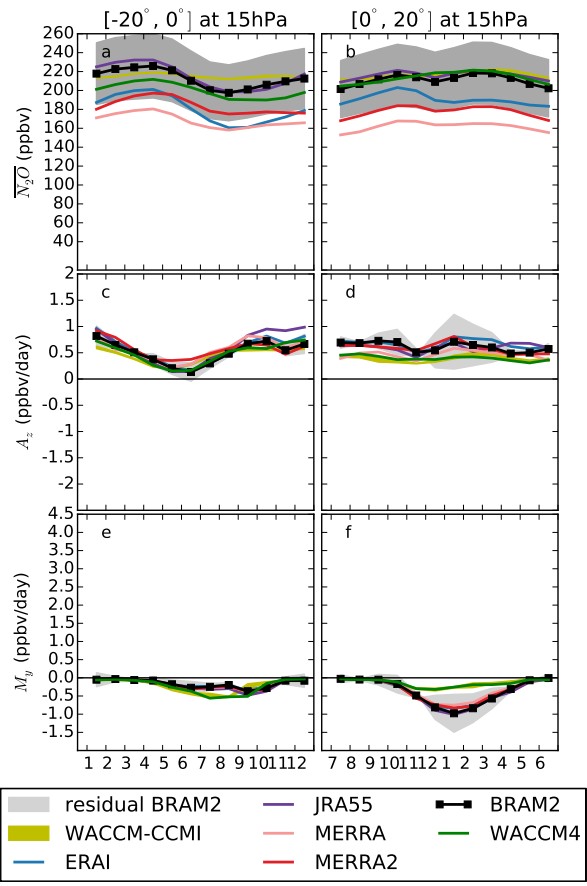

**Figure 9.** As for Fig. 6 but for the Tropics. Left column: southern tropics (0°-20° S). Right column: northern tropics (0°-20° N).

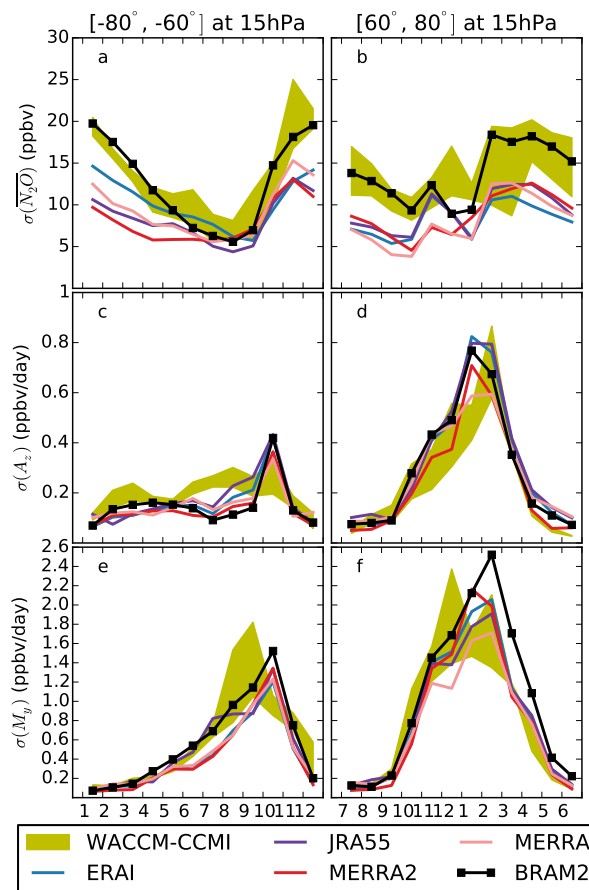

**Figure 10.** Monthly standard deviation over 2005-2014 at 15 hPa. First row: $N_2O$ volume mixing ratio [ppb], second row: horizontal mixing term [ppbv day$^{-1}$]; third row: vertical residual advection term [ppbv day$^{-1}$]. Left column: Antarctic region (60°-80° S), right column: Arctic region (60°-80° N). The color code is shown in the legend. The yellow envelope shows the 3 realizations of the WACCM-CCMI simulation.

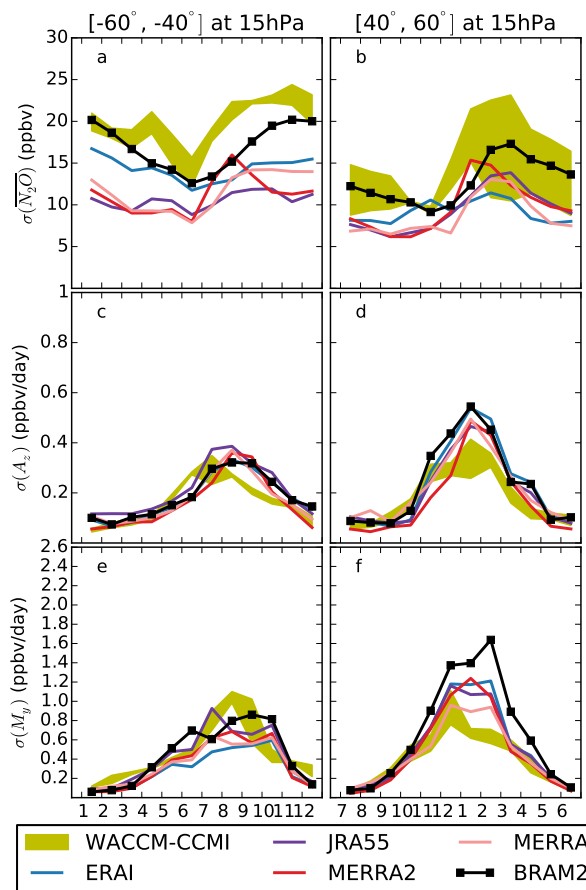

**Figure 11.** As Fig. 10 but the middle latitudes. Left column: southern mid-latitudes (40°-60° S). Right column: northern mid-latitudes (40°-60° N).

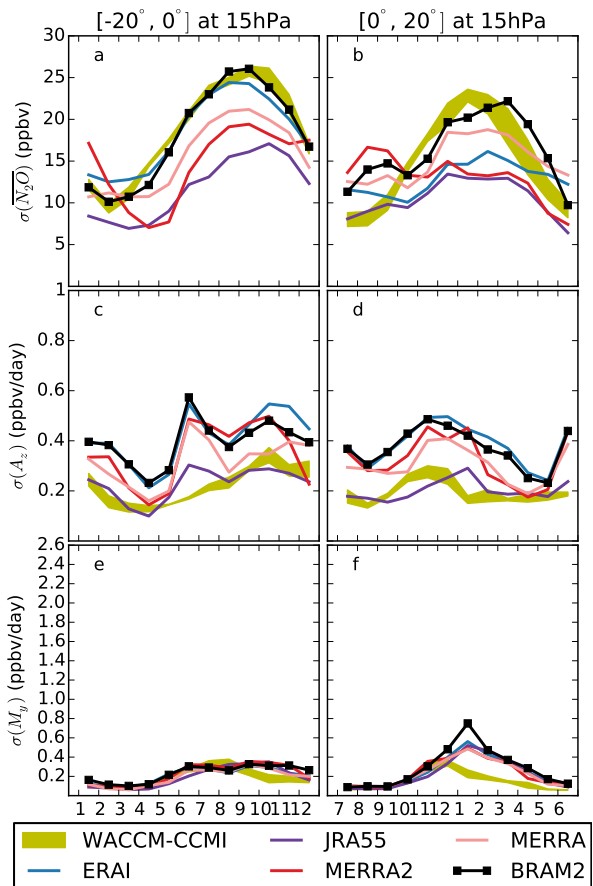

**Figure 12.** As Fig. 10 but the tropics. Left column: southern tropics (0°-20° S). Right column: northern tropics (0°-20° N).

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
