# Peer review of "Climatological impact of the Brewer-Dobson Circulation on the $N_2O$ budget in WACCM, a chemical reanalysis and a CTM driven by four dynamical reanalyses"

_Atmospheric Chemistry and Physics, 2020_

## Referee Comment (RC1) · Mohamadou Diallo (Referee) · 27 May 2020

**Comments on "Climatological impact of the Brewer-Dobson Circulation on the N2O budget in WACCM, a chemical reanalysis and a CTM driven by four dynamical reanalyses" by Daniele Minganti et al.**

The manuscript presents an evaluation of the climatological impact of the stratospheric BDC on the long-lived tracer N2O using the inter-comparison approach of the TEM budget estimated from BRAM2 reanalysis, WACCAM CCM, and BASCOE CTM models. The information is very useful as in the context of a changing climate, any BDC changes will impact UTLS trace gas budget, which might, in turn, radiatively impact surface climate, therefore, it is important understanding the role of the advection and mixing processes in BDC changes, including the vertical residual advection and horizontal mixing. Although the paper contains some interesting material, which should be published, the manuscript itself could be significantly improved qualitatively in some parts (introduction and results). Some paragraphs and sections are poor, therefore, they need to be revised by enhancing the discussion about the scientific content, the structure of results presentations as well as the wording to improve the quality of the paper. Particularly, the differences between WACCM and reanalyses and their possible physical causes could be significantly emphasized. Appropriate references need to be used at the right places instead and properly discussed when necessary. I recommend major revisions. In the following here are my major points and general concerns:

**Major points:**

1. The introduction is poorly written, appropriate references are not properly used at some places, and some sentences are vague (not specific).

2. It is important to show the contribution of the remaining terms such as the vertical mixing and horizontal advection in zonal mean as they are not negligible but just small than the vertical advection and horizontal mixing. This can be added as a supplement information.

3. As the calculation of w* from CCM in CCMI project leads to a bias due to stratospheric shrinking (Eichinger & Shacha, 2020), this make wonder if the w* from WACCM-CCMI calculated consistently with the w* from BASCOE?

4. The scientific discussion of the figure 1 and 2 in the two paragraphs (234-239) is not clear and very poor. Differences/similarities in different terms and in different products are just omitted. All terms contributing to N2O are not well identified and reported.

5. Why is there some differences in the vertical and horizontal mixing and residual terms in the SH between WACCM and reanalyses?

6. So far, ERAi is the reanalysis, which shows a closer pattern changes in the last decade of trace gases closer to observations, including O3, HCl, etc… but it's not shown in figure 3 and 4. A similar panel should be added in the supplement and discussed as well as the horizontal advection and vertical mixing term.

7. The scientific discussion of the figure 3 and 4 related to summer and winter variations of advective and mixing terms is poor and can be improved as well as linked to age spectrum/age of air published articles (Li et al., 2012, Diallo et al, 2012, Ploeger and Birner, 2016).

8. It would be very instructive to reproduce the figure 8 in Randel et al, 1994 which will compare WACCM ensemble mean versus all reanalysis means.

9. The results discussed in "climatological seasonal cycles" section is not clear. It is missing a clear structural organization and not all panels are discussed. Thus, it is very difficult to follow. One suggestion would be to organize the discussion by latitude bins and by term: "In the tropic, …", "In the mid-latitudes, …" and "In the polar region, …"

10. Is there any physical explanation of the spread in the tropical and mid-latitudinal N2O vmr in figure 8? What is the contribution of different QBO representation and modulation of the upwelling to the differences?

11. The results' discussion in section 3.3 are also poor. Need to be improved.

12. The main issue of the paper is results part is poor. The scientific content of the figures are better discussed in the discussion part than in the main part of the paper. This gives to a reader the feeling that he is reading twice the same article. It would be great to put necessary elements in the main part of the manuscript when commenting the figures. This could be done by moving the information in the Discussion session to where it belongs for each figure in the main text.

13. The differences in the tropics, mid-latitude and high latitude need to be discuss clear by taking into account the difference in the QBO. Showing a tropical mean cross-section (5S-5N) of N2O vmr from reanalysis means versus WACCM ensemble means as time series over the dataset period will be great for discussion and for illustration of the possible differences related to QBO (timing, amplitude, phases, …). For insight, please see Park et al. 2017 (fig 9 and 12). In addition for the polar region discussion, it would be very instructive too related the discussion to Randel et al, 1994, where a case study of SSW have been illustrated using N2O budget.

**Minor points:**

1. Page 1, line 1-2, please rephrase the sentence it sounds wrong "*from* the well-mixed tropical *troposphere to* the polar stratosphere" and "…, *chemistry, ozone distribution* and recovery"

2. Page 2, line 33-34, the BDC is the stratospheric circulation and it is not a tropospheric circulation. Please rephrase this sentence "The stratospheric circulation is mainly characterized by the Brewer Dobson Circulation….. from the troposphere…"

3. Page 2, line 38, please replace "The BDC is generated by Rossby waves propagating" by "The BDC is driven by Rossby wave breaking into …"

4. Page 2, line 39, please rephrase "This departure"

5. Page 2, line 41-43, note that the residual circulation can be split into 3 branches: transition, shallow and deep branch for more detail see Lin & Fu (2013). Please improve the discussion by including the relevant previous studies: Haynes et al., 1991, Rosenlof and Holton, 1993; Newman and Nash, 2000; and Birner and Bönich (2011). Please add also the term "breaking" after "synoptic-scale" and "Rossby" and replace

"generate/generated" by "drive/driven" in the whole manuscript. The paragraph (line 38-43) is very poor and need to be improve, and also the natural variability modulations, including QBO and ENSO, of the BDC branches, trace gas transport need to be mentioned see Yang et al, 2014; Baldwin et al 2002, Tweedy et al., 2017, and Diallo et al, 2018,  2019.

6. Page 2, line 50, Please rephrase this sentence "Simulations by Chemistry Climate Model (CCM)…" by "Chemistry Climate Model (CCM) simulations…"

7. Page 2, line 54, the references in the sentence "Observations of long-lived chemical tracers (e.g. H2O, N2O) are often used to derive estimates of the BDC…" is not the appropriate one. Please use the right articles, which examined BDC from H2O, N2O, like e.g. Hegglin et al 2014; Andrews et al. 2001; Kracher et al. 2016; Schoeberl et al, 2008 and H. K. Roscoe, 2006.

8. Page 2-3, line 55-56, the sentence is not correct because the balloon observation trend in the whole NH but only for the deep branch. Please be specific.

9.  Page 3, line 58, please "Stiller et al. 2012" among the early papers using SF6 satellite observation to estimate decadal BDC trends.

10. Page 3, line 59-60, please cite Diallo et al, 2012 and Monge-Sanz et al 2012 among the early papers using reanalysis and observation to assess BDC changes. Add Ploeger et al., 2019 as well.

11. Page 3, line 59-60, the whole sentence "A number..." seems a bit off here as it is break the continuity from the previous session and mixes again reanalysis, climate model & observations while mainly talking about BDC derive from observations and its limitation.

12. Page 3, line 64-65, CLaMS is a Lagrangian transport model driven with reanalyses not a climate model, therefore, the citation of Ploeger et al 2019 is out of place here. Please move it to line 59-60.

13. Page 3, line 66, this "nitrous oxide (N2O)" is already mentioned in page 2, line 53 but online define now.

14. Page 3, line 77, please be specific here by replacing "from several reanalysis datasets." With "from the Chemical ObsErvation (BASCOE) Chemistry-Transport Model (CTM) driven by several reanalysis datasets (Chabrillat et al., 2018)."

15. Page 3, line 77, remove "Dynamical" and replace by "Reanalysis products"

16. Page 3, line 81, move "Fujiwara et al., 2017; Cameron" after "models".

17. Page 3, line 86-88, please citations for each reanalysis product (e.g. Dee et al. 2011, Kobayashi et al 2015, Rienecker et al. 2011, Gelaro et al., 2017).

18. Page 4, line 97-99, the description section 3.1, 3.2 and 3.3 could be combine into section 3 to avoid redundant description.

19.  Page 4, line 102, "Data and methods". There is no "s" to "method".

20. Page 4, line 107-108, please precise what you did "ran" by yourself or "downloaded/use" existing simulations. Rephrase this sentence "We ran one realization of the public version of WACCM (hereafter WACCM4, Marsh et al., 2013), that we downloaded at https://svn-ccsm-models.cgd.ucar.edu/cesm1/release_tags/cesm1_2_2cesm1_2_2."

21. Page 4, line 104, replace "trasport (see Sect. 4)."  by "transport (see Sect. 4 for detailed analysis)". The same remark for "dataset (see Sec. 2.3)".

22. Page 4, line 119, the "… (Lin, 2004)." is not correctly reported in the reference.

23. Page 5, line 124-126, please replace the existence by these ones "In this study, the considered WACCM versions are not able to internally generate the Quasi-Biennial Oscillation (QBO, see e.g. Baldwin et al., 2001). Thus, the QBO is forcing (nudged) by a relaxation of stratospheric winds to observations in the Tropics (Matthes et al., 2010)."

24. Page 5, line 130, add coma after "In addition"

25. Page 5, line 137-138, please rephrase this sentence "The transport module requires on input only the surface pressure and horizontal wind fields from reanalyses, as it relies on mass continuity to derive vertical mass fluxes"

26. Page 5, line 135, please add a comma before "which"

27. Page 5, line 139-141, please add a comma after "but" and "In this way".

28. Page 5, line 147, please rephrase this sentence "For this work the BASCOE CTM provided daily mean outputs over the 2005-2014 period *as for the WACCM experiment*."

29. Page 5, line 150, for analogy to the tow previous model description, this part "The TEM diagnosis is also applied to N2O" is out of place here. First describe the BRAMS2 and then…

30. Page 6, line 164, please remove this "Livesey, in preparation"

31. Page 6, line 170, please the sentence after "temperatures," and start a new one.

32. Page 6 line 180, please add a comma after "Hence"

33. Page 7, line 195, replace "hence retaining" by "while conserving"

34. Page 7, line 201, please a comma before "which"

35. Page 7, line 202, add a comma after "Furthermore in WACCM"

36. Page 7, line 206, replace "timestep" by "time step"

37. Page 7, line 205-207, this sentence can combine to one concise sentence avoid the use of "This". Please rephrase "Finally, the daily mean fields are interpolated from their native hybrid-sigma levels to constant pressure levels prior to the TEM analysis. This could lead to numerical errors in the lower stratosphere."

38. Page 7, line 207, please add a comma after "For WACCM-CCMI"

39. Page 7, line 211, the term "realistic" does not fit well with second part of the sentence "but". What lead to the different representation of large-scale transport is not the fact that the temperature and winds are realistic but because the reanalyses have some differences in wind and temperature. Please see Fig. 5 in Tao et al 2019. You can rephrase the existing sentence as following "The four dynamical reanalyses used in this study provide comparable (consistent) temperature and winds in the stratosphere, but can also lead to a different representation of large-scale transport (e.g. Chabrillat et al., 2018) due to the biases in the temperature and wind fields (Kawatani et al., 2016, Tao et al., 2019). "

40. Page 7, line 213, add a comma after "In the rest of the paper"

41. Page 7, line 214, replace "BASCOE reanalysis BRAM2" by either "BASCOE reanalysis" or "BRAM2 product"

42. Page 8, line 217, add a comma after "In Figs. 1 and 2"

43. Page 8, line 219, replace "the strongest" by "stronger …". In addition DJF & JJA can be term as boreal winter and summer season.

44. Page 8, line 223, regarding the Figure 1, please replace "time der" by "X_t" or "tendency" and redo the figure that the My (green) appear properly in all panels. The fact tendency, residual & horizontal bold line are all in black make different components hard to distinguish. Please fix it.

45. Page 8, line 225, please rephrase "In the northern tropics the N2O decrease due to horizontal mixing is clearly". Also the tendency term of WACCM-CMM is near zero in the NH. I don't see any directional sign therefore the sentence does not match what the panel is showing. Maybe for WACCM panel you can change the vertical scale and note that in the figure caption that the vertical scale of WACCM is different from the reanalyses.

46. Page 8, line 225-226, the interpretation in this sentence is wrong "In the northern tropics ... sufficient to do so." Overall the Ay term in consistent between WACCM and the reanalyses at all latitudes.

47. Page 8, line 226-229, please rephrase this sentence "At the higher latitudes the main terms contributing to the N2O TEM budget are the positive horizontal mixing term in the N2O increase, and the negative vertical advection and vertical mixing terms for the N2O decrease in all the datasets, with negligible contributions from the other terms." It's not clear and poor.

48. Page 8, line 230-231, what about the except of MERRA where the horizontal advection is comparable to Production-lost term as well as the JRA "Ay" increase in the NH. Here also the discussion is poor.

49. Page 8, line 232, this statement is not true for the reanalysis "a general balance between the My and Ay" because for some reanalysis the residual and P-L term are as large as the "My".

50. Page 8, line 233-234, the term "Ay" also contribute in the mid lat.

51. Page 8, line 235, please replace "is affected mostly" by "is mostly affected…"

52. Page 8, line 235-239, Why their differences in the vertical and horizontal mixing and residual terms in the SH between WACCM and reanalyses is not discussed here?

53. Figure 3 and 4, it would be good to add the arrows indicating the residual mean circulation v* and w* as well as the zero zonal mean wind but remove the full zonal men wind fields.

54. Page 9, line 245, add a comma after "CCMI"

55. Page 9 line 250, add a comma after "During the DJF season" and before "but"

56. Page 9 Why the colorbars in figures 3 and 4 have a different scales?

57. Page 9 Why the differences between summer and winter term are not discussed?

58. Page 9, line 259, add a comma after "In the JJA season""

59. Page 9, line 259-267, why the large "My" term from BRAM2 is not mentioned?

60. Page 9, line 262, replace "very positive values" by "large positive values"

61. Page 9-10, regarding the figures 5 and 6, over the whole manuscript you have always discussed NH and then SH. Why then starting with the SH when it comes to figure 5 and 6? It would be good to keep a fix structure.

62. Page 10, line 270-271, the affirmation regarding "My" and "Az" terms showing maxima at 15hPa is wrong because the "Az" terms maximum is around 5 hPa for WACCM-JRA55 and a bit high for the others reanalyses in both seasons DJF & JJA figures. You previous argument was that it's level of better assimilation of meteorological observations according to Manney et al. 2003. Please correct that.

63. Page 9, line 274, add a comma after "For WACCM-CCMI"

64. Page 9, line 275-281, this information should move to the caption. In addition, BRAM2 is a BASCOE reanalysis, while the other reanalysis products (ERAi, JRA55, MERRA) use well-established assimilation system constrained with observations. I don't see why BRAM2 is consider here as the "truth"?

65. Page 9, line 282, replace "We first investigate" by "First, we investigate…"

66. Pages 9-10, line 283-285, Is there any possible physical explanation of ERAi underestimation in tropics? Is there any link to the upwelling or extent of the tropical pipe? Or just a different location of the maximum for ERAi compare to JRA-WACMM?

67. Page 9-10, line 283-287, the discussion is not clear and very hard to follow. Why "the subtropics 40-60" is just not mentioned in the N2O vmr? All panels in the figure have to be discussed, if not please do not show them. It will be clearer and easier to follow if the discussion is done by latitude band e.g. "In the tropic, …", "In the mid-latitudes, …" and "In the polar region, …"

68. Page 10, line 289, replace "We then investigate" by "Second, we investigate…"

69. Page 10, line 322-323, the sentence is not clear and can be split into 2 sentences and formulated clearly.

70. Page 10, line 326, add a comma after "Finally". Same after "In the Tropics from Novermber to April (Fig. 6(g))", same after "In the middle latitudes (Fig. 6(h))", same after "In the arctic region (Fig. 6(i))"

71. In this section 3.2, differences are reported but there is no physically explained attempt.

72. Page 11, line replace "After reporting on the climatological annual cycles, it is desirable to estimate their inter-annual variability. To this end," by " To analyse the inter-annual variability of the annual cycle, we…"

73. Redo panel f) and i) of figure 6 in order to get the quantities shown properly. It is not necessary to keep the same y-axis scaling identical for "Az" and "My" terms.

74. Page 11, line 341, replace "We first consider" by "First, we consider"

75. Page 11, line 342-343, in the $[0°, 20°]$ at 15hPa, BRAMS N2O mixing ratio is more closer to the reanalyses at the first half of the year.

76. Page 11, line 344, add a comma after "In the northern mid-latitudes (Fig.7(d))"

77. Redo panel a) and b) of figure 8.

78. Page 11, line 345-346, why there is no attempt of physical explanation or to link of the spread to differences in upwelling or tropical pipe in the dataset?

79. Page 11, line 347, add a comma after "In the middle latitudes (Figs. 7(e) and 7(h))"

80. Page 11, line 348, add a comma after "In the antarctic region (Fig. 8(c))"

81. Page 11, line 348-350, what is the physical explanation of the hemispheric differences in the Az and My? The strength of the polar? Sudden stratospheric warming?

82. Page 11, line 349, replace "the vortex break-up," by "the breaking vortex period"

83. Page 12, line 350-351, replace "We now move to the variability of the horizontal mixing term My starting from the Tropics (Figs. 7(j) and 7(k)). In the southern tropics (Fig. 7(j))" by "Regarding the variability of the horizontal mixing in the southern tropics (Figs. 7(j, k)), My term shows… In the northern tropics (Fig. 7(k)), My….."

84. Page 12, line 355, add a comma after "In the mid-latitudes"

85. Page 12, line 338, add a comma after "In the antarctic region (Fig. 8(e))".

86. Page 12, line 360 add a comma after "The Arctic (Fig. 8(f)) "

87. Page 12, line 360 add a comma after "Among the reanalyses"

88. Page 12, line 370, please don't oversell the agreement. Replace "excellent agreement" by "fairly good" and complete the sentence "but some differences also occur at …". In addition this part of the sentence "while the CTM delivers overall smaller variabilities." is not true as the reanalysis also show spread in the tropics.

89. Page 12, line 376, add a comma after " Above the Arctic in the middle stratosphere"

90. Page 13, line 408, add a comma after "During the SH spring"

**References:**

1) Lin, P. and Fu, Q.: Changes in various branches of the Brewer–Dobson circulation from an ensemble of chemistry climate models, J. Geophys. Res.-Atmos., 118, 73–84, https://doi.org/10.1029/2012JD018813, 2013

2) Haynes, P. H., McIntyre, M. E., Shepherd, T. G., Marks, C. J., and Shine, K. P.: On the "Downward Control" of Extratropical Diabatic Circulations by Eddy-Induced Mean Zonal Forces, J. Atmos. Sci., 48, 651–678, 1991

3) Bönisch, H. B., Engel, A., Birner, T., Hoor, P., Tarasick, D. W., and Ray, E. A.: On the structural changes in the Brewer-Dobson circulation after 2000, Atmos. Chem. Phys., 11, 3937–3948, https://doi.org/10.5194/acp-11-3937-2011, 2011.

4) Newman, P. A. and Nash, E. R.: Quantifying the wave driving of the stratosphere, J. Geophys. Res.-Atmos., 105, 12485–12497, https://doi.org/10.1029/1999JD901191, 2000.

5) Rosenlof, K. and Holton, J.: Estimates of the stratospheric residual circulation using the downward control principle, J. Geophys. Res., 98, 10465–10479, 1993.

6) Yang, H., Chen, G., and Domeisen, D. I. V.: Sensitivities of the Lower Stratospheric Transport and Mixing to Tropical SST Heating, J. Atmos. Sci., 71, 2674–2694, https://doi.org/10.1175/JAS-D-13-0276.1, 2014.

7) Diallo, M., Konopka, P., Santee, M. L., Müller, R., Tao, M., Walker, K. A., Legras, B., Riese, M., Ern, M., and Ploeger, F.: Structural changes in the shallow and transition branch of the Brewer–Dobson circulation induced by El Niño, Atmos. Chem. Phys., 19, 425–446, https://doi.org/10.5194/acp-19-425-2019, 2019.

8) Diallo, M., Riese, M., Birner, T., Konopka, P., Müller, R., Hegglin, M. I., Santee, M. L., Baldwin, M., Legras, B., and Ploeger, F.: Response of stratospheric water vapor and ozone to the unusual timing of El Niño and the QBO disruption in 2015–2016, Atmos. Chem. Phys., 18, 13055–13073, https://doi.org/10.5194/acp-18-13055-2018, 2018.

9) Andrews, A. E., Boering, K. A., Daube, B. C., Wofsy, S. C., Loewenstein, M., Jost, H., Podolske, J. R., Webster, C. R., Her-man, R. L., Scott, D. C., Flesch, G. J., Moyer, E. J., Elkins, J. W., Dutton, G. S., Hurst, D. F., Moore, F. L., Ray, E. A., Ro-mashkin, P. A., and Strahan, S. E.: Mean age of stratospheric air derived from in situ observations of CO2, CH4 and N2O, J. Geophys. Res., 106, 32295–32314, doi: 10.1029/2001JD000465,2001a

10) Kracher, D., C. H. Reick, E. Manzini , M. G. Schultz, and O. Stein (2016), Climate change reduces warming potential of nitrous oxide by an enhanced Brewer-Dobson circulation, Geophys. Res. Lett., 43, 5851–5859, doi: 10.1002/ 2016GL068390

11) Laura E. Revell et al. GEOPHYSICAL RESEARCH LETTERS, VOL. 39, L15806, doi:10.1029/2012GL052143, 2012

12) Roscoe, 2006: The Brewer–Dobson circulation in the stratosphere and mesosphere – Is there a trend? https://doi.org/10.1016/j.asr.2006.02.078

13) Stiller, G. P., von Clarmann, T., Haenel, F., Funke, B., Glatthor, N., Grabowski, U., Kellmann, S., Kiefer, M., Linden, A., Lossow, S., and López-Puertas, M.: Observed temporal evolution of global mean age of stratospheric air for the 2002 to 2010 period, Atmos. Chem. Phys., 12, 3311–3331, https://doi.org/10.5194/acp-12-3311-2012, 2012.

14) Diallo, M., Legras, B., and Chédin, A.: Age of stratospheric air in the ERA-Interim, Atmos. Chem. Phys., 12, 12133–12154, https://doi.org/10.5194/acp-12-12133-2012, 2012.

15) Monge-Sanz et al. (2012) , QJ, doi: 10.1002/qj.1996

16) Kawatani, Y., Hamilton, K., Miyazaki, K., Fujiwara, M., and Anstey, J. A.: Representation of the tropical stratospheric zonal wind in global atmospheric reanalyses, Atmos. Chem. Phys., 16, 6681–6699, https://doi.org/10.5194/acp-16-6681-2016, 2016.

17) Tao, M., Konopka, P., Ploeger, F., Yan, X., Wright, J. S., Diallo, M., Fueglistaler, S., and Riese, M.: Multitimescale variations in modeled stratospheric water vapor derived from three modern reanalysis products, Atmos. Chem. Phys., 19, 6509–6534, https://doi.org/10.5194/acp-19-6509-2019, 2019.

18) Li, F., Waugh, D. W., Douglass, A. R., Newman, P. A., Pawson, S., Stolarski, R. S., Strahan, S. E., and Nielsen, J. E.: Seasonal variations of stratospheric age spectra in the Goddard Earth Observing System Chemistry Climate Model (GEOSCCM), J. Geophys. Res., 117, D05 134,doi:10.1029/2011JD016877, 2012.

19) Ploeger, F. and Birner, T.: Seasonal and inter-annual variability of lower stratospheric age of air spectra, Atmos. Chem. Phys., 16, 10195–10213, https://doi.org/10.5194/acp-16-10195-2016, 2016.

20) Park, M., Randel, W. J., Kinnison, D. E.,Bourassa, A. E., Degenstein, D. A., Roth,C. Z.,...Santee, M. L. (2017). Variability of stratospheric reactive nitrogen and ozone related to the QBO. Journal ofGeophysical Research: Atmospheres,122,10,103–10,118, https://doi.org/10.1002/2017JD027061

---

## Referee Comment (RC2) · Anonymous Referee #2 · 28 May 2020

In this manuscript the authors compare stratospheric transport processes in a state of the art CCM, in a CTM driven by four different dynamical reanalysis, and a CTM driven by chemical reanalysis. To do so they use the Transformed Eulerian Mean (TEM) framework to analyze the tracer budget of the long-lived tracer N$_2$O . The seasonal patterns, the annual cycle and the variability of the N$_2$O TEM budget (here the focus is on vertical advection and horizontal mixing term) at 15 hPa (higher stratosphere) are analyzed and compared between the different datasets.  Overall the paper is of interest for ACP. However the manuscript should be highly improved in structure and wording! I have the feeling that in some sections the text lacks an organized structure. E.g. when describing the figures, the text jumps from one figure panel to another and it is really hard to follow. I recommend publication after carefully reading over the text again and rephrasing where it is necessaire, and after considering the specific comments below.

**Specific comments/questions:**

- page 1, line 2**:** reword: " … from well-mixed tropical troposphere to polar stratosphere….":  This is a bit too short, here one has the impression, that tracers are transported directly from trop. troposphere to the polar region.

-page 1, line 7:  insert "in "-> .... in a chemical reanalysis

-page 1, line 10:  …. have not been compared **….** *before*.

- page 1, line 14: Please clarify, I do not understand the sentence: "….reflecting the large diversity in mean AoA obtained with the same experiments." The present study does not look at AoA with CTM experiments.

- page 2, line 27: include that you compare interannual variability *between the different datasets*.

- page 2, line 33: reword and clarify this sentence to e.g. "The Brewer Dobson Circulation is characterized by upwelling of tropospheric air to the stratosphere in the tropics, followed by …. " . Note however that the BDC includes both residual circulation (net mass transport) and two-way mixing.  Moreover the  downwelling  takes not only place in the high, but also in the mid-latitudes (change to-> extratropical downwelling) and not only  in wintertime, although in the respective winter hemisphere it is much stronger.

- page 2, line 46:  Why should mixing be limited to a specific latitudinal region of the winter stratosphere?  In the surf zone mixing is only stronger. (see e.g. Fig. 1 in Bönisch et al. 2011)

- page 2, line 51: change to:"… due to the increase in well mixed greenhouse gases (e.g. Butchart et al 2014,…) and due to increased ozone depleting substances (e.g. Polvani et al. 2018 …) "

-page 3, line 56 and line 63: Here the study of Fritch et al. 2019 (https://www.atmos-chem-phys-discuss.net/acp-2019-974) is interesting.

- page 3, line 60: *… observational* trends in the …

- page 3, line 65:  Say why is it important to do this separation?

- page 3, line 72: Could you write more about the study of Tweedy et al. 2017, as they are **also** looking at the N$_2$O TEM continuity equation in GEOSCCM!

- page 3, line 75: In Abalos et al. 2013 the stratospheric N$_2$O buget isn't shown.

- page 3, line 85: change to: …four different dynamical reanalyses are used here to drive simulations ….

- page 3, line 88: Please  clarify: Is only WACCM4  compared to BRAM2?

- page 4, line 93: Are there studies with CTMs driven by reanalyses that studied tracer transport in TEM framework?

- page 4, line 107-118:  You explain the differences of WACCM-4 and WACCM-CCMI  by model development. But are there also differences in the setup of the simulations (e.g. different SSTs, ….)

- page 4, section 2:  I recommend to include a table to give an overview over the different simulations (CCM, CTM with diff. reanalysis).

- page 5, line 132:  WACM-> WACCM

- page 5, line 137:  … as input…

- page 6, line 161: What do you mean with situation of interest?

- page 6, line 182: "N$_2$O balance" -> In this section you use tracer X to explain the TEM diagnostics, but here you change back to N$_2$O. Perhaps you use N$_2$O instead of X in the entire section?

- page 7, line 200:  Can you be a bit clearer, please: You are giving the causes of the non-zero residual for WACCM, but what about the residuals in the CTM, and the chemical reanalysis? Is it only the timestep in BASCOE?

-page 7, line 205: "…while …" -> "…even though …"

- page 7, line 209: Note that Tweedy et al. 2017 looked at N$_2$O TEM buget at 85 hPa in the tropics.

- page 7, line 213: Why does w* vary in reanalyes data? Perhaps you can add one sentence more about Abalos et al. 2015.

-page 8, line 219:  delete "the" -> …. are strongest …

- page 8, line 220:  You motivate the choice of the 15 hPa level with large differences between the CCM and CTM simulations in this region. Where do you see this? I suppose in Figs. 3 +4.  And why isn't it interesting to see what is going on in the lower stratosphere?

 - page 8, line 16: The terms, "vertical advection", "horizontal mixing" and their abbreviations Ay and My are mixed within the manuscript, even between one sentence these terms are mixed (e.g. page 8, line 225).  Can you please use the terms consistently?

-page 8, line 226: "higher latitudes" -> I can see this mainly in the northern higher latitudes.

- page 8, line 232 (and also line 229):"… especially in the reanalyses Az and the residual play a **minor** role": I wouldn't say, that this effect is "minor"!

- page 8 line 238: spelling: r*ea*nalyses

- page 9, line 253:  You only show thee reanalyses  here,  not four.

-page 9, line 266:  middle stratospheric ->middle stratosphere

-page 9, line 257: "(Fig. 3(f), (i), (l))" ->right columns of Fig. 3

-page 9, line 269: Motivate why you are choosing a single level in the middle stratosphere (15 hPa). What about the lower stratosphere?

-page 9-11, description of the climatological seasonal cycles: In my opinion this section is very hard to read, as the SH and NH are separated into two pictures. I recommend to merge Fig.5 and 6 to one Figure and then describe first the tropical, mid-latitude and polar $N_2O$ (upper raw), second the vertical advection Az (middle row) and third horizontal mixing My (bottom raw).  Thus it is easier to see the differences in NH and SH, the text is better structured and you do not have to repeat patterns that are similar.

- page 9, line 278-281: What do you mean with uncertainty – the 1 sigma standard deviation?

- page 9, line 282: " We first investigate the $N_2O$ mixing ratio in the SH. In the tropic (Fic 5c and 6a)…." -> Fig. 6a is not in the SH!

-page 9, line 283: Please point out here more clearly, that BRAM2 is used as reference, and that this is the case for the entire section.

-page 10, line 286:  change to: …is smaller than in BRAMS in all simulations.

-page 10, line 284-288:  You missed to describe the mid-latitudes….

-page 10, line 287:  You wanted to talk about $N_2O$, not about Az and My…

-page 10, line 300: "…expect for JRA55" ->  expect JRA55

-page 10, line 305: "**It** is yet comparable…" -> What? The uncertainty.

-page 10, line 311: Replace differ to different.

-page 11, line 337: Do you use the 1-sigma standard deviation?

-page 11, line 335-340:  I think it is easier for the reader if you plot the standard deviation the same way as in Fig. 5+6. I do not see a real advantage of plotting the results in this order.  And as recommended before it would be nice to have Fig. 7+8 in one plot and restructure the text accordingly.

-page 11, line 343: Why does the variability in WACCM-CCMI strongly depends on the considered realization? Shouldn't the internal variability between these ensemble simulations be similar?

-page 12, subsection "polar regions": The structure of this subsection was not clear to me during reading: you first write about the wintertime North Pole, then about the wintertime South, then you jump to the SH spring and to Antarctic and Arctic inter-annual variability. Perhaps you can give an introducing sentence of what you will discuss in this section.

-page 12, line 375: What do you mean with "Above the Arctic in the middle stratosphere … (Fig.6)"?  Do you refer to the 15 hPa level in Fig. 6?

-page 12, line 376:  I cannot see that $N_2O$ abundance in polar regions (Fig. 6c) are in good agreement in WACCM and BRAMS in the wintertime …

-page 12, line 379: Compared to which reanalysis? To all?  Before you were comparing with BRAMS.

-page 12, line 381: Replace "Fig. 6 bottom raw", to Fig. 6 g+h. And why are you talking about tropics and mid-latitudes here? In this chapter you wanted to discuss the polar regions.

-page 13, line 383: Do you mean the aging by mixing term in the polar regions of Fig. 2 in Dietmüller et al. 2018? Moreover reword "Note that …" This is a poor transition between the two sentences.

-page 13, line 386: Include that TEM AoA buget was done in CCM simulations.

-page 13, line 391: Can you explain, why the TEM formulation is different in this study?

-page 13, line 392: "… agreement: our residual term is larger …" But you are listing the differences here.

-page 13, line 396: Perhaps change to "…SH winter". (Also in other parts of the paper)

-page 13, line 397: Again: What do you mean with "above 30 hPa"? Do you mean the 15 hPa level (latitude band 60-80S), as you are refering to Fig. 5?

-page 13, line 399: You are talking about Fig. 4, not about Fig 5!

-page 13, line 401: Are these studies are giving an explanation for the mixing inside the vortex. If yes, can you please give the explanation here.

-page 13, line 403: Make clear, that it is overestimated in WACCM … (and overestimated according to what?)

-page 13, line 404: Change to : … (see black thin lines in Fig. 4).

-page 13, line 405: You do not show the residual terms in Fig. 5.

-page 13, line 408: Say, why you are now looking at SH spring.

-page 13, line 409:" … better agreement …" Better compared to what?

-page 14, line 418: Replace "reanalyses" with dynamical reanalyses. And why is BRAM2 not included in this comparison?

-page 14, line 434: Please explain critical lines.

-page 14, line 448: vmr -> mixing ratio

**Comments to the Figures:**

- Fig. 1+2: Can you please replace "time der" to $dN_2O/dt$ in the legend.

- You are showing different colorbars in Fig. 3 and 4!

-Fig 5+6, y-axis: Replace X with $N_2O$.

---

## Author Comment (AC1) · 23 Jul 2020

[12pt,a4paper]article hyperref natbib [pdftex]graphicx [textheight=700pt, tex-twidth=480pt]geometry

Response to Reviewer#1 for: Climatological impact of the Brewer-Dobson Circulation on the  $N_2O$  budget in WACCM, a chemical reanalysis and a CTM driven by four dynamical reanalyses

Minganti et al., ACPD, 2020

We thank the reviewer for his in-depth review and useful comments. Following the reviewer's suggestion, we changed the structure of the paper and added supplemental figures. In order to better interpret our results, we inserted new figures showing the Eliassen-Palm Flux Divergence across the datasets. We also strove to improve the text by improving the structure and clarity of the Introduction and scientific discussions and by taking into account the additional references suggested by the reviewer. In our replies below the italic type is used for the reviewer's comments, the plain text for authors' answers and the bold type for the revised text in the manuscript.

**Replies to general comments.**

• Although the paper contains some interesting material, which should be published, the manuscript itself could be significantly improved qualitatively in some parts (Introduction and results). Some paragraphs and sections are poor, therefore, they need to be revised by enhancing the discussion about the scientific content, the structure of results presentations as well as the wording to improve the quality of the paper.

The Introduction was throughly revised and changed. We added new references (see list of references below) and enhanced the discussion by adding the relevant scientific content and removing (or reducing) when necessary. Regarding the results Section, we merged together the Section 3 with Section 4, and the manuscript was restructured in the following layout: Interactive comment

Section 3. Latitude pressure cross sections

Section 4. Climatological seasonal cycles

Section 4.1 Polar regions

Section 4.2 Middle latitudes

Section 4.3 Tropics

Section 5. Interannual variability of the seasonal cycles

Section 6. Summary and Conclusions

This new structure allowed to remove some purely descriptive parts in former Section 3, and add scientific content in former Section 4, i.e. the comparisons with relevant previous studies and physical interpretations of the differences/similarities in the different datasets. The layout of Figs. 5 and 6 changed as well: we separated them by latitude bands (one figure for the polar regions, one figure for the surf zones and one for the tropics) in order to better follow the flow in the Section 4 of the revised manuscript and its subsections. Fig. 9 was also modified according to this new structure, and it is described in the response to the comment below.

• Particularly, the differences between WACCM and reanalyses and their possible physical causes could be significantly emphasized.

The differences (or similarities) between WACCM and the reanalyses are better addressed now, as the pertinent studies comparing WACCM and the reanalyses or Aura MLS observations are considered and discussed. Furthermore, we expanded the Fig. 9 to include the northern and southern middle latitudes and polar regions, while the Tropics were moved in the supplement (Fig. S7). We separated it by latitude bands, i.e. one figure for the polar regions and one figure for the surf zones. We also added an additional row in each figure, showing the divergence of the Eliassen-Palm flux, as a measure of the forcing from resolved
waves for all the considered datasets (BRAM2 is not shown because it uses the dynamical fields from ERAI). WACCM shows an underestimation of the divergence of the Eliassen-Palm flux, that allowed to enhance the discussion about the differences in the mid-stratospheric  $A_z$  and  $M_y$ . The Sect.3.3 was also improved in terms of scientific discussion thanks to the merging with the pertinent parts of Sect. 4, that were expanded and improved accordingly.

**• Appropriate references need to be used at the right places instead and properly discussed when necessary.**

Additional references were added through the revised manuscript. In the Introduction, we added Lin and Fu (2013); Fueglistaler et al. (2009); Birner and Bönisch (2011); Haynes et al. (1991); Rosenlof and Holton (1993); Newman and Nash (2000); Bönisch et al. (2011) for the description of the BDC. For the natural variability of the BDC we added Riese et al. (2012); Yang et al. (2014); Diallo et al. (2019, 2018); Salby and Callaghan (2005). In the part about trend studies of the BDC we added Fritsch et al. (2019). In the reanalyses and CTM description we included Gerber et al. (2010); Rao et al. (2015); Long et al. (2017); Waugh and Hall (2002); Chipperfield (2006); Monge-Sanz et al. (2012); Ménard et al. (2020). For the description of the chemical reanalysis BRAM2 we added Errera et al. (2008); Lahoz and Errera (2010). In Sect. 2 we included the suggested references In Sect. 3 we included Li et al. (2012) for the discussion of the seasonality of the BDC, we added also Roscoe et al. (2012) for the discussion of the differences in  $M_{\mu}$  above the Antarctic, and Ploeger and Birner (2016); Konopka et al. (2010) for the discussion of the lower branch of the BDC. In Sect. 4 we added Konopka et al. (2015); Gerber (2012) in connection to the divergence of the Eliassen-Palm flux, and Sato and Hirano (2019) for the discussion about  $A_z$ in the middle latitudes. In Sect. 5 we added the suggested reference Park et al. (2017) to discuss the inter-annual variability of  $N_2O$ .
**Replies to Major points.**

1. The Introduction is poorly written, appropriate references are not properly used at some places, and some sentences are vague (not specific).

The Introduction was deeply revised according to the comments of both reviewers. BDC. The description of the BDC was improved by describing its different branches, as well as how the wave breaking that leads to the BDC is quantified. The natural variability of the BDC is also discussed. The trend part was de-emphasized, as the current manuscript does not look at BDC trends (which will be the topic of a follow-up study). The Introduction now states that it is important to study the climatological behaviour before trend studies.

CCM and WACCM. The sentences/paragraphs about CCMs and WACCM were put together into one paragraph.

Dynamical reanalyses, CTM and BASCOE. The former Introduction was too vague on these topics, therefore they were re-structured and clarified. The Introduction of dynamical reanalyses has been expanded to mention S-RIP. CTMs driven by reanalyses are described and related to BDC studies of Age of Air. Finally, the BASCOE CTM description was slightly expanded.

All the sentences about Chemical reanalyses and BRAM2 were also merged into one paragraph that starts with a general description of the added value of a chemical reanalysis, continues with the use of chemical reanalyses in TEM studies, and ends with a short description of BRAM2. We conclude the Introduction by summarizing the approach of the paper and providing its structure.

2. It is important to show the contribution of the remaining terms such as the vertical mixing and horizontal advection in zonal mean as they are not negligible but just small than the vertical advection and horizontal mixing. This can be added as a supplement information.
Figures 1 and 2 are not meant for describing the differences/similarities between the datasets, rather for showing how the  $N_2O$  TEM budget and how the different terms balance each other depending on the latitude. The scientific discussion of the differences between datasets, and their possible physical causes, belongs to the following Sections. Yet, this was not stated in the manuscript, and understandably raised some confusion. We now state explicitly the purpose of these figures have shortened their description to focus on the physical meanings of the budget terms as follows:

Figs. 1 and 2 show the  $N_2O$  TEM budget terms at 15 hPa for all the datasets for the boreal winter (December-January-February, DJF mean) and summer (June-July-August, JJA mean) respectively. The 15 hPa level (around 30 km altitude) was chosen because large differences can be found

The contributions of  $A_y$ ,  $M_z$  and P - L for DJF and JJA are shown in the Supplement (Figs. S1 and S2) and appropriately mentioned in Section 3.

3. As the calculation of  $w^*$  from CCM in CCMI project leads to a bias due to stratospheric shrinking (Eichinger & Shacha, 2020), this make wonder if the  $w^*$  from WACCM-CCMI calculated consistently with the  $w^*$  from BASCOE?

The WACCM output we used includes only the basic meteorological variables, i.e. surface pressure, temperature and horizontal and vertical winds fields.  $w^*$  is re-calculated consistently across all datasets through equation 3b using the daily 3-D output of meridional and vertical wind velocity and temperature from WACCM and the dynamical reanalyses. These calculations are performed as recommended by the CCMI project (Chrysanthou et al., 2019).

4. The scientific discussion of the figure 1 and 2 in the two paragraphs (234-239) is not clear and very poor. Differences/similarities in different terms and in different products are just omitted. All terms contributing to N2O are not well identified and reported.

**ACPD**
between WACCM-CCMI, BRAM2, and the CTM runs at this level, and because the dynamical reanalyses are not constrained as well by meteorological observations at higher levels (Manney et al., 2003). Figs. 1 and 2 aim to show how the dynamical and chemical terms of the budget balance each other to recover the tendency  $\bar{\chi}_t$  at different latitudes. The discussion about the differences between the datasets, and their possible physical causes, are addressed in the next Sections.

The vertical advection term  $A_z$  shows how the upwelling contributes to increasing the  $N_2O$  abundances in the tropics and summertime midlatitudes, and how polar downwelling contributes to decreasing the  $N_2O$ abundances in the winter hemisphere. The horizontal transport out of the tropics due to eddies, as represented by  $M_{u}$ , reduces the  $N_2O$  abundance in the tropical latitudes of the wintertime hemisphere, and increases the  $N_2O$ mixing ratio at high latitudes in the winter hemisphere. The other terms of the TEM budget are weaker than  $A_z$  and  $M_y$ : the meridional advection term  $A_{u}$  tends to increase the  $N_{2}O$  abundance in the winter subtropics and extratropics, while the vertical transport term due to eddy mixing,  $M_z$  decreases it over northern polar latitudes and the chemistry term P-L shows that  $N_2O$ destruction by photodissociation and O(1D) oxidation contributes to the budget in the tropics and also in the summertime hemisphere. All budget terms are weaker in the summer hemisphere than the winter hemisphere. Over the southern polar winter latitudes, the reanalyses deliver negative  $M_u$ that are balanced by large positive residuals, which implies a less robust TEM balance (Fig. 2). This is not the case with WACCM, where  $M_{ij}$  tends to increase the  $N_2O$  abundance in the polar vortex. Such differences between the datasets are highlighted and discussed in the next sections.

5. Why is there some differences in the vertical and horizontal mixing and residual terms in the SH between WACCM and reanalyses?
The differences in  $M_y$  between WACCM and the reanalyses above the winter South Pole are discussed in Sects. 3 (fifth and sixth paragraphs) of the revised manuscript:

In the austral winter, over the Antarctic Polar cap and below 30 hPa,  $M_{\mu}$ agrees remarkably well in all datasets (Fig. 4). Closer to the vortex edge and above 30 hPa, the wintertime decrease of  $N_2O$  is mainly due to downwelling in WACCM-CCMI, while the reanalyses, especially BRAM2, show that the horizontal mixing plays a major role (Fig. 4). The impact of horizontal mixing on  $N_2O$  inside the wintertime polar vortex is not negligible (e.g. de la CÃÂamara et al., 2013; Abalos et al., 2016a), as Rossby waves breaking occurs there as well as in the surf zone. In constrast with the reanalyses, in WACCM-CCMI the  $M_u$  contribution is close to zero in the Antarctic vortex and maximum along the vortex edge (Fig. 4). This disagreement can be related to differences in the zonal wind: it is overestimated in WACCM above 30 km in subpolar latitudes compared to MERRA (Garcia et al., 2017) and the polar jet is not tilted equatorward as in the reanalyses (see black thin lines in Fig. 4, and Fig. 3 of Roscoe et al., 2012). Yet, the differences in  $M_{u}$  and  $A_{z}$  above the Antarctic in winter should be put into perspective with the relatively large residual terms that points to incomplete TEM budgets in the reanalyses (Fig. 4 and S4 right columns). Near the antarctic polar vortex, the assumptions of the TEM analysis (such as small amplitude waves) are less valid leading to larger errors in the evaluation of the mean transport and eddy ïÂňÂĆuxes (Miyazaki and Iwasaki, 2005). Since the relative importance of the residual is considerable above the Antarctic in the reanalyses (Fig. 4), it is necessary to better understand its physical meaning. Dietm $\tilde{A}_{1}^{1}$ ller et al. (2017) applied the TEM continuity equation to the Age of Air (AoA) in CCM simulations. Computing the "resolved aging by mixing" (i.e. the AoA counterpart of  $M_u + M_z$ ) as the time integral of the local mixing tendency along the residual circulation trajectories, and the
"total aging by mixing" as the difference between the mean AoA and the residual circulation transit time, they defined the "aging by mixing on unresolved scales" (i.e. by diffusion) as the difference between the latter and the former. This "aging by diffusion", which can be related by construction to our residual term, arises around  $60\hat{a}\hat{A}\hat{U}\hat{A}\hat{e}$  S from the gradients due to the polar vortex edge. Even though we use a real tracer ( $N_2O$ ), we find a qualitative agreement with this analysis based on AoA: our residual term is larger in regions characterized by strong gradients such as the antarctic vortex edge, and larger with dynamics constrained to a reanalysis than with a free-running CCM (see EMAC results in Fig. 1d by Dietm $\tilde{A}\frac{1}{4}$ ller et al., 2017). We thus interpret the residual as the sum of mixing at unresolved scales and numerical errors (Abalos et al., 2017).

They are also discussed in Sect. 4.1 of the revised manuscript :

We now turn to the contribution from  $M_y$ . In the antarctic region,  $M_y$  is very different among the datasets during winter: in BRAM2 it contributes to the  $N_2O$  decrease during fall and winter, with the strongest contribution in July, but with the CTM simulations this contribution is two times weaker, while in WACCM-CCMI the horizontal mixing has almost no effect on  $N_2O$ (Fig. 6(m)). As already mentioned, the TEM analysis suffers from large residuals in the wintertime antarctic region. Yet we note that the disagreement between WACCM-CCMI and BRAM2 is significant, because in fall and winter the envelope of WACCM-CCMI realizations falls completely outside of the possible BRAM2 values when accounting for the residual. During the austral spring, the vortex breakup leads to an increased wave activity reaching the Antarctic (Randel and Newman, 1998), and mid-stratospheric  $M_y$  is in better agreement among all datasets compared to austral winter. Note that WACCM-CCMI exhibits large internal variability in this season (Fig. 6(m)). ACPD
and briefly mentioned in Sect. 2.4:

The BASCOE datasets have a coarser horizontal resolution than their input reanalyses (especially BRAM2; see Table 1). This affects the accuracy of the vertical and horizontal derivatives, with possible implications for the residual.

Again, Figs. 1 and 2 are not meant for the discussion of differences between datasets (this is left to Sects. 3 and 4), but only for showing the TEM budget and pave the way for the following discussion.

6. So far, ERAi is the reanalysis, which shows a closer pattern changes in the last decade of trace gases closer to observations, including O3, HCl, etc... but it's not shown in figure 3 and 4. A similar panel should be added in the supplement and discussed as well as the horizontal advection and vertical mixing term.

The full  $N_2O$  TEM budget obtained with ERAI and MERRA, for DJF and JJA, are now shown in Figs. S3 and S4 of the supplement.

7. The scientific discussion of the figure 3 and 4 related to summer and winter variations of advective and mixing terms is poor and can be improved as well as linked to age spectrum/age of air published articles (Li et al., 2012, Diallo et al, 2012, Ploeger and Birner, 2016).

The summer and winter variations are now addressed through the seasonality of the deep branch of the BDC on the TEM budget (first paragraph of the revised section 3):

Large differences arise in the dynamical terms of the budget between summer and winter for both hemispheres in the extratropics. The strong seasonality of the deep branch of the BDC and of the transport barriers are the causes of these differences, as also shown for the seasonal variations of the Age of Air spectum (Li et al., 2012). ACPD
and through the differences between the shallow and deep branches of the BDC, which are discussed the the third paragraph of revised section 3:.

In the lower stratosphere,  $A_z$  shows the contribution of the residual advection by the shallow branch of the BDC to the  $N_2O$  abundances in the winter and summer hemispheres. The two-cell structure, consisting in upwelling of  $N_2O$  in the subtropics and downwelling in the extratropics, consistently agrees across all datasets.

... and in a new paragraph at the end of section 3:

In the summertime lower stratosphere, we note a stronger contribution of  $M_y$  to the  $N_2O$  abundances above the subtropical jets in both hemispheres and for all datasets compared to higher levels in summer (Figs. 3 and 4 middle columns). This behavior is consistent with calculations of the effective diffusivity and age spectra (Haynes and Shuckburgh, 2000; Ploeger and Birner, 2016). It is due to transient Rossby waves that cannot travel further up into the stratosphere due to the presence of critical lines, i.e. where the phase velocity of the wave matches the background wind velocity, generally leading to wave breaking (Abalos et al., 2016b). In particular, above the northern tropics during the boreal summer (Figs. 4, S2 and S4), the horizontal mixing is primarily associated with the Asian monsoon anticyclone, and causes a decrease in  $N_2O$  (Konopka et al., 2010; Tweedy et al., 2017). In the lower stratosphere, the contributions from  $M_y$ combine with that from  $A_z$  in the total impact of the shallow branch of the BDC on  $N_2O$  all year round (Diallo et al., 2012).

8. It would be very instructive to reproduce the figure 8 in Randel et al, 1994 which will compare WACCM ensemble mean versus all reanalysis means.

We reproduced it for DJF for the WACCM ensemble and the reanalysis ensemble mean, and they are shown in the Supplement (Fig. S5 and Fig. S6 **ACPD**
respectively). This is mentioned in the second paragraph of revised section 3:

We also reproduced the results of Randel et al. (1994, Fig. 8) for the WACCM-CCMI multi-model mean and the reanalysis mean in DJF (Figs S5 and S6 respectively). The WACCM-CCMI and the reanalysis means agree with the Community Climate Model version 2 of the early 1990's with regard to the general pattern of the TEM terms, but both deliver stronger contributions, especially the reanalyses mean.

9. The results discussed in "climatological seasonal cycles" section is not clear. It is missing a clear structural organization and not all panels are discussed. Thus, it is very difficult to follow. One suggestion would be to organize the discussion by latitude bins and by term: "In the tropic, ...", "In the mid-latitudes, ..." and "In the polar region, ..."

We agree with the comment from the reviewer. As stated before, the Section "climatological seasonal cycles" was merged with the Discussion section and divided in three subsections: Polar regions, middle latitudes and Tropics. This allows the structured discussion by latitude bands that the reviewer suggested.

10. Is there any physical explanation of the spread in the tropical and mid-latitudinal N2O vmr in figure 8? What is the contribution of different QBO representation and modulation of the upwelling to the differences?

Regarding the tropical regions, the differences between the datasets are discussed in more detail, and in the revised section 5 we now illustrate the contribution of the QBO on WACCM and BRAM2as follows:

In the Tropics, the inter-annual variability of the  $N_2O$  mixing ratio in both hemispheres depends considerably on the dataset (Figs. 10(b) and (c)). WACCM-CCMI and the BASCOE reanalysis of Aura MLS BRAM2 show very similar variabilities, especially in the southern Tropics. Similar results are found by Park et al. (2017), who showed a good agreement between the ACPD
WACCM model and MLS observations in the middle stratosphere in terms of the inter-annual variability of  $N_2O$  due to the QBO (the major source of variability in the tropical stratosphere, Baldwin et al., 2001). Among the CTM simulations, ERAI succeeds to deliver  $\sigma(\bar{X})$  as large as BRAM2 and WACCM-CCMI in the southern tropics, but not in the northern tropics.

As stated in the last paragraph of the conclusions, a detailed study of the impact of the QBO on  $N_2O$  or the TEM quantities does not belong to this paper, but to a follow-up study that will investigate inter-annual changes.

11. The results' discussion in section 3.3 are also poor. Need to be improved.

The Sect. 3.3 was merged with the relevant parts of Sect. 4, to become Sect. 5 in the revised manuscript. The text is less descriptive and the scientific discussion is improved, using existing and new references.

12. The main issue of the paper is results part is poor. The scientific content of the figures are better discussed in the discussion part than in the main part of the paper. This gives to a reader the feeling that he is reading twice the same article. It would be great to put necessary elements in the main part of the manuscript when commenting the figures. This could be done by moving the information in the Discussion session to where it belongs for each figure in the main text.

Indeed, the reviewer is right. The results part (Sect. 3) was merged with the Discussion (Sect. 4), and new subsections were created (see above). This allows to enhance the scientific discussion and cut the descriptive parts that were not necessary.

13. The differences in the tropics, mid-latitude and high latitude need to be discuss clear by taking into account the difference in the QBO. Showing a tropical mean cross-section (5S-5N) of N2O vmr from reanalysis means versus WACCM ensemble means as time series over the dataset period will be great for discussion

**ACPD**
and for illustration of the possible differences related to QBO (timing, amplitude, phases, ...). For insight, please see Park et al. 2017 (fig 9 and 12). In addition for the polar region discussion, it would be very instructive too related the discussion to Randel et al, 1994, where a case study of SSW have been illustrated using N2O budget.

As announced in the title, the scope of the paper is limited to climatologies. Time series will be investigated in a follow-up study about inter-annual changes. Thus, we decided not to show the suggested time series plot of the reanalysis mean vs WACCM mean. A reference to the work from Park et al. (2017) was added to the third paragraph of revised Section 5:

WACCM-CCMI and the BASCOE reanalysis of Aura MLS show very similar variabilities, especially in the southern Tropics. Since the QBO is the major source of variability in the tropical stratosphere (Baldwin et al., 2001), this confirms an earlier comparison that showed a good agreement between the WACCM model and MLS observations in the middle stratosphere in terms of the inter-annual variability of  $N_2O$  due to the QBO (Park et al., 2017)

as well as a connection to the SSW case study in Randel et al., 1994 for the Arctic (second paragraph of Section 5):

Above the Arctic,  $M_y$  and  $A_z$  are most variable during winter, reflecting the frequent disruptions of the northern polar vortex by sudden stratospheric warmings (SSWs, Butler et al., 2017). A case study of the effect of a SSW on the  $N_2O$  TEM budget showed that  $A_z$  and  $M_y$  contribute more to this budget during the SSW event than in the corresponding seasonal mean. Thus, the large wintertime variability of  $A_z$  and  $M_y$  is explained by the occurrence of seven major SSWs detected in the reanalyses for the 2005-2014 period (Butler et al., 2017). **ACPD**
**Replies to minor points**

1. Page 1, line 1-2, please rephrase the sentence it sounds wrong "from the wellmixed tropical troposphere to the polar stratosphere" and "..., chemistry, ozone distribution and recovery"

The sentence was rephrased:

The Brewer-Dobson Circulation (BDC) is a stratospheric circulation characterized by upwelling of tropospheric air in the Tropics, poleward flow in the stratosphere, and downwelling at mid and high latitudes, with important implications for chemical tracers distribution, stratospheric heat and momentum budgets and mass exchange with the troposphere.

 Page 2, line 33-34, the BDC is the stratospheric circulation and it is not a tropospheric circulation. Please rephrase this sentence "The stratospheric circulation is mainly characterized by the Brewer Dobson Circulation.... from the troposphere..."

The sentence was rephrased:

The Brewer-Dobson Circulation (BDC, Dobson et al., 1929; Brewer, 1949; Dobson, 1956) in the stratosphere is characterized by upwelling of tropospheric air to the stratosphere in the Tropics, followed by poleward transport in the stratosphere and extratropical downwelling.

3. Page 2, line 38, please replace "The BDC is generated by Rossby waves propagating" by "The BDC is driven by Rossby wave breaking into ..."

Done.

4. *Page 2, line 39, please rephrase "This departure"* The part is rephrased as follows: Interactive comment

**...away from its radiative equilibrium. This is balanced by a meridional...**

5. Page 2, line 41-43, note that the residual circulation can be split into 3 branches: transition, shallow and deep branch for more detail see Lin & Fu (2013). Please improve the discussion by including the relevant previous studies: Haynes et al., 1991, Rosenlof and Holton, 1993; Newman and Nash, 2000; and Birner and BÃÂűnich (2011). Please add also the term "breaking" after "synoptic-scale" and "Rossby" and replace "generate/generated" by "drive/driven" in the whole manuscript. The paragraph (line 38-43) is very poor and need to be improve, and also the natural variability modulations, including QBO and ENSO, of the BDC branches, trace gas transport need to be mentioned see Yang et al, 2014; Baldwin et al 2002, Tweedy et al., 2017, and Diallo et al, 2018, 2019.

The discussion has been improved and the suggested references added as follows:

The BDC is driven by tropospheric waves breaking into the stratosphere (Charney and Drazin 1961), which transfer angular momentum and force the stratosphere away from its radiative equilibrium. This is balanced by a meridional (poleward) displacement of air masses, which implies tropical upwelling and extra-tropical downwelling (Holton, 2004). The residual circulation can be further separated in three branches: the transition, the shallow and the deep branch (Lin and Fu, 2013). The transition branch encompasses the upper part of the transition layer between the troposphere and the stratosphere (the tropical tropopause layer, Fueglistaler et al., 2009). The shallow branch is an all year-round lower stratospheric two-cell system driven by breaking of synoptic-scale waves, and the deep branch is driven by Rossby and gravity waves breaking in the middle and high parts of the stratosphere during winter (Plumb, 2002; Birner and Bonisch, 2011). The contributions of different wave types to the driving of the BDC branches has been quantified using the downward control principle, which states

ACPD
that the poleward mass flux across an isentropic surface is controlled by the Rossby or gravity waves breaking above that level (Haynes et al., 1991; Rosenlof and Holton, 1993), and using eddy heat flux calculations as an estimate of the wave activity from the troposphere (e.g., Newman and Nash, 2000).

- Page 2, line 50, Please rephrase this sentence "Simulations by Chemistry Climate Model (CCM)..." by "Chemistry Climate Model (CCM) simulations..." Done.
- Page 2, line 54, the references in the sentence "Observations of long-lived chemical tracers (e.g. H2O, N2O) are often used to derive estimates of the BDC..." is not the appropriate one. Please use the right articles, which examined BDC from H2O, N2O, like e.g. Hegglin et al 2014; Andrews et al. 2001; Kracher et al. 2016; Schoeberl et al, 2008 and H. K. Roscoe, 2006.

As stated in the reply to major point 1, the part of the Introduction dealing with long-term trends was de-emphasized because this manuscript is about the climatology of the BDC, not its trends. In the revised manuscript, studies of BDC trends are introduced with one paragraph citing a few model papers and some observational papers including some of those suggested here by the reviewer.

8. Page 2-3, line 55-56, the sentence is not correct because the balloon observation trend in the whole NH but only for the deep branch. Please be specific.

The sentence was corrected as follows:

... but balloon-borne observations of  $SF_6$  and  $CO_2$  in the Northern Hemisphere (NH) middle latitudes show a non-significant trend of the deep branch of the BDC in the past decades (Engel et al., 2009, 2017).

9. Page 3, line 58, please "Stiller et al. 2012" among the early papers using SF6 satellite observation to estimate decadal BDC trends.
The text was changed and the reference added.

10. Page 3, line 59-60, please cite Diallo et al, 2012 and Monge-Sanz et al 2012 among the early papers using reanalysis and observation to assess BDC changes. Add Ploeger et al., 2019 as well.

As the paragraph about the BDC changes was reduced, this sentence sentence was removed.

11. Page 3, line 59-60, the whole sentence "A number..." seems a bit off here as it is break the continuity from the previous session and mixes again reanalysis, climate model & observations while mainly talking about BDC derive from observations and its limitation.

The reviewer is right, and the sentence was removed from the manuscript.

12. Page 3, line 64-65, CLaMS is a Lagrangian transport model driven with reanalyses not a climate model, therefore, the citation of Ploeger et al 2019 is out of place here. Please move it to line 59-60.

The citation to Ploeger et al., 2019 was moved to the paragraph of the Introduction that explains CTM studies about AoA:

Recent intercomparisons showed that the AoA depends to a large extent on the input reanalysis, both using the kinematic approach (Chabrillat et al., 2018) and the diabatic approach (Ploeger et al., 2019).

13. Page 3, line 66, this "nitrous oxide (N2O)" is already mentioned in page 2, line 53 but online define now.

The first occurrence of the nitrous oxide formula is now at Page 1 line 5:

Since the photochemical losses of nitrous oxide ( $N_2O$ ) are well-known,....
and the "nitrous oxide  $(N_2O)$ " at Page 3 line 75 is replaced by " $N_2O$ ": In this study we use  $N_2O$  as ...

14. Page 3, line 77, please be specific here by replacing "from several reanalysis datasets." With "from the Chemical ObsErvation (BASCOE) Chemistry-Transport Model (CTM) driven by several reanalysis datasets (Chabrillat et al., 2018)."

The paragraph was rearranged, we now mention the BASCOE CTM and the reanalyses used to drive it in a separate paragraph:

Here we use the same CTM as for the kinematic AoA study, i.e. the Belgian Assimilation System of Chemical ObsErvation (BASCOE) CTM. Observations of another long-lived stratospheric tracer, HCFC-22, were recently interpreted with WACCM and BASCOE CTM simulations, showing the interest of this model intercomparison (Prignon et al., 2019). In order to contribute further to the S-RIP BDC activity, four different dynamical reanalyses are used here to drive the BASCOE CTM simulations, compute the  $N_2O$  TEM budget and compare its components with the results derived from WACCM. Namely we consider: the European Centre for Medium-Range Weather Forecasts Interim Reanalysis (ERA-Interim, Dee et al., 2011), the Japanese 55-year Reanalysis (JRA55, Kobayashi et al., 2015), the Modern-Era Retrospective analysis for Research and Applications version 1 (MERRA Rienecker et al., 2011), and version 2 (MERRA2 Gelaro et al., 2017).

- 15. Page 3, line 77, remove "Dynamical" and replace by "Reanalysis products" Done.
- 16. Page 3, line 81, move "Fujiwara et al., 2017; Cameron" after "models".

Done, and the reference to Fujiwara et al., 2017 was removed:

Reanalyses are made using different assimilation methods and forecast models (Cameron et al., 2019), and ....
17. Page 3, line 86-88, please citations for each reanalysis product (e.g. Dee et al. 2011, Kobayashi et al 2015, Rienecker et al. 2011, Gelaro et al., 2017).

The citations were added both in the Introduction (see reply to minor point 14 above) and also in a new Table 1 that provides an overview of all the datasets used in this study.

18. Page 4, line 97-99, the description section 3.1, 3.2 and 3.3 could be combine into section 3 to avoid redundant description.

Thanks to the new structure of the manuscript, the description of the Sections does not include subsections anymore:

In Section 3 we analyse the seasonal mean patterns of the TEM  $N_2O$  budget in each dataset and their differences. Sections 4 and 5 investigate respectively the mean annual cycle and the variability of the  $N_2O$  TEM budget terms, with a focus on the differences between the datasets. Section 6 concludes the study with a summary of our findings and possible future research.

- 19. Page 4, line 102, "Data and methods". There is no "s" to "method". Done.
- 20. Page 4, line 107-108, please precise what you did "ran" by yourself or "downloaded/use" existing simulations. Rephrase this sentence "We ran one realization of the public version of WACCM (hereafter WACCM4, Marsh et al., 2013), that we downloaded at https://svn-ccsm-models.cgd.ucar.edu/cesm1/release\_tags/ cesm1\_2\_2cesm1\_2\_2."

The sentence was rephrased as:

We ran one realization of the public version of WACCM (hereafter WACCM4, Marsh et al., 2013), with a similar setup (e.g. lower boundary conditions) as the CTM experiments; the source code of WACCM4
is available for download at https://svn-ccsm-models.cgd.ucar.edu/cesm1/ release\_tags/cesm1\_2\_2cesm1\_2\_2.

- 21. Page 4, line 104, replace "trasport (see Sect. 4)." by "transport (see Sect. 4 for detailed analysis)". The same remark for "dataset (see Sec. 2.3)".
  Done.
- 22. *Page 4, line 119, the "... (Lin, 2004)." is not correctly reported in the reference.* The reference was corrected.
- 23. Page 5, line 124-126, please replace the existence by these ones "In this study, the considered WACCM versions are not able to internally generate the Quasi-Biennial Oscillation (QBO, see e.g. Baldwin et al., 2001). Thus, the QBO is forcing (nudged) by a relaxation of stratospheric winds to observations in the Tropics (Matthes et al., 2010)."

Done.

- 24. Page 5, line 130, add coma after "In addition" Done.
- 25. Page 5, line 137-138, please rephrase this sentence "The transport module requires on input only the surface pressure and horizontal wind fields from reanalyses, as it relies on mass continuity to derive vertical mass fluxes"

The sentence was rephrased:

Chabrillat et al. (2018) explain in detail the preprocessing procedure that allows the BASCOE CTM to be driven by arbitrary reanalysis datasets, and the set-up of model transport. As usual for kinematic transport modules, the FFSL scheme only needs the surface pressure and horizontal wind fields from reanalyses as input, because it is set on a coarser grid than Interactive comment

the input reanalyses, and relies on mass continuity to derive vertical mass fluxes corresponding to its own grid.

- 26. *Page 5, line 135, please add a comma before "which"* Done.
- 27. Page 5, line 139-141, please add a comma after "but" and "In this way". Done.
- 28. Page 5, line 147, please rephrase this sentence "For this work the BASCOE CTM provided daily mean outputs over the 2005-2014 period as for the WACCM experiment."

The sentence was rephrased:

As for the WACCM experiment, we used the daily mean outputs from the BASCOE CTM over the 2005-2014 period.

29. Page 5, line 150, for analogy to the tow previous model description, this part "The TEM diagnosis is also applied to N2O" is out of place here. First describe the BRAMS2 and then...

We do not mention the TEM  $N_2O$  budget at that stage anymore, and the sentence was rephrased:

BRAM2 is the BASCOE Reanalysis of Aura MLS, version 2, which covers the period....

- 30. *Page 6, line 164, please remove this "Livesey, in preparation"* Done.
- 31. Page 6, line 170, please the sentence after "temperatures," and start a new one.

**ACPD**
There is now a period after the temperatures (definition of  $M^{(z)}$ ), and the new sentence starts with the definition of  $v^*$  and  $w^*$ :

 $M^{(z)}\equiv \dots$  .

 $v^*$  and  $w^*$  are...

- 32. Page 6 line 180, please add a comma after "Hence" Done.
- 33. Page 7, line 195, replace "hence retaining" by "while conserving"

The sentence was rephrased:

Before any TEM calculation all the input fields are interpolated to constant pressure levels from the hybrid-sigma coefficients, that retain the same vertical resolution as the original vertical grid of each dataset (Table 1).

- 34. Page 7, line 201, please a comma before "which" Done.
- 35. Page 7, line 202, add a comma after "Furthermore in WACCM" Done.
- 36. *Page 7, line 206, replace "timestep" by "time step"* Done.
- 37. Page 7, line 205-207, this sentence can combine to one concise sentence avoid the use of "This". Please rephrase "Finally, the daily mean fields are interpolated from their native hybrid-sigma levels to constant pressure levels prior to the TEM analysis. This could lead to numerical errors in the lower stratosphere."

**ACPD**
The sentence was rephrased:

The daily mean fields are interpolated from their native hybrid-sigma levels to constant pressure levels prior to the TEM analysis, leading to numerical errors in the lower stratosphere.

- 38. Page 7, line 207, please add a comma after "For WACCM-CCMI" Done.
- 39. Page 7, line 211, the term "realistic" does not fit well with second part of the sentence "but". What lead to the different representation of large-scale transport is not the fact that the temperature and winds are realistic but because the reanalyses have some differences in wind and temperature. Please see Fig. 5 in Tao et al 2019. You can rephrase the existing sentence as following "The four dynamical reanalyses used in this study provide comparable (consistent) temperature and winds in the stratosphere, but can also lead to a different representation of largescale transport (e.g. Chabrillat et al., 2018) due to the biases in the temperature and wind fields (Kawatani et al., 2016, Tao et al., 2019). "

The sentence was rephrased as suggested.

The four dynamical reanalyses used in this study provide overall consistent temperature and winds in the stratosphere, but can lead to a different representation of large-scale transport (e.g. Chabrillat et al., 2018) due to the biases in the temperature and wind fields (Kawatani et al., 2016; Tao et al., 2019). Note that the TEM quantities are not directly constrained by observations, especially the upwelling velocity  $\bar{w}^*$ , that can vary considerably in the dynamical reanalyses, as it is a small residual quantity (Abalos et al., 2015).

40. *Page 7, line 213, add a comma after "In the rest of the paper"* Done.
41. Page 7, line 214, replace "BASCOE reanalysis BRAM2" by either "BASCOE reanalysis" or "BRAM2 product"

"BASCOE reanalysis BRAM2" was replaced with "BRAM2 product".

42. Page 8, line 217, add a comma after "n Figs. 1 and 2"

The sentence was rephrased:

**Figs. 1 and 2 show the ....**

43. Page 8, line 219, replace "the strongest" by "stronger ...". In addition DJF & JJA can be term as boreal winter and summer season.

The whole sentence was removed from the manuscript.

44. Page 8, line 223, regarding the Figure 1, please replace "time der" by " $X_t$ " or "tendency" and redo the figure that the My (green) appear properly in all panels. The fact tendency, residual & horizontal bold line are all in black make different components hard to distinguish. Please fix it.

"time der" was replaced by " $X_t$ ". The y-scale was widened so  $M_y$  could appear properly in the all panels. The horizontal bold line (i.e. the zero line) was removed.

45. Page 8, line 225, please rephrase "In the northern tropics the N2O decrease due to horizontal mixing is clearly". Also the tendency term of WACCM-CMM is near zero in the NH. I don't see any directional sign therefore the sentence does not match what the panel is showing. Maybe for WACCM panel you can change the vertical scale and note that in the figure caption that the vertical scale of WACCM is different from the reanalyses.

The whole sentence was indeed confusing. The discussion of Figs. 1 and 2 was reduced because it was repetitive and it aims to describe only the most important points.
46. Page 8, line 225-226, the interpretation in this sentence is wrong "In the northern tropics ... sufficient to do so." Overall the Ay term in consistent between WACCM and the reanalyses at all latitudes.

The discussion about Figs. 1 and 2 was changed, see our reply to major point 4.

47. Page 8, line 226-229, please rephrase this sentence "At the higher latitudes the main terms contributing to the N2O TEM budget are the positive horizontal mixing term in the N2O increase, and the negative vertical advection and vertical mixing terms for the N2O decrease in all the datasets, with negligible contributions from the other terms." It's not clear and poor.

The discussion about Figs. 1 and 2 was changed, see comment above.

48. Page 8,line 230-231, what about the except of MERRA where the horizontal advection is comparable to Production-lost term as well as the JRA "Ay" increase in the NH. Here also the discussion is poor.

As mentioned before, Figs. 1 and 2 are not meant to discuss differences in the datasets (this is left to the next Sections), but only to show how the terms of the TEM budget balance each other. The discussion about Figs. 1 and 2 was changed, see comments above.

49. Page 8, line 232, this statement is not true for the reanalysis "a general balance between the My and Ay" because for some reanalysis the residual and P-L term are as large as the "My".

We agree with the reviewer, but, again, we do not wish to compare the datasets at this point of the manuscript. The discussion about Figs. 1 and 2 was changed, see comments above.

50. Page 8, line 233-234, the term "Ay" also contribute in the mid lat.
The discussion about Figs. 1 and 2 was changed, see comments above.

51. Page 8, line 235, please replace "is affected mostly" by "is mostly affected..."

As the paragraph was largely changed, this is not included anymore.

52. Page 8, line 235-239, Why their differences in the vertical and horizontal mixing and residual terms in the SH between WACCM and reanalyses is not discussed here?

As mentioned above, Figs. 1 and 2 are meant only for illustrating the various terms of the TEM budget, and how they balance each other at different latitudes. This is now explicitly stated in the discussion of Figs.1 and 2. The differences between datasets are discussed in detail in Sect. 3,4 and 5 of the revised manuscript.

53. Figure 3 and 4, it would be good to add the arrows indicating the residual mean circulation  $v^*$  and  $w^*$  as well as the zero zonal mean wind but remove the full zonal men wind fields.

We thank the reviewer for the comment, but we chose not to show the residual advection and not remove the full zonal wind because we think that the full zonal mean wind is useful for showing the polar jet, as it is related to the discussion of Fig. 4, and the addition of the arrows of  $v^*$  and  $w^*$  would make the panel rather difficult to interpret.

- 54. Page 9, line 245, add a comma after "CCMI" Done.
- 55. Page 9 line 250, add a comma after "During the DJF season" and before "but" Done, and we replaced "DJF season" by "boreal winter".

**ACPD**
- 56. *Page 9 Why the colorbars in figures 3 and 4 have a different scales?* We now use the same color scale [-2,2] ppbv/day for both the figures.
- 57. *Page 9 Why the differences between summer and winter term are not discussed?* Those differences are discused in the revised manuscript (Sect. 3). See our reply above to major point 7.
- 58. Page 9, line 259, add a comma after "In the JJA season""

Thanks to the new manuscript structure, this paragraph was removed.

59. Page 9, line 259-267, why the large "My" term from BRAM2 is not mentioned?

This is now discussed in the fifth paragraph of section 3:

In the austral winter, over the Antarctic Pole and below 30 hPa,  $M_y$  agrees remarkably well in all datasets (Fig. 4). Closer to the vortex edge and above 30 hPa, the wintertime decrease of  $N_2O$  in the middle stratosphere is mainly due to downwelling in WACCM-CCMI, while the reanalyses, especially BRAM2, show that the horizontal mixing also plays a major role (Fig. 4).

- 60. *Page 9, line 262, replace "very positive values" by "large positive values"* With the new manuscript structure, this paragraph was removed.
- 61. Page 9-10, regarding the figures 5 and 6, over the whole manuscript you have always discussed NH and then SH. Why then starting with the SH when it comes to figure 5 and 6? It would be good to keep a fix structure.

In the revised manuscript, the discussion of the Figs. 5 and 6 (now merged into Fig. 5) is separated in subsections organized by latitude band (Polar region, middle latitudes, Tropics), rather than by hemisphere. This allows to better describe similarities/differences between the hemispheres, and to avoid repetition whenever the patterns are similar.

ACPD
62. Page 10, line 270-271, the affirmation regarding "My" and "Az" terms showing maxima at 15hPa is wrong because the "Az" terms maximum is around 5 hPa for WACCM-JRA55 and a bit high for the others reanalyses in both seasons DJF & JJA figures. You previous argument was that it's level of better assimilation of meteorological observations according to Manney et al. 2003. Please correct that.

The sentence was removed as the same statement was already in Sect. 2.4.

63. Page 9, line 274, add a comma after "For WACCM-CCMI"

Done.

64. Page 9, line 275-281, this information should move to the caption. In addition, BRAM2 is a BASCOE reanalysis, while the other reanalysis products (ERAi, JRA55, MERRA) use well-established assimilation system constrained with observations. I don't see why BRAM2 is consider here as the "truth"?

The part from "The color codes..." until "remain cautious" was moved to the caption of Fig. 5 of the revised manuscript. Regarding BRAM2, it is constrained by  $N_2O$  observations, which is not the case for the CTM nor for any of its 4 driving reanalyses. We do not consider BRAM2 as the "truth" more than we would consider an observational dataset to be the "truth". A whole paragraph explains this in section 2.4, both in the ACPD and revised versions, with the revised version stating:

In the rest of the paper, we will assume that the BRAM2 product provides the best available approximation of the TEM budget for  $N_2O$ , at least where the residual is smaller than the vertical advection and horizontal mixing terms. This assumption relies on the combination in BRAM2 of dynamical constraints from ERA-Interim with chemical constraints from MLS (Errera et al., 2019) ACPD
Furthermore the caption of Figure 5 in the revised manuscript states:

BRAM2 is depicted with a black line and symbols, as usually done for observations, because it is constrained by both dynamical and chemical observations.

65. *Page 9, line 282, replace "We first investigate" by "First, we investigate..."* Done and moved to Sect. 4.1 page 12 line 376.

First, we investigate the  $N_2O$  mixing ratio...

66. Pages 9-10, line 283-285, Is there any possible physical explanation of ERAi underestimation in tropics? Is there any link to the upwelling or extent of the tropical pipe? Or just a different location of the maximum for ERAi compare to JRA-WACMM?

The physical reason behind the underestimation of  $N_2O$  in ERAI compared to JRA55 is the faster upwelling in JRA55 (evaluated by Chabrillat et al., 2018 through mean AoA) compared to ERAI (because of the inverse relationship between  $N_2O$  and mean Age of Air). Unfortunately, we did not have mean AoA output from WACCM to draw similar conclusions for the CCM. This is discussed in Sect. 4.3 of the revised manuscript:

In the tropical regions, the  $N_2O$  mixing ratios in WACCM-CCMI agrees well with the reanalysis of Aura MLS, while the CTM results show large differences in the  $N_2O$  abundances depending on the input reanalysis (Fig. 9(c) and 9(d)). In regions where the mAoA is less than 4.5 years and  $N_2O$ is greater than 150 ppb, i.e. in the tropical regions and lower stratospheric middle latitudes (Strahan et al., 2011), the  $N_2O$  mixing ratio is inverserly proportional to the mAoA, because faster upwelling (younger air) implies more  $N_2O$  transported from lower levels, decreasing its residence time and resulting in a limited chemical destruction (Hall et al., 1999; Galytska et al., **ACPD**
2019). The dynamical reanalyses also produce large differences in mAoA at 15 hPa: MERRA delivers the oldest mAoA and MERRA2, ERAI and JRA55 progressively show younger mAoA (Fig. 4(b) in Chabrillat et al., 2018). Hence the large discrepancies in  $N_2O$  mixing ratio can be explained by the large differences in AoA, while  $M_y$  and  $A_z$  contribute to rates of change of  $N_2O$ .

67. Page 9-10, line 283-287, the discussion is not clear and very hard to follow. Why "the subtropics 40-60" is just not mentioned in the N2O vmr? All panels in the figure have to be discussed, if not please do not show them. It will be clearer and easier to follow if the discussion is done by latitude band e.g. "In the tropic, ...", "In the mid-latitudes, ..." and "In the polar region, ..."

Indeed, the structure was confusing. As mentioned before, we changed the layout of the manuscript, merging the Sections 3 and 4. In the revised manuscript, the Sect. 4 "Climatological seasonal cycles" is divided in three subsections by latitude bands: Polar regions, middle latitudes and Tropics.

- 68. Page 10, line 289, replace "We then investigate" by "Second, we investigate..." Done, and moved to page 15 line 464.
- 69. Page 10, line 322-323, the sentence is not clear and can be split into 2 sentences and formulated clearly.

As a result of the structure of the manuscript, this sentence does not mention anymore the middle altitudes:

 $A_z$  is positive all year round showing the effect of tropical upwelling, and agrees very well in the reanalyses (Figs. 9(i) and (j)), as a result of the good agreement in the tropical upwelling velocity at 15 hPa (Fig. S7 bottom row), and also as depicted by mAoA diagnostics (Fig. 4(d) in Chabrillat et al., 2018).

ACPD
70. Page 10, line 326, add a comma after "Finally". Same after "In the Tropics from Novermber to April (Fig. 6(g))", same after "In the middle latitudes (Fig. 6(h))", same after "In the arctic region (Fig. 6(i))"

With the new manuscript structure, these parts were removed, or moved to the correct places and corrected.

71. In this section 3.2, differences are reported but there is no physically explained attempt.

As mentioned before, we merged the Sections 3 and 4 to address this problem. We reduced the purely descriptive parts, and we moved (and enhanced where possible) the relevant scientific discussion to where it belongs for each figure.

72. Page 11, line replace "After reporting on the climatological annual cycles, it is desirable to estimate their inter-annual variability. To this end," by " To analyse the inter-annual variability of the annual cycle, we..."

Done.

- 73. Redo panel f) and i) of figure 6 in order to get the quantities shown properly. It is not necessary to keep the same y-axis scaling identical for "Az" and "My" terms. Done.
- 74. Page 11, line 341, replace "We first consider" by "First, we consider" Done.
- 75. Page 11, line 342-343, in the [0°, 20°] at 15hPa, BRAMS N2O mixing ratio is more closer to the reanalyses at the first half of the year.

The sentence was rephrased for the  $[0^\circ$  ,  $20^\circ$  ] latitudinal band:
WACCM-CCMI and the BASCOE reanalysis of Aura MLS BRAM2 show very similar variabilities, especially in the southern Tropics.

76. Page 11, line 344, add a comma after "In the northern mid-latitudes (Fig.7(d))"

The middle latitudes were moved to the supplement:

In the middle latitudes of both hemispheres, the inter-annual variability of  $A_z$  and  $M_u$  peaks in winter as its mean value (Fig. S8).

77. Redo panel a) and b) of figure 8.

Done.

78. Page 11, line 345-346, why there is no attempt of physical explanation or to link of the spread to differences in upwelling or tropical pipe in the dataset?

In the revised manuscript, the Sect. 3.3 was merged with the relevant parts of Sect. 4 and the scientific discussion was improved, while some purely descriptive parts were removed.

79. Page 11, line 347, add a comma after "In the middle latitudes (Figs. 7(e) and 7(h))"

This part was removed from the revised manuscript.

- 80. *Page 11, line 348, add a comma after "In the antarctic region (Fig. 8(c))"* This part was removed from the revised manuscript.
- 81. Page 11, line 348-350, what is the physical explanation of the hemispheric differences in the Az and My? The strength of the polar? Sudden stratospheric warming?

The differences between the Arctic and Antarctic are discussed in Sect. 5 of the revised manuscript:
We now look at the interannual variability of  $A_z$  and  $M_y$  in the polar regions. Above the Antarctic, the inter-annual variability of  $A_z$  and  $M_y$  is maximum during spring (Figs. 10(e) and (i)), due to the large inter-annual variability in vortex breakup dates (Strahan et al., 2015). While the maximum variability of  $M_{\mu}$  is consistently reached in October in all the reanalyses, WACCM-CCMI simulates an earlier maximum (September) that does not correspond with the maximum in its mean values (Fig. 5(m)). The lower wintertime variability of both  $A_z$  and  $M_y$  would increase if a longer period was considered to include the exceptional Antarctic vortices of 2002 (Newman and Nash, 2005) and 2019 (Yamazaki et al., 2019). Above the Arctic,  $M_{\mu}$ and  $A_z$  are most variable during winter, reflecting the frequent disruptions of the northern polar vortex by sudden stratospheric warmings (SSWs, Butler et al., 2017). A case study of the effect of a SSW on the  $N_2O$  TEM budget was examined in Randel et al. (1994). They found a stronger  $A_z$  and  $M_{u}$  contribution (among the other TEM terms) during the SSW event than in the seasona

---

## Author Comment (AC2) · 23 Jul 2020

[12pt,a4paper]article hyperref natbib [pdftex]graphicx subfig [textheight=700pt, textwidth=480pt]geometry pbox gensymb

**Response to Reviewer#2 for: Climatological impact of the Brewer-Dobson Circulation on the $N_2O$ budget in WACCM, a chemical reanalysis and a CTM driven by four dynamical reanalyses**

**Minganti et al., ACPD, 2020**

We thank the reviewer for his/her useful comments. In our replies below the italic type is used for the reviewer's comments, the plain text for authors' answers and the bold type for the revised text in the manuscript.

**Replies to general comments.**

*However the manuscript should be highly improved in structure and wording! I have the feeling that in some sections the text lacks an organized structure. E.g. when describing the figures, the text jumps from one figure panel to another and it is really hard to follow. I recommend publication after carefully reading over the text again and rephrasing where it is necessaire*

As recommended by both reviewers, the structure of the manuscript was changed: the Sect. 3 was merged with Sect. 4, and the manuscript was restructured as follows:

Section 1. Introduction

Section 2. Data and Method

Section 3. Latitude pressure cross sections

Section 4. Climatological seasonal cycles

Subsection 4.1 Polar regions

Subsection 4.2 Middle latitudes

This new structure allowed to remove some purely descriptive parts in Sect. 3, and to better follow the text by latitude band when discussing the figures (especially for Fig. 5 and 6). The change in the manuscript structure led to a chage of the layout of the figures as well. We separated them by latitude bands of each hemisphere (one figure for the polar regions, one figure for the surf zones and one for the tropics) in order to better follow the flow of the Section 4 of the revised manuscript and its subsections.

The Introduction was revised as well. Every major concept now gets his own paragraph(s), and some of them were improved, e.g. reanalyses and CTMs, while the paragraph about long-term trends of the BDC was de-emphasized, because this manuscript investigates only climatologies and inter-annual variabilities but not long-term changes.

All these structure changes, together with the reduction of the descriptive parts, intend to improve the wording/phrasing of the manuscript.

**Specific comments/questions.**

1. *-page 1, line 2: reword: " ... from well-mixed tropical troposphere to polar stratosphere....": This is a bit too short, here one has the impression, that tracers are transported directly from trop. troposphere to the polar region.*

   The sentence was rewordedas follows:

   **The Brewer-Dobson Circulation (BDC) is a stratospheric circulation characterized by upwelling of tropospheric air in the Tropics, poleward flow in the stratosphere, and downwelling at mid and high latitudes, with impor-**

**tant implications for chemical tracers distribution, stratospheric heat and momentum budgets and mass exchange with the troposphere.**

2. *-page 1, line 7: insert "in " − > .... in a chemical reanalysis*

    Done.

3. *-page 1, line 10: .... have not been compared ....* before*.*

    Done.

4. *- page 1, line 14: Please clarify, I do not understand the sentence: "....reflecting the large diversity in mean AoA obtained with the same experiments." The present study does not look at AoA with CTM experiments.*

    Here we referred to the study from Chabrillat et al., (2018). They used the same configuration of the BASCOE CTM as for the current manuscript to do Age of Air calculations. Anyway, the sentence was not clear and it is rephrased:

    **....reflecting the large diversity in the mean Age of Air obtained with the same CTM experiments in a previous study.**

5. *- page 2, line 27: include that you compare interannual variability* between the different datasets*.*

    Done.

6. *- page 2, line 33: reword and clarify this sentence to e.g. "The Brewer Dobson Circulation is characterized by upwelling of tropospheric air to the stratosphere in the tropics, followed by .... " . Note however that the BDC includes both residual circulation (net mass transport) and two-way mixing. Moreover the downwelling takes not only place in the high, but also in the mid-latitudes (change to − > extratropical downwelling) and not only in wintertime, although in the respective winter hemisphere it is much stronger.*

The sentence was re-written as follows:

**The Brewer-Dobson Circulation (BDC, Dobson et al., 1929; Brewer, 1949; Dobson, 1956) in the stratosphere is characterized by upwelling of tropospheric air to the stratosphere in the Tropics, followed by poleward transport in the stratosphere and extratropical downwelling. For tracer-transport purposes the BDC is often divided into an advective component, the residual mean meridional circulation (hereafter residual circulation), and a quasi-horizontal two-way mixing which causes net transport of tracers, not of mass (Butchart, 2014).**

7. *- page 2, line 46: Why should mixing be limited to a specific latitudinal region of the winter stratosphere? In the surf zone mixing is only stronger. (see e.g. Fig. 1 in BÃÂűnisch et al. 2011)*

   The sentence was modified and the reference was added:

   **The two-way mixing is stronger in a specific latitudinal region of the winter stratosphere, the "surf zone" (McIntyre and Palmer, 1983), and in the subtropical lower stratosphere all year round (e.g. Fig.1 of BÃÂűnisch et al., 2011).**

8. *- page 2, line 51: change to:"... due to the increase in well mixed greenhouse gases (e.g. Butchart et al 2014,...) and due to increased ozone depleting substances (e.g. Polvani et al. 2018 ...) "*

   Done.

   **....due to the increase in well-mixed greenhouse gases (Butchart et al., 2010; Hardiman et al., 2014; Palmeiro et al., 2014) and ozone-depleting substances (Polvani et al., 2018),....**

9. *-page 3, line 56 and line 63: Here the study of Fritch et al. 2019 (https://www.atmos-chem-phys-discuss.net/acp-2019-974) is interesting.*

The mentioned study is now included in the manuscript:

**The difficulty to derive observational trends in the BDC can be partly attributed to the spatial and temporal sparseness of the observations, together with its large dynamical variability and the uncertainty of trends derived from non-linearly increasing tracers (Garcia et al.; 2011, Hardiman et al., 2017; Fritsch et al., 2019).**

10. *- page 3, line 60: """" observational trends in the ...*

    Done.

11. *- page 3, line 65: Say why is it important to do this separation?*

    This sentence and the previous one ("Furthermore the observational datasets cannot discriminate....") were removed from the revised manuscript as this paragraph was de-emphasized.

12. *- page 3, line 72: Could you write more about the study of Tweedy et al. 2017, as they are also looking at the N2O TEM continuity equation in GEOSCCM!*

    A sentence about Tweedy et al., 2017 was added:

    **In the tropical lower stratosphere, the distinction between vertical and horizontal transport is important, as they impact differently the seasonality of $N_2O$ in the northern and southern Tropics (Tweedy et al., 2017).**

13. *- page 3, line 75: In Abalos et al. 2013 the stratospheric $N_2O$ buget isn't shown.*

    The reference to Abalos et al. (2013) was removed.

14. *- page 3, line 85: change to: ...four different dynamical reanalyses are used here to drive simulations ....*

    The paragraphs about the reanalyses and the CTM were changed. Now the mentioned part states:

**In order to contribute further to the S-RIP BDC activity, four different dynamical reanalyses are used here to drive the BASCOE CTM simulations, compute the $N_2O$ TEM budget and compare its components with the results derived from WACCM. Namely we consider:.....**

15. *- page 3, line 88: Please clarify: Is only WACCM4 compared to BRAM2?*

    Both WACCM and the CTM experiments are compared to BRAM2, this is now explicitly stated:

    **WACCM and the CTM experiments are also compared....**

16. *- page 4, line 93: Are there studies with CTMs driven by reanalyses that studied tracer transport in TEM framework?*

    To our knowledge, a few studies were performed using CTM in the TEM framework, but they used dynamical fields obtained from CCMs and not from reanalyses (e.g. Strahan et al., 1996). Hence they were not deemed relevant to this work and we did not include them in the manuscript.

17. *- page 4, line 107-118: You explain the differences of WACCM-4 and WACCM-CCMI by model development. But are there also differences in the setup of the simulations (e.g. different SSTs, ....)*

    The model setup of WACCM4 was as similar as possible to the CTM experiments, to allow fair comparison. This is now stated in the manuscript:

    **We ran one realization of the public version of WACCM (hereafter WACCM4, Marsh et al., 2013), with a similar setup (e.g. lower boundary conditions) as the CTM experiments;....**

18. *- page 4, section 2: I recommend to include a table to give an overview over the different simulations (CCM, CTM with diff. reanalysis).*

    The table in now included (Table 1).

| Dataset name | Reference | Dynamical Reanalysis | Chemical reanalysis of | Model grid | Top level |
|---|---|---|---|---|---|
| WACCM4 | Marsh et al., (2013) | none | none | 2.5x1.9, L66 | $5.1 \times 10^{-6}$ hPa |
| WACCM-CCMI | Garcia et al., (2017) | none | none | 2.5x1.9, L66 | $5.1 \times 10^{-6}$ hPa |
| ERAI | Chabrillat et al., (2018) | ERA-Interim (Dee et al., 2011) | none | 2.5x2, L60 | 0.1 hPa |
| JRA55 | Chabrillat et al., (2018) | JRA-55 (Kobayashi et al., 2015) | none | 2.5x2, L60 | 0.1 hPa |
| MERRA | Chabrillat et al., (2018) | MERRA (Rienecker et al., 2011) | none | 2.5x2, L72 | 0.01 hPa |
| MERRA2 | Chabrillat et al., (2018) | MERRA2 (Gelaro et al. 2017) | none | 2.5x2, L72 | 0.01 hPa |
| BRAM2 | Errera et al., (2019) | ERA-Interim (Dee et al., 2011) | MLS (Livesey et al., 2015) | 3.75x2.5, L37 | 0.1 hPa |

**Table 1.** Overview of the datasets used in this study.

19. *- page 5, line 132: WACM −> WACCM*

    Done.

20. *- page 5, line 137: ... as input...*

    The sentence was changed:

    **Chabrillat et al. (2018) explain in detail the preprocessing procedure that allows the BASCOE CTM to be driven by arbitrary reanalysis datasets, and the set-up of model transport.**

21. *- page 6, line 161: What do you mean with situation of interest?*

    "Situation of interest" was indeed misleading, a more appropriate wording would be "regions of interest". BRAM2 has been evaluated in several regions of interest in the middle atmosphere as defined in the BRAM2 paper (Errera et al., 2019): the middle stratosphere (MS) the tropical tropopause layer (TTL), the lower stratospheric polar vortex (LSPV) and the upper stratosphere polar vortex (USPV). The chemical species were only evaluated in some relevant regions, and BRAM2 $N_2O$ was evaluated in MS, LSPV and USPV. The text was rewritten more clearly:

    **BRAM2 $N_2O$ has been validated between 3 and 68 hPa against several instruments with a general agreement between 15 % depending on the instrument and the atmospheric region (the middle stratosphere or the polar**

**vortex, see Errera et al., 2019).** .

22. *- page 6, line 182: "$N_2O$ balance" $->$ In this section you use tracer X to explain the TEM diagnostics, but here you change back to N2O. Perhaps you use N2O instead of X in the entire section?*

    We now use $\chi$ in all the formulas, and "$N_2O$ balance" was changed to "tracer balance". Furthermore, we stated explicitly that $\chi$ represents the $N_2O$ concentrations in the revised manuscript:

    **...where $\chi$ is the volume mixing ratio of $N_2O$,...**

23. *- page 7, line 200: Can you be a bit clearer, please: You are giving the causes of the non-zero residual for WACCM, but what about the residuals in the CTM, and the chemical reanalysis? Is it only the timestep in BASCOE?*

    Regarding the reanalyses, the reason for the large residual could be the coarser resolution compared to their input reanalyses (especially for BRAM2), impacting the numerical errors in the the horizontal and vertical derivatives that are involved in the TEM analysis. For this reason, a new reanalysis of Aura MLS is planned (BRAM3) with the same horizontal and vertical resolution as in the CTM. The unresolved mixing can also play a large role, as discussed in Sect. 3 of the revised manuscript. Taking into account these two factors, the text was rewritten:

    **The BASCOE datasets have a coarser horizontal resolution than their input reanalyses (especially BRAM2; see Table 1). This affects the accuracy of the vertical and horizontal derivatives, with possible implications for the residual. The possible causes of the residual in all the reanalyses are discussed in more detail in Sect. 3**

24. *-page 7, line 205: "...while ..." $->$ "...even though ..."*

    Done.

25. *- page 7, line 209: Note that Tweedy et al. 2017 looked at N2O TEM buget at 85 hPa in the tropics.*

    In the revised manuscript, it is stated more clearly that they looked in the tropical lower stratosphere:

    **In order to validate our $N_2O$ TEM budget, we reproduced the findings reported in Tweedy et al. (2017, Fig. 7) with WACCM-CCMI in the tropical lower stratosphere, and we noticed similar results (not shown).**

26. *- page 7, line 213: Why does $w^*$ vary in reanalyes data? Perhaps you can add one sentence more about Abalos et al. 2015.*

    The sentence was slightly modified to include the main physical reason of the disagreement:

    **The upwelling velocity $\bar{w}^*$ can vary considerably in the dynamical reanalyses, , as it is a small residual quantity (Abalos et al., 2015).**

27. *-page 8, line 219: delete "the" $->$ .... are strongest ...*

    The sentence was removed from the revised manuscript.

28. *- page 8, line 220: You motivate the choice of the 15 hPa level with large differences between the CCM and CTM simulations in this region. Where do you see this? I suppose in Figs. 3 +4. And why isn't it interesting to see what is going on in the lower stratosphere?*

    Indeed those differences can be seen from Figures 3 and 4. We didn't look at the lower stratosphere because the vertical range of validity for BRAM2 is limited to 3-68 hPa (Errera et al., 2019).

29. *- page 8, line 16: The terms, "vertical advection", "horizontal mixing" and their abbreviations Ay and My are mixed within the manuscript, even between one*

*sentence these terms are mixed (e.g. page 8, line 225). Can you please use the terms consistently?*

In the description of Figs. 1 and 2, we kept using the full names and their abbreviations (e.g. the vertical advection term $A_z$) as we explain the methodology in that section. In the rest of the manuscript we use the abbreviations $A_z$ and $M_y$.

30. *-page 8, line 226: "higher latitudes" $->$ I can see this mainly in the northern higher latitudes.*

The description of Figs. 1 and 2 was largely reduced in order to remove purely descriptive sentences such as this one (lines 226-229).

31. *page 8, line 232 (and also line 229):"... especially in the reanalyses Az and the residual play a minor role": I wouldn't say, that this effect is "minor"!*

The whole paragraph was re-written, see comment above. Figures 1 and 2 are now described and discussed as follows:

**Figs. 1 and 2 show the $N_2O$ TEM budget terms at 15 hPa for all the datasets for the boreal winter (December-January-February, DJF mean) and summer (June-July-August, JJA mean) respectively. The 15 hPa level (around 30 km altitude) was chosen because large differences can be found between WACCM-CCMI, BRAM2, and the CTM runs at this level, and because the dynamical reanalyses are not constrained as well by meteorological observations at higher levels (Manney et al., 2003). Figs. 1 and 2 aim to show how the dynamical and chemical terms of the budget balance each other to recover the tendency $\bar{\chi}_t$ at different latitudes. The discussion about the differences between the datasets, and their possible physical causes, are addressed in the next Sections.**

**The vertical advection term $A_z$ shows how the upwelling contributes to increasing the $N_2O$ abundances in the tropics and summertime mid-latitudes, and how polar downwelling contributes to decreasing the $N_2O$**

[Figure]

**abundances in the winter hemisphere. The horizontal transport out of the tropics due to eddies, as represented by $M_y$, reduces the $N_2O$ abundance in the tropical latitudes of the wintertime hemisphere, and increases the $N_2O$ mixing ratio at high latitudes in the winter hemisphere. The other terms of the TEM budget are weaker than $A_z$ and $M_y$: the meridional advection term $A_y$ tends to increase the $N_2O$ abundance in the winter subtropics and extra-tropics, while the vertical transport term due to eddy mixing, $M_z$ decreases it over northern polar latitudes and the chemistry term $P-L$ shows that $N_2O$ destruction by photodissociation and $O(1D)$ oxidation contributes to the budget in the tropics and also in the summertime hemisphere. All budget terms are weaker in the summer hemisphere than the winter hemisphere. Over the southern polar winter latitudes, the reanalyses deliver negative $M_y$ that are balanced by large positive residuals, which implies a less robust TEM balance (Fig. 2). This is not the case with WACCM, where $M_y$ tends to increase the $N_2O$ abundance in the polar vortex. Such differences between the datasets are highlighted and discussed in the next sections.**

32. *- page 8 line 238: spelling: reanalyses*

    Done.

33. *- page 9, line 253: You only show thee reanalyses here, not four.*

    "...in the four reanalyses" was replaced by "...in the other reanalyses".

34. *-page 9, line 266: middle stratospheric $->$ middle stratosphere*

    Done.

35. *-page 9, line 257: "(Fig. 3(f), (i), (l))" $->$ right columns of Fig. 3*

    Done.

36. *-page 9, line 269: Motivate why you are choosing a single level in the middle stratosphere (15 hPa). What about the lower stratosphere?*

We tried several levels in the middle stratosphere and found that the differences between the datasets were most visible at 15 hPa while other levels did not bring added value to the intercomparison. With respect to the lower stratosphere, see reply 28 above.

37. *-page 9-11, description of the climatological seasonal cycles: In my opinion this section is very hard to read, as the SH and NH are separated into two pictures. I recommend to merge Fig.5 and 6 to one Figure and then describe first the tropical, mid-latitude and polar N2O (upper raw), second the vertical advection Az (middle row) and third horizontal mixing My (bottom raw). Thus it is easier to see the differences in NH and SH, the text is better structured and you do not have to repeat patterns that are similar.*

In order to follow this comment and another major comment by the first reviewer, Figs. 5 and 6 were re-organized into three figures, each of them covering both hemispheres. The revised Fig. 5, 7 and 9 show respectively the polar regions, mid-latitudes and tropics and are discussed in sections 4.1, 4.2 and 4.3 respectively. This new structure avoids any repetition while showing simultaneously, for each latitude band, the N2O cycle and the two main terms contributing to its TEM budget. Fig. 9 was also split into latitude regions and inserted as revised figs. 6 and 8, to contribute to the interpretation of our results in the polar regions and mid-latitudes. The tropical regions of Fig. 9 were moved to the Supplement.

38. *- page 9, line 278-281: What do you mean with uncertainty - the 1 sigma standard deviation?*

Yes indeed, as stated in Errera et al., (2019). This sentence was moved to the caption of Fig. 6 following a comment from Reviewer 1.

39. *- page 9, line 282: " We first investigate the $N_2O$ mixing ratio in the SH. In the tropic (Fic 5c and 6a)...." − > Fig. 6a is not in the SH!*

    After the rearrangement of the sections explained above, this sentence is not limited to the SH any more:

    **In the tropical regions the $N_2O$ mixing ratio (Figs. 9(c) and (d)) in WACCM-CCMI ....**

40. *-page 9, line 283: Please point out here more clearly, that BRAM2 is used as reference, and that this is the case for the entire section.*

    This is pointed out more clearly after the structure rearrangement:

    **In the following, we will consider BRAM2 as the reference when comparing $N_2O$ mixing ratios between datasets.**

41. *-page 10, line 286: change to: ...is smaller than in BRAMS in all simulations.*

    Done.

42. *-page 10, line 284-288: You missed to describe the mid-latitudes....*

    With the new manuscript structure, the middle latitudes are now discussed in a dedicated Sect. 4.2.

43. *-page 10, line 287: You wanted to talk about $N_2O$, not about Az and My...*

    That paragraph was confusing indeed. Now the discussion of the middle latitudes is put together in Sect 4.2. It starts with the $N_2O$ mixing ratio in both hemispheres, and continues with $A_z$ and $M_y$ for each hemisphere.

44. *-page 10, line 300: "...expect for JRA55" − > expect JRA55*

    Done.

45. *-page 10, line 305: "It is yet comparable..." − > What? The uncertainty.*

    The sentence was removed from the revised manuscript, as it did not add any relevant scientific point.

46. *-page 10, line 311: Replace differ to different.*

    Done.

47. *-page 11, line 337: Do you use the 1-sigma standard deviation?*

    Yes indeed. The text could be more precise, as implicitly suggested by the reviewer. The revised sentence now states:

    **... we compute for each month the 1-sigma standard deviations of the** $N_2O$ **mixing ratio,** $M_y$ **and** $A_z$ **across the ten simulated years.**

48. *-page 11, line 335-340: I think it is easier for the reader if you plot the standard deviation the same way as in Fig. 5+6. I do not see a real advantage of plotting the results in this order. And as recommended before it would be nice to have Fig. 7+8 in one plot and restructure the text accordingly.*

    We merged Figs. 7 and 8 into Fig. 10 of the revised manuscript. The text was restructured accordingly, and according to the new sections layout.

49. *-page 11, line 343: Why does the variability in WACCM-CCMI strongly depends on the considered realization? Shouldn't the internal variability between these ensemble simulations be similar?*

    This was a surprising result, as in the other latitude bands the internal variability of WACCM does not play a major role. Strong differences between ensemble members with respect to inter-annual variability indicate that the considered period is not long enough to explore the inter-annual variability in the northern mid-latitudes, and that the mean variability from this ensemble (with only 3 members) would not be representative of the internal variability of WACCM. Fortunately, our

study did not investigate the ensemble mean but showed instead the full range from the 3 WACCM realizations. This will be stated in the revised manuscript.

50. *-page 12, subsection "polar regions": The structure of this subsection was not clear to me during reading: you first write about the wintertime North Pole, then about the wintertime South, then you jump to the SH spring and to Antarctic and Arctic inter-annual variability. Perhaps you can give an introducing sentence of what you will discuss in this section.*

    With the new structure of the manuscript mentioned above, this subsection was merged with Sect. 3, and all the information (wintertime North Pole, wintertime South,...) were moved to the right places when describing the figures.

51. *-page 12, line 375: What do you mean with "Above the Arctic in the middle stratosphere ... (Fig.6)"? Do you refer to the 15 hPa level in Fig. 6?*

    Yes. This paragraph was moved to Sect. 4.1 of the revised manuscript.

52. *-page 12, line 376: I cannot see that $N_2O$ abundance in polar regions (Fig. 6c) are in good agreement in WACCM and BRAMS in the wintertime ...*

    The reviewer is right, and sentence was modified:

    **Above the Arctic in the middle stratosphere, the $N_2O$ abundances simulated by WACCM agree with the BRAM2 reanalysis, except in December and January, and....**

53. *-page 12, line 379: Compared to which reanalysis? To all? Before you were comparing with BRAMS.*

    Yes, we consider here all the reanalyses. The text was modified accordingly:

    **Compared to the dynamical reanalyses and BRAM2,....**

54. *-page 12, line 381: Replace "Fig. 6 bottom raw", to Fig. 6 g+h. And why are you talking about tropics and mid-latitudes here? In this chapter you wanted to discuss the polar regions.*

  The references to Tropics and mid-latitudes were removed as a consequence of the manuscript structure. The sentence was re-written:

  **Compared to the dynamical reanalyses and BRAM2, WACCM shows in the Arctic a 2-fold underestimation of the $N_2O$ changes due to horizontal mixing during winter.**

55. *-page 13, line 383: Do you mean the aging by mixing term in the polar regions of Fig. 2 in Dietmã¼ller et al. 2018? Moreover reword "Note that ..." This is a poor transition between the two sentences.*

  Yes, we mean aging by mixing. The sentence was modified for clarity:

  **It should be emphasized that WACCM is among the CCMI models with the lowest contribution of aging by mixing to Age of Air (Fig. 2 in Dietmuller et al., 2018).**

56. *-page 13, line 386: Include that TEM AoA buget was done in CCM simulations.*

  Done.

  **Dietmuller et al. (2017) applied the TEM continuity equation to the Age of Air (AoA) in CCM simulations.**

57. *-page 13, line 391: Can you explain, why the TEM formulation is different in this study?*

  Our formulation was misleading. The differences arise only from the different nature of AoA and $N_2O$: AoA does not have chemical sources nor sinks in the stratosphere, while $N_2O$ is destroyed in the tropical higher stratosphere. Since

the definition of the dynamical TEM terms does not change, we removed "with a different TEM formulation", and the sentence now reads:

**Even though we use a real tracer ($N_2O$), we find a qualitative agreement with this analysis based on AoA: our residual term is larger in regions characterized by strong gradients such as the antarctic vortex edge, and larger with dynamics constrained to a reanalysis than with a free-running CCM (see EMAC results in Fig. 1d by Dietmuller et al., 2017).**

58. *-page 13, line 392: "... agreement: our residual term is larger ..." But you are listing the differences here.*

The second difference ("with a different TEM formulation") was removed. The point of this paragragh is that we find qualitative agreement between their "aging by diffusion" and our residual term, since both are computed as the remaining of the respective TEM budgets. We hope that the revised sentence makes this clearer (see previous comment).

59. *-page 13, line 396: Perhaps change to "....SH winter". (Also in other parts of the paper)*

The sentence was re-written:

**In the austral winter, over the Antarctic Polar cap and below 30 hPa, $M_y$ agrees remarkably well in all datasets (Fig. 4).**

60. *-page 13, line 397: Again: What do you mean with "above 30 hPa"? Do you mean the 15 hPa level (latitude band 60-80S), as you are refering to Fig. 5?*

We referred to Fig. 4 of the ACPD manuscript. This sentence was moved and adapted to Sect. 3 in the revised manuscript, where it still refers to Fig. 4; this is now clearer because Fig. 5 is introduced only in the next section.

61. *-page 13, line 399: You are talking about Fig. 4, not about Fig 5!*

Same reply as for the previous comment.

62. *-page 13, line 401: Are these studies are giving an explanation for the mixing inside the vortex. If yes, can you please give the explanation here.*

De la Camara et al., 2013 states that the Rossby waves breaking can contribute to the tracer mixing inside the polar vortex and occasionally across its edge. The sentence was re-written as:

**The impact of horizontal mixing on $N_2O$ inside the wintertime polar vortex is not negligible (e.g. de la Camara et al., 2013; Abalos et al., 2016a), as Rossby waves breaking occurs there as well as in the surf zone.**

63. *-page 13, line 403: Make clear, that it is overestimated in WACCM ... (and overestimated according to what?)*

Garcia et al. (2017) compared the winds simulated by WACCM to the winds from MERRA. This is stated more precisely in the revised manuscript:

**This disagreement can be related to differences in the zonal wind: it is overestimated in WACCM above 30 km in subpolar latitudes compared to MERRA (Garcia et al., 2017) and the polar jet is not tilted equatorward as in the reanalyses (see black thin lines in Fig. 4, and Fig. 3 of Roscoe et al., 2012).**

64. *-page 13, line 404: Change to : ... (see black thin lines in Fig. 4).*

Done.

65. *-page 13, line 405: You do not show the residual terms in Fig. 5.*

The sentence refers to Fig. 4, as the residual terms were not shown in Fig. 5. The sentence was moved to Sect. 3 of the revised manuscript, and changed:

**Yet, the differences in $M_y$ and $A_z$ above the Antarctic in winter should be put into perspective with the large residual term that points to an incomplete TEM budget (Fig. 4 right column).**

66. *-page 13, line 408: Say, why you are now looking at SH spring.*

Indeed, that change to SH spring was confusing as it was not introduced. After the change in the structure of the manuscript, this part was moved to Sect. 4.1 of the revised manuscript, and now it follows the discussion of the wintertime $M_y$ at 15 hPa over the antarctic.

67. *-page 13, line 409:"... better agreement ..." Better compared to what?*

After the change in the manuscript structure, this sentence was moved to Sect. 4.1 of the revised manuscript for the description of Fig. 6:

**During the austral spring, the vortex breakup leads to an increased wave activity reaching the Antarctic (Randel and Newman, 1998), and mid-stratospheric $M_y$ is in better agreement among all datasets compared to austral winter.**

68. *-page 14, line 418: Replace "reanalyses" with dynamical reanalyses. And why is BRAM2 not included in this comparison?*

The word "reanalyses" was replaced by "dynamical reanalyses". BRAM2 is not included in this comparison because it is dynamically constrained to the winds from the ERA-Interim reanalysis, and its results are nearly identical with those of the CTM simulation driven by ERA-Interim, i.e. these differences are only due to the coarser resolution of BRAM2 and they are negligible.

69. *-page 14, line 434: Please explain critical lines.*

This is explained in the revised manuscript as follows:

> **It is due to transient Rossby waves that cannot travel further up into the stratosphere due to the presence of critical lines, i.e. where the phase velocity of the wave matches the background wind velocity, generally leading to wave breaking (Abalos et al., 2016b).**

70. *-page 14, line 448: vmr − > mixing ratio*

     Done.

     *Comments to the Figures:*

71. *- Fig. 1+2: Can you please replace "time der" to dN2O/dt in the legend.*

     Done.

72. *- You are showing different colorbars in Fig. 3 and 4!*

     We now use the same color scale [-2,2] ppbv/day for both figures.

73. *-Fig 5+6, y-axis: Replace X with $N_2O$.*

     Done.

**References**

Fritsch, F., Garny, H., Engel, A., Bönisch, H., and Eichinger, R. (2019). Sensitivity of age of air trends on the derivation method for non-linear increasing tracers. *Atmospheric Chemistry and Physics Discussions*, 2019:1–23.

Strahan, S., Nielsen, J., and Cerniglia, M. (1996). Long-lived tracer transport in the antarctic stratosphere. *Journal of Geophysical Research: Atmospheres*, 101(D21):26615–26629.